# CHARACTERISTIC ROOT ANALYSIS AND REGULARIZATION FOR LINEAR TIME SERIES FORECASTING

**Zheng Wang**[1,†,*]**, Kaixuan Zhang**[1,†]**, Wanfang Chen**[1,†]**, Xiaonan Lu**[1,†]**,
Longyuan Li**[1,†]**, Tobias Schlagenhauf**[2]
[1]Bosch Center for AI (BCAI) & Bosch (China) Investment Co., Ltd., [2]Robert Bosch GmbH
[†]{david.wang3, kaixuan.zhang, wanfang.chen, xiaonan.lu
  longyuan.li}@cn.bosch.com

## ABSTRACT

Time series forecasting remains a critical challenge across numerous domains, yet the effectiveness of complex models often varies unpredictably across datasets. Recent studies highlight the surprising competitiveness of simple linear models, suggesting that their robustness and interpretability warrant deeper theoretical investigation. This paper presents a systematic study of linear models for time series forecasting, with a focus on the role of characteristic roots in temporal dynamics. We begin by analyzing the noise-free setting, where we show that characteristic roots govern long-term behavior and explain how design choices such as instance normalization and channel independence affect model capabilities. We then extend our analysis to the noisy regime, revealing that models tend to produce spurious roots. This leads to the identification of a key data-scaling property: mitigating the influence of noise requires disproportionately large training data, highlighting the need for structural regularization. To address these challenges, we propose two complementary strategies for robust root restructuring. The first uses rank reduction techniques, including **Reduced-Rank Regression (RRR)** and **Direct Weight Rank Reduction (DWRR)**, to recover the low-dimensional latent dynamics. The second, a novel adaptive method called **Root Purge**, encourages the model to learn a noise-suppressing null space during training. Extensive experiments on standard benchmarks demonstrate the effectiveness of both approaches, validating our theoretical insights and achieving state-of-the-art results in several settings. Our findings underscore the potential of integrating classical theories for linear systems with modern learning techniques to build robust, interpretable, and data-efficient forecasting models. The code is publicly available at: https://github.com/Wangzzzzzzzz/RootPurge.

## 1 INTRODUCTION

Time series forecasting is a foundational task in a wide range of critical applications, including finance, weather prediction, traffic modeling, and energy systems (Hamilton, 2020). Despite its importance, long-term forecasting remains a particularly challenging problem due to inherent uncertainty, noise, and complex temporal dependencies in real-world data (Kong et al., 2025). In response, the research community has devoted substantial efforts to developing increasingly sophisticated model architectures (Woo et al., 2023; Xu et al., 2023; Zhou et al., 2022; Nie et al., 2022), from deep recurrent networks to attention-based transformers, in an attempt to capture long-range dependencies and improve accuracy.

However, as emphasized in the position paper (Brigato et al., 2025), no single model consistently outperforms others across all datasets and forecasting horizons. This observation echoes the *"No Free Lunch" theorem* (Adam et al., 2019): in the absence of strong assumptions or domain-specific priors, it is impossible to design a universally superior forecasting model. This limitation prompts a rethinking of current approaches and highlights the need for more fundamental, theory-driven insights into what makes a time series model effective.

---

*Corresponding to david.wang3@cn.bosch.com

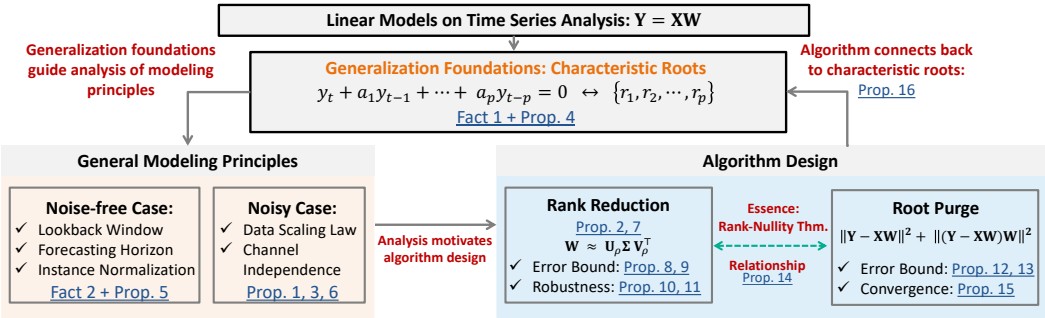

Figure 1: Structure of the paper and its main contributions.

Recent studies show that linear systems, despite their simplicity, often match or exceed the performance of complex nonlinear models, particularly in long-term forecasting (Zeng et al., 2023; Toner and Darlow, 2024; Li et al., 2023). Their robustness, interpretability, and analytical clarity make them a strong foundation for forecasting. Motivated by this, we develop a theoretical framework to study the core properties of linear systems, focusing on the role of dominant characteristic roots that encode the essential structure of the data. At the same time, real-world time series are rarely clean; they are typically contaminated with noise that obscures the underlying structure (Lim and Zohren, 2021). Complex deep learning models are especially prone to overfitting such noise, often resulting in poor generalization and unreliable forecasts. To address this, we draw inspiration from recent findings (Shi et al., 2024) showing that only a small subset of data components—those capturing the essential structure—meaningfully impact the prediction accuracy. In linear dynamical systems, these correspond to dominant characteristic roots that govern the system behavior. Our approach focuses on identifying these core roots while systematically suppressing noise and spurious dynamics.

In this work, we present a systematic analysis of linear models for time series forecasting, emphasizing the role of characteristic roots in determining model expressivity and dynamic behavior (refer to Figure 1 for a road map of this paper). We begin by examining how forecasting horizon and lookback window interact with characteristic roots in noise-free settings, showing that common practices such as instance normalization and channel-independent modeling naturally arise from this framework. We then extend our analysis to noisy settings, where models tend to learn spurious roots—artifacts of noise that distort prediction and obscure true dynamics. This reveals a key data-scaling property: mitigating the impact of noise demands significantly more training data. This reduces data efficiency and underscores the need for structural regularization. Motivated by these insights, we propose two complementary strategies for robust root identification. The first leverages rank reduction, including **Reduced-Rank Regression** (Izenman, 1975) and **Direct Weight Rank Reduction**, to enforce a low-dimensional structure aligned with the latent dynamics. The second introduces **Root Purge**, a novel adaptive training method that promotes the learning of an appropriate null space, actively suppressing noise while preserving informative signal components. We evaluate both approaches on a range of standard time series forecasting benchmarks and demonstrate that they consistently enhance model robustness and accuracy, often achieving state-of-the-art performance. Side experiments and controlled toy examples further validate our theoretical insights. Overall, our findings emphasize the value of integrating classical linear system theory with modern optimization techniques, and pave the way for future work on robust, interpretable, and scalable forecasting models. The contributions in this paper can be summarized as follows:

- **Theoretical Analysis**: We provide a systematic study of linear models for time series forecasting, analyzing the role of characteristic roots in both noise-free and noisy settings, and uncovering a key data-scaling property that motivates structural regularization.

- **Proposed Methods**: We introduce two strategies for robust root identification: rank reduction techniques to enforce a low-dimensional structure, and Root Purge, a novel adaptive training method that suppresses noise via null-space learning.

- **Empirical Validation**: We verify the effectiveness of our approaches through strong performance on standard benchmarks. We also validate our theory with additional controlled experiments.

## 2 BACKGROUND

**Related Work.** Time series forecasting aims to predict future values by learning patterns like trends and seasonality from historical data. Various modeling paradigms have emerged to tackle long-term forecasting challenges (Kong et al., 2025). Transformer-based models have been prominent: Nie et al. (2022) segments series into patch-level tokens with channel-wise modeling, while Zhou et al. (2022) combines seasonal-trend decomposition with transformers. Woo et al. (2023) proposes time-index models that condition on future timestamps to support both interpolation and extrapolation. Beyond time-domain methods, frequency-based approaches like Xu et al. (2023) apply low-pass filtering and complex linear layers to enhance long-term modeling. In parallel, minimalist models have gained traction: Lin et al. (2024) achieves strong results with just 1,000 parameters via structured sparsity, and Zeng et al. (2023) shows that simple linear models can outperform transformers by separately modeling trend and seasonality. Recent evaluations (Brigato et al., 2025) show that no single method dominates across all settings, emphasizing the need for diverse benchmarks and context-aware model selection. Motivated by this, we pursue a theoretical perspective on time series forecasting, aiming for models that balance simplicity, interpretability, and robustness across diverse temporal dynamics. To see a detailed related work and further background, please refer to Appendix B.1.

**Problem Formulation.** Let $\{\mathbf{y}_t\}_{t=1}^T$ be a multi-channel time series of length $T$, where each observation $\mathbf{y}_t \in \mathbb{R}^m$ represents an $m$-dimensional vector at time $t$. The goal of multi-horizon forecasting is to learn a mapping $f$ that predicts future values as

$$\hat{\mathbf{Y}}_{t+1:t+H} = f(\mathbf{y}_t, \mathbf{y}_{t-1}, \ldots, \mathbf{y}_{t-L+1}),$$

where $\hat{\mathbf{Y}}_{t+1:t+H} = [\hat{\mathbf{y}}_{t+1}, \hat{\mathbf{y}}_{t+2}, \ldots, \hat{\mathbf{y}}_{t+H}]$ is an $m \times H$ matrix of forecasts. Here, $H$ is the length of forecasting horizon, $L$ is the length of lookback window, and $f$ is the forecast model, which is required to capture temporal dependencies, account for noise and non-stationarity, and generalize historical patterns to unseen future data.

**Preliminary.** We consider the class of single-channel (i.e., $m = 1$) *homogeneous linear difference equations* due to their analytical tractability in modeling temporal dependencies, taking the form:

$$y_t + a_1 y_{t-1} + \cdots + a_p y_{t-p} = 0,$$

where $p$ is the order of the difference equation and $\{a_i\}_{i=1}^p$ are constant coefficients. The general solution[1] at time step $t$ is a linear combination of *characteristic roots* raised to power $t$:

$$y_t = C_1 r_1^t + C_2 r_2^t + \cdots + C_p r_p^t,$$

where $\{C_i\}_{i=1}^p$ are constants determined by initial conditions, and $\{r_i\}_{i=1}^p$ are obtained by solving the *characteristic equation*: $r^p + a_1 r^{p-1} + \cdots + a_p = 0$.

## 3 THEORETICAL ANALYSIS

Linear dynamical models strike a valuable balance between analytical simplicity and expressive power, making them effective at capturing core temporal patterns such as trends, oscillations, and exponential behaviors. In this section, we explore the foundations of linear modeling in time series forecasting. Our goal is to understand the principles governing linear dynamics through the lens of characteristic roots, enabling us to interpret, justify, and generalize widely used modeling choices.

We begin by formalizing the core problem following standard conventions. Given a normalized single-channel time series, we define linear forecasting as the solution to the least-squares objective[2]:

$$\min_{\mathbf{W}} \|\mathbf{Y}_{\text{fut}} - \mathbf{Y}_{\text{his}} \mathbf{W}\|_F^2, \tag{1}$$

where $\mathbf{Y}_{\text{his}} \in \mathbb{R}^{N \times L}$ and $\mathbf{Y}_{\text{fut}} \in \mathbb{R}^{N \times H}$ represent the collection of $N$ history and future segments from the same normalized sequence, $\mathbf{W} \in \mathbb{R}^{L \times H}$ represents the coefficient matrix. Each row in $\mathbf{Y}_{\text{his}}$

---

[1]Here we consider the simplest case where all characteristic roots are distinct.

[2]The omission of a bias term is justified by the fact that linear difference equations with constant biases can be algebraically transformed into equivalent homogeneous systems; see Appendix B.3 and C.1 for justifications.

and $\mathbf{Y}_{\text{fut}}$, denoted as $\mathbf{y}_{\text{his}} \in \mathbb{R}^L$, $\mathbf{y}_{\text{fut}} \in \mathbb{R}^H$, corresponds to one observed history and future segment, respectively. Solving Equation (1) is equivalent to independently estimating the $j$-th column of $\mathbf{Y}_{\text{fut}}$, corresponding to the $j$-th forecasting horizon ($j$ steps ahead to the future), by solving:

$$\min_{\mathbf{W}} \sum_t \left( y_{t+j} - \sum_{k=1}^{L} W_{kj} \cdot y_{t-k+1} \right)^2, \qquad j = 1, \ldots, H. \tag{2}$$

This formulation reflects a regression-based interpretation of temporal forecasting and serves as a direct proxy for learning the underlying recurrence relations.

## 3.1 Role of Characteristic Roots and Noise-Free Linear Modeling Principles

Recall from Section 2 that for linear models governed by difference equations, solutions are determined by their characteristic roots. Therefore, we can arrive at Fact 1 as follows, derived from the properties of general solutions. This highlights a key generalization property of linear models: the ability to cover a wide range of temporal behaviors (details in Appendix D.1, Corollary 1, and Corollary 2) through appropriate choice of roots, rather than through complex parameterization.

> **Fact 1.** *A linear model can represent any time series whose characteristic roots are a subset of its own.*

For better illustration of this fact, we provide a toy example in Appendix C.3. Based on Fact 1, in the noise-free setting, the optimization objective defined in Equation (1) can often achieve zero given appropriate model capacity. This allows us to convert a linear time series model into a difference equation and study the impact of key design choices of forecasting horizon and lookback window.

It is common practice (Xu et al., 2023; Lin et al., 2024) to model each forecast step independently as in Equation (2), resulting in $H$ separate regression problems, each corresponding to a specific forecasting horizon. Further, it is common that we use a long lookback window length $L$, whereas the underlying process follows a simpler minimal recurrence relation of order $K$, with $K < L$. Both of these choices introduce redundancy in the input representation. Nonetheless, this redundancy is not detrimental to learning and can, in fact, offer flexibility in parameterization. We summarize our key insight as follows, with more details in Appendix C.2 and D.2:

> **Fact 2.** *The characteristic root set of a linear higher-horizon model, or one with an extended lookback window, always preserves, as a subset, the roots that govern the fundamental system dynamics.*

This observation offers theoretical support for independently modeling each horizon, as higher-horizon models remain consistent with the system's true dynamics. Moreover, increasing the lookback window does not alter the set of roots but rather introduces multiple equivalent representations.

We highlight two common time series modeling techniques that align with our framework above, with more details in Appendix C.4. A further empirical study of Channel-independent (CI) modeling and an alternative technique called INC is provided in Table 3, Section 5.4.

**Remark 1.** *Instance normalization introduces a unit root, allowing the model to generalize across sequences with arbitrary mean shifts. Channel-independent modeling remains effective when the model has sufficient degrees of freedom to capture the union of characteristic roots across all channels.*

## 3.2 Data Scaling Property Under Noisy Observations

In practical applications, time series data are often contaminated with stochastic noise, which presents a fundamental challenge for model estimation and generalization. In the presence of noise, the least-squares loss for a linear forecasting model takes the following form for a single segment:

$$\mathbb{E} \left[ \| (\mathbf{y}_{\text{fut}}^* - \mathbf{W}^\top \mathbf{y}_{\text{his}}^*) + (\boldsymbol{\varepsilon}_{\text{fut}} - \mathbf{W}^\top \boldsymbol{\varepsilon}_{\text{his}}) \|_2^2 \right],$$

where $\mathbf{y}_{\text{fut}}^*$, $\mathbf{y}_{\text{his}}^*$ are noise-free history and future segments, respectively, and $\boldsymbol{\varepsilon}_{\text{his}}$, $\boldsymbol{\varepsilon}_{\text{fut}}$ denote additive noise on these segments. This loss naturally decomposes into two components: signal fitting error and noise-induced error. In the *over-idealized* case where the learned weight matrix $\mathbf{W}$ perfectly recovers the signal dynamics, only the noise term $\mathbb{E} \left[ \| \boldsymbol{\varepsilon}_{\text{fut}} - \mathbf{W}^\top \boldsymbol{\varepsilon}_{\text{his}} \|_2^2 \right]$ remains.

To better understand the noise sensitivity of mean squared error (MSE)-based linear models, we analyze a simplified setting in which both inputs and outputs consist solely of Gaussian noise. The following proposition summarizes such asymptotic behavior (see details in Appendix C.5 and D.3):

> **Proposition 1.** *For a linear model forecasting a zero-mean noise with finite second moments, the learned weights converge at a rate proportional to $\mathcal{O}(1/\sqrt{T})$, where $T$ is the length of the observed time series.*

This result illustrates an important limitation of classical least squares training: despite its consistency and unbiasedness, convergence is slow under noise. Even with perfect signal recovery, noise effects decay only at a sublinear rate. This scaling behavior highlights that MSE-based training is data-inefficient in high-noise regimes, requiring a substantial number of observations to achieve low-variance estimates of the underlying dynamics.

## 4    Noise-Aware Linear Forecasting via Root Restructuring

In noisy environments, the data-scaling property highlights a key limitation of linear forecasting models: robustness to noise requires a substantial amount of data. This challenge is compounded when the model must also accurately recover the underlying signal dynamics, as misidentifying them leads to fitting errors. From the perspective of characteristic roots, reliable forecasting hinges on correctly identifying and preserving the roots that capture the signal's temporal structure. However, noise can obscure these roots. To address this, we next present two complementary approaches designed to improve root restructuring and enhance robustness in practical settings.

### 4.1    Rank Reduction Methods

In time series forecasting, the lookback window length $L$ and the structure of the underlying dynamics jointly determine the model's expressive capacity. In the absence of noise, the matrix $\mathbf{Y}_{\text{his}}^{*}$ formed by slicing a deterministic time series has rank $\min(L, K)$, where $K$ denotes the number of characteristic roots. However, once noise is introduced, this low-rank structure becomes obscured—the observed data matrix typically becomes full rank with high probability, even when the underlying signal remains confined to a lower-dimensional subspace.

This observation motivates the use of *rank reduction* as a denoising strategy. A straightforward, though suboptimal, approach involves applying truncated singular value decomposition (SVD) directly to the history matrix $\mathbf{Y}_{\text{his}}$ and future matrix $\mathbf{Y}_{\text{fut}}$. By projecting these matrices onto lower-dimensional subspaces, one can suppress the variance introduced by noise and partially recover the latent signal structure. However, direct manipulation of $\mathbf{Y}_{\text{his}}$ and $\mathbf{Y}_{\text{fut}}$ is neither necessary nor ideal. It can be shown that imposing a low-rank constraint on the model's weight matrix $\mathbf{W}$ achieves an equivalent effect (see Proposition 2 below). This constraint implicitly projects the input and output data onto learned low-dimensional subspaces during training, without the need to alter the raw sequences themselves. A low-rank $\mathbf{W}$ functions as a bottleneck, aligning $\mathbf{Y}_{\text{his}}$ and $\mathbf{Y}_{\text{fut}}$ along directions of maximal shared variation and filtering out noise-dominated components.

> **Proposition 2.** *Constraining $\mathbf{W}$ to be low-rank implicitly projects $\mathbf{Y}_{his}$ and $\mathbf{Y}_{fut}$ onto low-dimensional subspaces.*

See Appendix D.4 for a detailed version and its proof. See also Appendix D.6 for how this improves noise robustness. To operationalize this idea, we propose two practical strategies for incorporating rank constraints into linear models, with detailed algorithms provided in Algorithms 1 and 2.

**Reduced-Rank Regression (RRR).**    This approach (Izenman, 1975) explicitly enforces a low-rank constraint in the optimization target. It begins by computing the ordinary least squares (OLS) solution, then projects the forecast outputs onto a lower-dimensional subspace using truncated SVD. The weight matrix is re-estimated to best match this projected output. By restricting the model to operate within a smaller set of latent directions, RRR compresses the mapping from inputs to outputs and aligns both with a shared low-dimensional representation. This joint dimensionality reduction often improves generalization, particularly when the underlying data-generating process is itself low-rank.

| **Algorithm 1** Reduced-Rank Regression | **Algorithm 2** Direct Weight Rank Reduction |
|---|---|
| **Input:** $\mathbf{Y}_{\text{his}}, \mathbf{Y}_{\text{fut}}, \rho$; | **Input:** $\mathbf{Y}_{\text{his}}, \mathbf{Y}_{\text{fut}}, \rho$; |
| **Output:** $\mathbf{W}$; | **Output:** $\mathbf{W}$; |
| 1: compute the OLS solution: $\mathbf{W}_{\text{OLS}} = (\mathbf{Y}_{\text{his}}^{\top}\mathbf{Y}_{\text{his}})^{-1}\mathbf{Y}_{\text{his}}^{\top}\mathbf{Y}_{\text{fut}}$; | 1: compute the OLS solution: $\mathbf{W}_{\text{OLS}} = (\mathbf{Y}_{\text{his}}^{\top}\mathbf{Y}_{\text{his}})^{-1}\mathbf{Y}_{\text{his}}^{\top}\mathbf{Y}_{\text{fut}}$; |
| 2: calculate estimation : $\hat{\mathbf{Y}}_{\text{fut}} = \mathbf{Y}_{\text{his}}\mathbf{W}_{\text{OLS}}$ | 2: perform SVD on $\mathbf{W}_{\text{OLS}}$: $\mathbf{W}_{\text{OLS}} = \mathbf{U}\boldsymbol{\Sigma}\mathbf{V}^{\top}$; |
| 3: perform SVD on $\hat{\mathbf{Y}}_{\text{fut}}$: $\hat{\mathbf{Y}}_{\text{fut}} = \mathbf{U}\boldsymbol{\Sigma}\mathbf{V}^{\top}$; | |
| 4: truncate to rank $\rho$: $\mathbf{W}_{\text{RRR}} = \mathbf{W}_{\text{OLS}}\mathbf{V}_{\rho}\mathbf{V}_{\rho}^{\top}$; | 3: truncate to rank $\rho$: $\mathbf{W}_{\text{DWRR}} = \mathbf{U}_{\rho}\boldsymbol{\Sigma}_{\rho}\mathbf{V}_{\rho}^{\top}$; |
| 5: **return** $\mathbf{W}_{\text{RRR}}$; | 4: **return** $\mathbf{W}_{\text{DWRR}}$; |

**Direct Weight Rank Reduction (DWRR).** In contrast to RRR, DWRR applies rank reduction directly on the weight. The model is first trained without any constraints. Truncated SVD is then applied as a post-processing step to produce a low-rank approximation by discarding the smaller singular values.

See Appendix D.5 for additional theoretical analysis of Rank-Reduction methods.

## 4.2 ROOT PURGE FOR DYNAMIC RANK ADJUSTMENT

Previously, we described how post-training SVD-based root restructuring offers a simple way to enforce low-rank structure. However, this approach is rule-based and depends on fixed assumptions, such as accurate rank estimates, that may not hold in practice. To overcome these limitations, we propose a more flexible alternative: an adaptive, training-integrated method for root restructuring.

We introduce a modified training loss designed to enable online root restructuring by encouraging the model to learn a dynamically adjusted null space. Specifically, we define the following loss function:

$$\min_{\mathbf{W}} \underbrace{\|\mathbf{Y}_{\text{fut}} - \mathcal{G}_{\mathbf{W}}(\mathbf{Y}_{\text{his}})\|_F^2}_{\text{root-seeking}} + \lambda \underbrace{\|\mathcal{G}_{\mathbf{W}} \circ \mathcal{P}\left(\mathbf{Y}_{\text{fut}} - \mathcal{G}_{\mathbf{W}}(\mathbf{Y}_{\text{his}})\right)\|_F^2}_{\text{root-purging}} \qquad (3)$$

Here, $\mathcal{G}_{\mathbf{W}} : \mathbb{R}^{N \times L} \to \mathbb{R}^{N \times H}$ denotes a linear transformation governed by the learned weight matrix $\mathbf{W}$. The operator $\mathcal{P} : \mathbb{R}^{N \times H} \to \mathbb{R}^{N \times L}$ ensures dimensional consistency by either cropping or zero-padding the output[4]. The loss consists of two complementary components: (1) The root-seeking term reflects the standard prediction loss commonly used in time series forecasting; (2) The root-purging term serves as a regularizer, encouraging the model to learn a null space that suppresses noise.

**Intuition Behind Root Purge.** The root-purging term is grounded in a straightforward yet powerful principle: when a model encounters pure noise which is assumed to have zero mean, its output should ideally be zero. This term operates in two stages. First, it estimates the noise by computing the residual between the model's prediction and the ground truth. Second, it re-applies the model to this residual to assess whether it lies in the null space of the transformation. If the residual indeed represents noise and the model has correctly captured its null space, the resulting output should be close to zero. Although the purging term is unlikely to vanish completely—since noise typically spans a full-rank space—its minimization encourages the model to distinguish signal from noise. In doing so, it promotes the discovery of a low-rank structure that aligns with the underlying temporal dynamics, enhancing robustness to stochastic perturbations.

**Optimization in Time vs. Frequency Domains.** The proposed loss formulation is agnostic to the domain in which the optimization is performed, as long as the transformation applied by the model remains linear. Specifically, the function $\mathcal{G}_{\mathbf{W}}$ used in both the root-seeking and root-purging terms can be defined differently depending on the chosen representation:

---

[3]In practice: if $H < L$, we zero-pad the output to length $L$ and scale $\lambda$ by $L/H$; if $H \geq L$, we crop to the first $L$ columns and keep $\lambda$ unchanged.

[4]For more memory-efficient training, we apply a stop-gradient operation to $\mathcal{P}(\cdot)$. Empirically, this has minimal impact on the performance of Root Purge. See Appendix E.14 for further details.

- **Time domain:** $\mathcal{G}_{\mathbf{W}}$ is the linear transformation defined by the weight matrix $\mathbf{W}_T$.

- **Frequency domain:** $\mathcal{G}_{\mathbf{W}}$ is defined as $\mathcal{F}^{-1} \circ \mathbf{W}_F \circ \mathcal{F}$, where $\mathcal{F}$ and $\mathcal{F}^{-1}$ denote the forward and inverse discrete Fourier transforms, respectively.

The second formulation effectively applies a linear filter in the frequency domain, making it especially well-suited for signals with pronounced periodic or oscillatory patterns. By enabling optimization in either the time or the frequency domain, our framework offers practitioners the flexibility to align model design with the inherent structure of their data, enhancing both computational efficiency and interpretability, without compromising the theoretical guarantees of our approach.

**Relationship between Root Purge and Rank Reduction.** The root-purging mechanism is fundamentally connected to rank reduction through the rank-nullity theorem (Axler, 2015), which states that increasing the dimension of a linear transformation's null space necessarily reduces its rank. This relationship allows the model to dynamically adjust the rank of its transformation matrix during training. The optimization balances two opposing forces: if the rank is too low to capture the underlying dynamics, the prediction (root-seeking) loss increases, encouraging the model to raise its rank; if the rank is high enough to fit the signal, the root-purging term becomes more influential, driving the model to expand its null space—thus reducing rank and suppressing noise. Through this adaptive process, the model self-regulates its capacity, learning to extract the underlying temporal structure while remaining robust to stochastic perturbations.

To see additional theoretical analysis of Root Purge, please refer to Appendix D.7.

## 5 EXPERIMENTS

### 5.1 EXPERIMENTAL SETUP

**Datasets.** We evaluate our methods on several widely used real-world datasets for long-term time series forecasting, including Traffic, Electricity, Weather, Exchange, and ETT (Zhou et al., 2021). Due to our focus on data scaling properties, we present results from smaller-scale datasets like ETT, Exchange, and Weather in the main text, with a full set of results provided in the Appendix E.

**Baselines.** Despite our theoretical analysis being primarily grounded in linear models, we include baselines from a broad range of state-of-the-art time series forecasting methods. Specifically, we evaluate against three major categories: (1) transformer-based models: FEDformer (Zhou et al., 2022) and PatchTST (Nie et al., 2022); (2) convolution-based models: TimesNet (Wu et al., 2023), TSLANet (Eldele et al., 2024), and Plain FilterNet (Yi et al., 2024); and (3) linear models: FITS (Xu et al., 2023), SparseTSF (Lin et al., 2024), and DLinear (Zeng et al., 2023).

### 5.2 MAIN RESULTS

We evaluate our structural regularization approaches for linear models. Table 1 shows the results with rank reduction using RRR and Root Purge in the frequency domain, and results on time domain models are given in Appendix E.7 and E.8. Experiments are repeated five times, and we report the average performance. The confidence interval for Root Purge is omitted, as run-to-run differences are negligible except for the Exchange dataset (within $\pm 0.002$). Both RRR and Root Purge consistently outperform other baselines. RRR surpasses methods that require extensive hyperparameter tuning, providing simpler rank adjustment without retraining. Meanwhile, Root Purge pushes the performance limits of linear time series forecasting across multiple benchmarks while maintaining simplicity and computational efficiency. These methods are especially effective on smaller datasets, where models relying solely on data scaling tend to underperform, in line with our theoretical analysis.

### 5.3 THEORETICAL PROPERTIES VERIFICATION

In this section, we validate the theoretical properties of our proposed methods in three stages: from singular value shrinkage (the mathematical mechanism) to improved data scaling (the key practical challenge) and finally back to characteristic root restructuring (the foundational motivation). We

Table 1: Forecasting result for horizon $H \in \{96, 192, 336, 720\}$ with lookback window of length $L = 720$. For RRR, we tune the rank on the validation set, select the top three with lowest validation MSE, and report the best test result. For Root Purge, we select the best test MSE over $\lambda \in [0.125, 0.25, 0.5]$. The best results are highlighted in **red**, and second-best in **blue**.

| Dataset | $H$ | FEDformer | FilterNet | TSLANet | TimesNet | PatchTST | DLinear | SparseTSF | FITS | **RRR** | **Root Purge** |
|---|---|---|---|---|---|---|---|---|---|---|---|
| ETTh1 | 96 | 0.375 | 0.386 | 0.387 | 0.384 | 0.385 | 0.384 | **0.362** | 0.379 | 0.367 | **0.359** |
| | 192 | 0.427 | 0.420 | 0.421 | 0.436 | 0.413 | 0.443 | 0.403 | 0.414 | **0.401** | **0.394** |
| | 336 | 0.459 | 0.449 | 0.468 | 0.491 | 0.440 | 0.446 | 0.434 | 0.435 | **0.430** | **0.423** |
| | 720 | 0.484 | 0.500 | 0.529 | 0.521 | 0.456 | 0.504 | 0.426 | 0.431 | **0.425** | **0.421** |
| ETTh2 | 96 | 0.340 | 0.309 | 0.299 | 0.340 | 0.274 | 0.282 | 0.294 | 0.272 | **0.268** | **0.268** |
| | 192 | 0.433 | 0.376 | 0.369 | 0.402 | 0.338 | 0.350 | 0.339 | 0.331 | **0.329** | **0.328** |
| | 336 | 0.508 | 0.418 | 0.390 | 0.452 | 0.367 | 0.414 | 0.359 | **0.354** | **0.352** | 0.355 |
| | 720 | 0.480 | 0.484 | 0.444 | 0.462 | 0.391 | 0.588 | 0.383 | 0.379 | **0.376** | **0.377** |
| ETTm1 | 96 | 0.362 | 0.315 | 0.307 | 0.338 | **0.292** | **0.301** | 0.314 | 0.310 | 0.306 | 0.305 |
| | 192 | 0.393 | 0.364 | 0.349 | 0.374 | **0.330** | 0.335 | 0.343 | 0.338 | 0.336 | **0.333** |
| | 336 | 0.442 | 0.391 | 0.384 | 0.410 | 0.365 | 0.371 | 0.369 | 0.366 | **0.365** | **0.360** |
| | 720 | 0.483 | 0.457 | 0.471 | 0.478 | 0.419 | 0.426 | 0.418 | 0.415 | **0.414** | **0.412** |
| ETTm2 | 96 | 0.189 | 0.180 | 0.197 | 0.187 | 0.163 | 0.171 | 0.165 | 0.163 | **0.161** | **0.161** |
| | 192 | 0.256 | 0.236 | 0.251 | 0.249 | 0.219 | 0.237 | 0.218 | 0.217 | **0.216** | **0.216** |
| | 336 | 0.326 | 0.292 | 0.303 | 0.321 | 0.276 | 0.294 | 0.272 | 0.269 | **0.268** | **0.269** |
| | 720 | 0.437 | 0.366 | 0.378 | 0.408 | 0.368 | 0.426 | 0.350 | 0.350 | **0.348** | **0.350** |
| Weather | 96 | 0.246 | 0.149 | 0.171 | 0.172 | 0.151 | 0.174 | 0.172 | 0.144 | **0.140** | **0.142** |
| | 192 | 0.292 | 0.194 | 0.219 | 0.219 | 0.195 | 0.217 | 0.215 | 0.188 | **0.182** | **0.186** |
| | 336 | 0.378 | 0.244 | 0.267 | 0.280 | 0.249 | 0.262 | 0.260 | 0.239 | **0.232** | **0.238** |
| | 720 | 0.447 | 0.316 | 0.332 | 0.365 | 0.321 | 0.332 | 0.318 | **0.309** | **0.304** | 0.310 |
| Exchange | 96 | 0.148 | 0.110 | 0.130 | 0.107 | 0.110 | 0.088 | 0.090 | 0.086 | **0.084** | **0.082** |
| | 192 | 0.271 | 0.230 | 0.243 | 0.226 | 0.284 | 0.182 | 0.182 | 0.177 | **0.174** | **0.172** |
| | 336 | 0.460 | 0.384 | 0.484 | 0.367 | 0.448 | 0.330 | 0.330 | 0.331 | **0.324** | **0.324** |
| | 720 | 1.195 | 1.062 | 1.079 | 0.964 | 1.092 | 1.060 | 1.051 | **0.936** | **0.915** | 0.941 |
| Number of $1^{\text{st}}$ Places | | 0 | 0 | 0 | 0 | 2 | 0 | 0 | 0 | 9 | 13 |

then further assess their sensitivity to hyperparameters and the possibility of alternative architectural choices in the next section.

**Effect on Singular Value Spectrum.** While RRR performs explicit rank reduction on the weight matrix, the Root Purge performs rank reduction implicitly via the rank-nullity tradeoff. In this section, we perform a case study on ETTh1 and ETTm1 to show how the singular values of $\mathbf{W}$ change after training via Root Purge. Since our model is trained in the frequency domain, we revert the transformed model back to the temporal domain for singular value decomposition (SVD). By leveraging the properties of linear operators, we input the identity matrix to obtain the corresponding weight matrix in the time domain. As shown in Figure 2, the degree of Root Purge does not affect the overall trend of the singular value distribution across both datasets. Increasing the regularization coefficient $\lambda$ leads to a more pronounced shrinkage of singular values. Notably, significant singular values remain unchanged even with a large regularization coefficient. This underscores the role of the root-seeking loss, which prevents the noise-free signal subspaces to be suppressed.

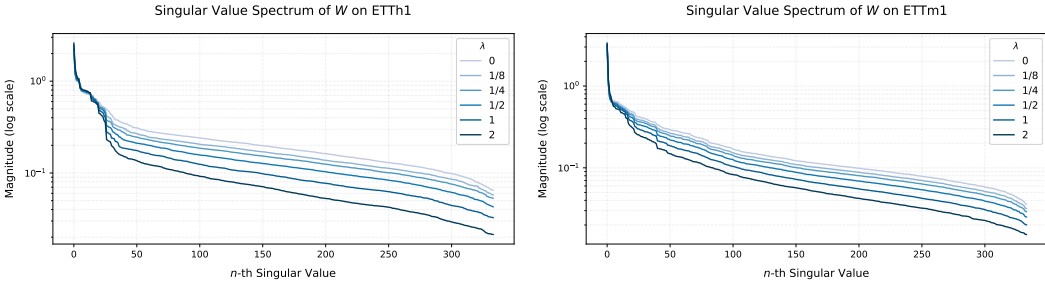

Figure 2: First 336 singular value magnitudes on ETTh1 and ETTm1 under different values of $\lambda$ (log scale). As $\lambda$ increases, Root Purge pushes the weight matrix $\mathbf{W}$ to have more smaller singular values, while the significant singular values remain largely unaffected.

**Data Scaling & Noise Robustness Property.** We further use a toy example to illustrate the scaling behavior of different methods under noisy conditions. Specifically, we examine the effects of two key factors: training data volume and noise magnitude. We construct a synthetic signal $\{y_t\}_{t \in [0,T]}$ with both trend and periodic components: $y_t = \sin(2t) + \cos(5t) + 0.5t + \sigma \cdot \varepsilon_t$, where $\varepsilon_t \sim \mathcal{N}(0, 1)$

and observations are taken at increments of $0.01$ in $t$. To examine data scaling behavior, we fix the noise level at $\sigma = 0.5$ and vary the size of the training dataset. As shown in Figure 3 (left), baseline models struggle under limited data and require substantially more samples to mitigate the impact of noise, revealing a clear data scaling inefficiency. In contrast, our proposed methods, RRR and Root Purge, maintain consistently strong performance across all dataset sizes, with minimal performance drop as data size decreases. To assess noise robustness, we fix the training interval to $t \in [0, 200]$ and vary the noise level $\sigma$. Results in Figure 3 (right) show that while baseline models degrade more significantly with increasing noise, RRR and Root Purge remain stable and robust throughout.

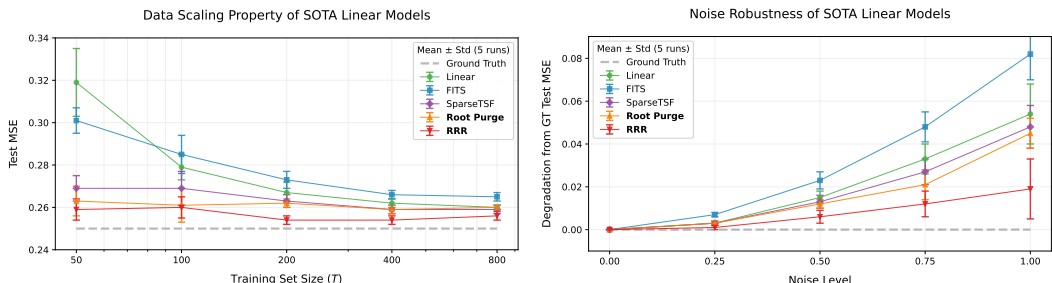

Figure 3: Data scaling and noise robustness of state-of-the-art linear time-series models. (left) RRR and Root Purge exhibit near-constant performance in data-scaling benchmarks. (right) Both methods exhibit robust performance under increasing noise levels, outperforming baseline models.

Table 2: Comparison of Root Distance to Ground Truth across models (root pairings determined by Hungarian algorithm (Kuhn, 1955)). The results demonstrate the effectiveness of Rank Reduction and Root Purge in recovering roots closer to the true values compared to standard linear models.

| Model | Root Distance to Ground Truth (mean $\pm$ std) |
|---|---|
| RRR | $\mathbf{0.036 \pm 0.014}$ |
| Root Purge | $\mathbf{0.045 \pm 0.009}$ |
| Standard Linear Model (OLS) | $0.064 \pm 0.025$ |

**Analysis of Characteristic Roots.** Finally, to close the loop of on our framework, we hereby directly evaluate how Rank Reduction Regression (RRR) and Root Purge affect the learned root distribution in a controlled synthetic setting. Such a direct comparison is usually infeasible on real-world data because the underlying noise-free dynamics and their true characteristic roots are unknown. Therefore, we conduct experiments using the aforementioned synthetic dynamics: $y_t = \sin(2t) + \cos(5t) + 0.5t + \sigma \cdot \varepsilon_t$. For each method, we compute the mean and standard deviation of the distances between all learned roots and the ground truth roots, with root pairs matched using the Hungarian algorithm (Kuhn, 1955). As shown in Table 2, the roots estimated by OLS tend to drift farther from the ground-truth characteristic roots, whereas those estimated by Root Purge and RRR remain closer. This observation confirms that our proposed techniques indeed help prevent the learning of spurious roots, thereby leading to the performance improvements demonstrated earlier. Additional experimental details and qualitative evaluations are also provided in Appendix E.13.

## 5.4 PRACTICAL ROBUSTNESS ANALYSIS

**Hyperparameter Sensitivity.** In our Root Purge method, as in Equation (3), the only tunable hyperparameter is $\lambda$, which controls the balance between prediction error and regularizer for noise suppression. This section investigates the model's sensitivity to this hyperparameter on the ETTh1 and ETTm1 datasets. We train a linear forecasting model with Root Purge using: $\lambda \in [1/32, 1/16, 1/8, 1/4, 1/2, 1, 2]$, and report the average performance across all forecasting horizons. Figure 4 shows that the performance improves initially with increasing $\lambda$, reaches an optimum, and then deteriorates when $\lambda$ becomes too large. Notably, even very small values of $\lambda$ enhance model performance. This positive effect remains consistent across a broad range of $\lambda$ values, highlighting the robustness of our method.

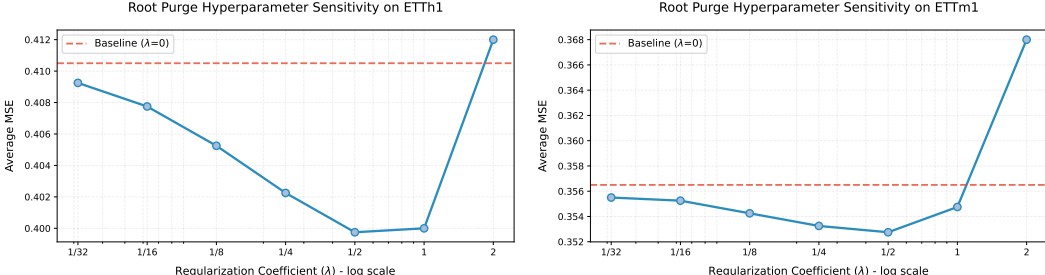

Figure 4: Average forecasting MSE on ETTh1 and ETTm1 across horizons $H = \{96, 192, 336, 720\}$ for different values of $\lambda$. Results indicate that a wide range of $\lambda$ improves predictions, whereas larger values may cause over-regularization. A break-down table for each horizon is in Appendix E.7.

**Channel Independent vs. Individual Channel Modeling.** The results presented so far are mainly based on the *channel independent* (CI) strategy, which models all channels using a shared weight matrix. A key advantage of CI is its ability to aggregate noise from all channels. This serves as implicit data augmentation and, by Proposition 1, enhances model robustness. However, the expressivity of CI is inherently limited by its shared parameterization, creating a tradeoff between robustness and modeling capacity. Here, we consider an alternative design: *individual channel* (INC) modeling, where each channel is assigned its own linear model (see Table 6 for more details of CI and INC). This increases model capacity and allows the system to better adapt to channel-specific dynamics. However, without the benefit of cross-channel noise sharing, INC models are more sensitive to noise and often underperform in raw form, as shown in Table 3. Notably, when combined with our proposed Root Purge strategy, the INC formulation becomes significantly more robust. Root Purge enables each individual model to effectively suppress channel-specific noise and isolate meaningful dynamics. In several cases, this leads to substantial performance gains over CI models. For example, on the ETTm1 dataset, the INC+Root Purge model even outperforms the previous non-linear state-of-the-art PatchTST (Nie et al., 2022) on the 96- and 192-step forecasting tasks. These results highlight how our method unlocks alternative, more expressive architectures that would otherwise be infeasible.

Table 3: Performance of INC linear models with Root Purge. While standalone INC models typically underperform due to limited data, applying Root Purge significantly mitigates this issue.

| Dataset | ETTh1 | | | | ETTh2 | | | | ETTm1 | | | | ETTm2 | | | |
|---|---|---|---|---|---|---|---|---|---|---|---|---|---|---|---|---|
| Horizon | 96 | 192 | 336 | 720 | 96 | 192 | 336 | 720 | 96 | 192 | 336 | 720 | 96 | 192 | 336 | 720 |
| (CI) Linear | 0.375 | 0.411 | 0.439 | 0.431 | 0.270 | 0.331 | 0.354 | 0.378 | 0.306 | 0.336 | 0.365 | 0.414 | 0.162 | 0.217 | 0.269 | 0.350 |
| (CI) Root Purge | 0.359 | 0.394 | **0.423** | **0.423** | **0.269** | 0.329 | 0.355 | 0.377 | 0.305 | 0.333 | 0.360 | **0.413** | 0.162 | 0.217 | **0.269** | **0.350** |
| INC Linear | 0.397 | 0.432 | 0.450 | 0.453 | 0.290 | 0.338 | 0.364 | 0.383 | 0.297 | 0.337 | 0.370 | 0.421 | 0.165 | 0.222 | 0.276 | 0.356 |
| INC Root Purge | **0.357** | **0.394** | 0.427 | 0.438 | 0.271 | **0.322** | **0.353** | **0.376** | **0.292** | **0.329** | **0.360** | 0.418 | **0.161** | **0.217** | 0.273 | 0.356 |

## 6 CONCLUSIONS

This paper presents a comprehensive study of linear models for time series forecasting, focusing on the role of *characteristic roots* in shaping model expressivity. We propose two complementary methods, Rank Reduction and Root Purge, for improved forecasting through root identification. Experiments validate our theoretical claims and demonstrate the effectiveness of both methods across a range of forecasting tasks. More detailed discussions and limitations are provided in Appendix F.

## REPRODUCIBILITY STATEMENT

Detailed descriptions of the metric and experimental setup, dataset we used, and baseline information are provided in Section 5, Appendix E.1, E.2, and E.3. Detailed descriptions, assumptions, and proofs of theories (Facts, Propositions, Remarks, Claims, and Corollaries) can be found in Appendix D. Source code is available in the supplementary material as a zip file. You can also find our source code at https://github.com/Wangzzzzzzzz/RootPurge.

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

CONTENTS

# A MIND MAP, NOTATIONS, AND SUMMARY OF THEOREMS

## A.1 MIND MAP

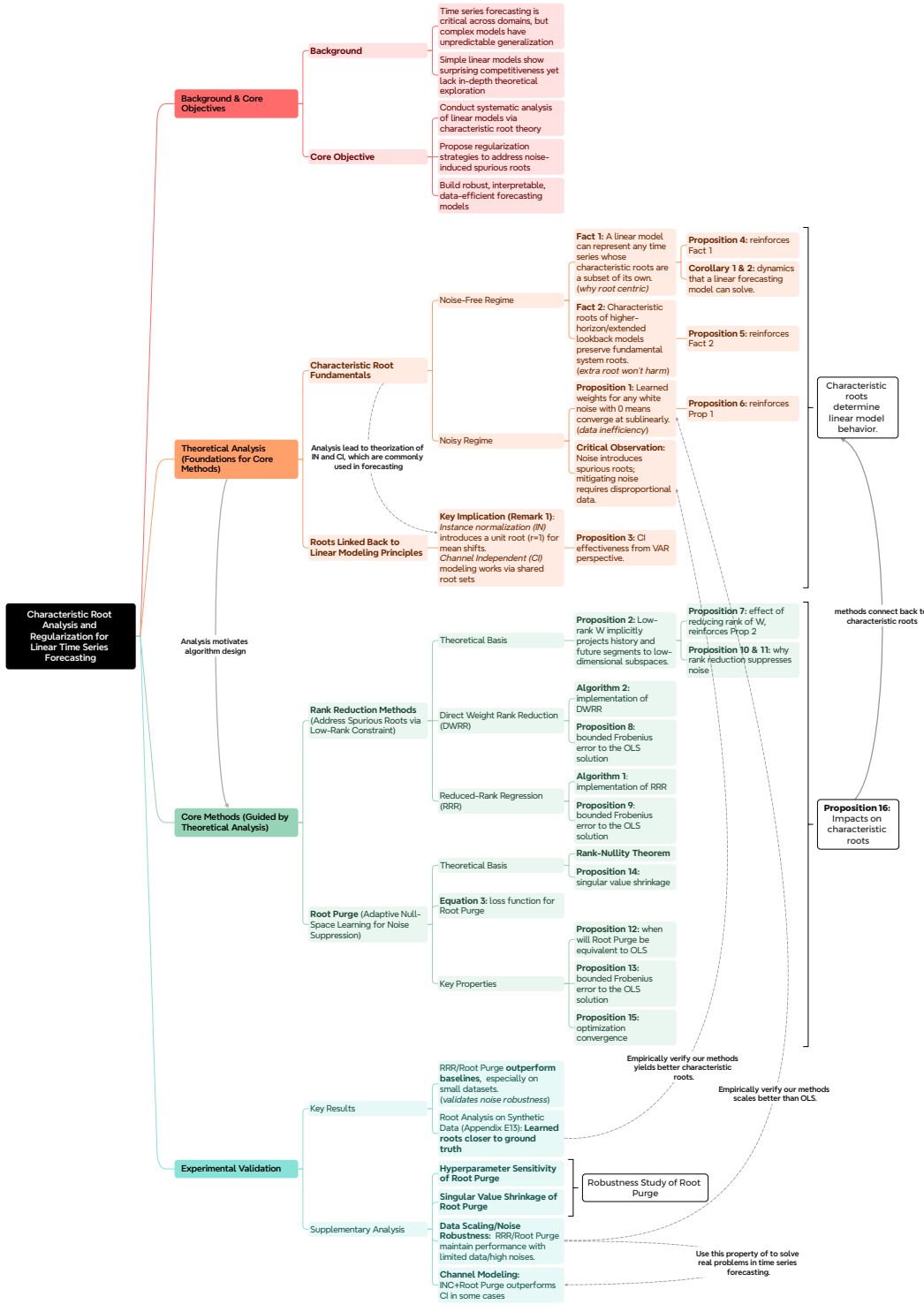

Figure 5: A mind map of the paper describing its logic flow and how we organize this paper.

This mind map summarizes the logical flow of the study, showing how the paper moves from theory to modeling principles and then to algorithm design. It begins with generalization foundations derived from the characteristic roots of linear recurrence relations, which guide the analysis of modeling principles (including lookback window, prediction horizon, instance normalization, channel independence, and data scaling laws) under both noise-free and noisy conditions. These theoretical insights motivate the development of two complementary algorithmic strategies—rank reduction (using reduced rank regression or direct weight rank reduction) and the proposed root purge method—which in turn connect back to the characteristic roots, forming a closed loop between theoretical analysis and practical algorithm design. Figure 5 thus provides a roadmap for understanding how the paper integrates classical linear systems theory with modern learning techniques to build robust, interpretable, and data-efficient time series forecasting models.

## A.2 NOTATION

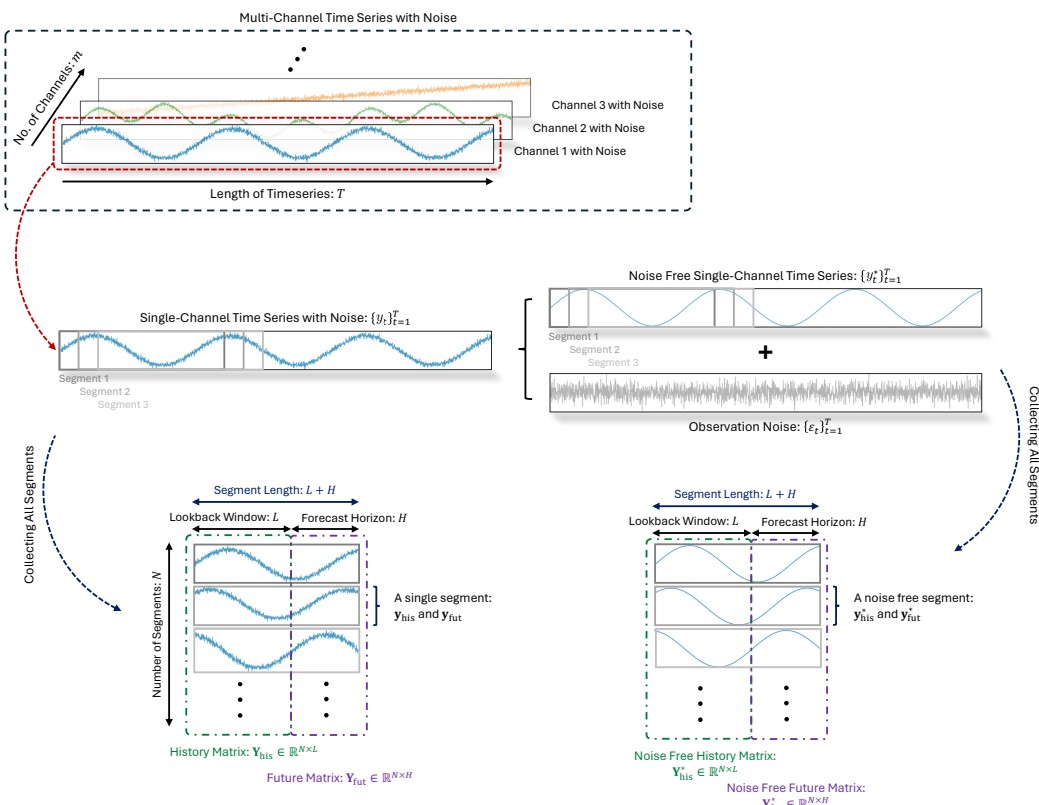

Figure 6: Mind map for common notations we used in our paper for a time series dataset. A more detailed tabular description can be found in Table 4.

**Remark 2** (On the Construction of $\mathbf{Y}_{his}$ and $\mathbf{Y}_{fut}$). *In our framework, $\mathbf{Y}_{his}$ and $\mathbf{Y}_{fut}$ are constructed such that each row corresponds to a univariate segment drawn from a shared underlying dynamical system. Specifically,*

$$\mathbf{Y}_{his} = \begin{bmatrix} y_{1,1} & y_{1,2} & \cdots & y_{1,L} \\ y_{2,1} & y_{2,2} & \cdots & y_{2,L} \\ \vdots & \vdots & \ddots & \vdots \\ y_{N,1} & y_{N,2} & \cdots & y_{N,L} \end{bmatrix}, \quad \mathbf{Y}_{fut} = \begin{bmatrix} y_{1,L+1} & y_{1,L+2} & \cdots & y_{1,L+H} \\ y_{2,L+1} & y_{2,L+2} & \cdots & y_{2,L+H} \\ \vdots & \vdots & \ddots & \vdots \\ y_{N,L+1} & y_{N,L+2} & \cdots & y_{N,L+H} \end{bmatrix}.$$

*In this setting, the future segment in each row of $\mathbf{Y}_{fut}$ immediately follows the corresponding history segment in $\mathbf{Y}_{his}$ along the time axis. This construction is especially suitable for multi-channel time series where independent samples (e.g., across entities or sensors) share the same temporal dynamics.*

Table 4: Summary of Notation

| Symbol | Description |
|---|---|
| $m$ | Number of channels (multivariate time series dimension) |
| $T$ | Total length of time series observations |
| $K$ | Order of the underlying recurrence/dynamic system |
| $L$ | Lookback window length (history segment length) |
| $H$ | Forecasting horizon (future segment length) |
| $N$ | Number of segments in the dataset |
| $\{C_i\}_{i=1}^p$ | Undetermined coefficients in the general solution |
| $\{r_i\}_{i=1}^p$ | Characteristic roots of the recurrence relation |
| $\{a_i\}_{i=1}^p$ | Coefficients of the recurrence relation |
| $\lambda$ | Regularization coefficient for root purging |
| $\sigma$ | Noise level (standard deviation) in observations |
| $\rho$ | Selected rank for low-rank approximation |
| $\mathcal{P}$ | Dimension alignment operator |
| $\mathcal{G}_{\mathbf{W}}$ | Linear operator parameterized by weight matrix $\mathbf{W}$ |
| $\mathcal{F}, \mathcal{F}^{-1}$ | Fourier transform and its inverse |
| $\mathbf{Y}_{\text{his}}^*$ | Collection of noise-free history segments |
| $\mathbf{Y}_{\text{fut}}^*$ | Collection of noise-free future segments |
| $\mathbf{Y}_{\text{his}}$ | Collection of noisy history segments |
| $\mathbf{Y}_{\text{fut}}$ | Collection of noisy future segments |
| $\mathbf{y}_{\text{his}}^*, \mathbf{y}_{\text{fut}}^*$ | Single noise-free history and future segment |
| $\mathbf{y}_{\text{his}}, \mathbf{y}_{\text{fut}}$ | Single noisy history and future segment |
| $\mathbf{y}_t, y_t$ | Time series observation at step $t$ (multi- and single-channel) |
| $\hat{\mathbf{Y}}_{\text{fut}}$ | Collection of all predicted future segments |
| $\mathbf{W}$ | Weight matrix of linear model; $W_{jk}$ denotes element at $(j, k)$ |
| $\mathbf{W}_{\text{OLS}}$ | OLS solution for weight matrix |
| $\mathbf{W}_{\text{RRR}}$ | Reduced-Rank Regression (RRR) solution |
| $\mathbf{W}_{\text{DWRR}}$ | Direct Weight Rank Reduction (DWRR) solution |
| $\mathbf{W}_T$ | Weight matrix in the time domain for root purging |
| $\mathbf{W}_F$ | Weight matrix in the frequency domain for root purging |
| $\mathbf{U}, \mathbf{\Sigma}, \mathbf{V}^\top$ | SVD of a matrix |
| $\mathbf{U}_\rho, \mathbf{\Sigma}_\rho, \mathbf{V}_\rho^\top$ | Rank-$\rho$ truncated SVD components |
| $\varepsilon_{\text{his}}, \varepsilon_{\text{fut}}$ | Noise on history and future segments |

*Alternatively, in traditional time series analysis, a more common approach is **Hankelization**. Given a sufficiently long univariate time series $\{y_t\}_{t=1}^T$, the data is organized by applying a sliding window over the temporal axis to extract overlapping segments. In this case, $N = T - H - L + 1$, and*

$$\mathbf{Y}_{his} = \begin{bmatrix} y_1 & y_2 & \cdots & y_L \\ y_2 & y_3 & \cdots & y_{L+1} \\ \vdots & \vdots & \ddots & \vdots \\ y_N & y_{N+1} & \cdots & y_{N+L-1} \end{bmatrix}, \quad \mathbf{Y}_{fut} = \begin{bmatrix} y_{L+1} & y_{L+2} & \cdots & y_{L+H} \\ y_{L+2} & y_{L+3} & \cdots & y_{L+H+1} \\ \vdots & \vdots & \ddots & \vdots \\ y_{L+N} & y_{L+N+1} & \cdots & y_{L+N+H-1} \end{bmatrix}.$$

*This formulation is widely adopted in signal processing and control theory. Under this view, each row represents a segment from a single long sequence, with consecutive rows overlapping temporally. Both formulations are compatible with our proposed framework, and the choice between them can be made depending on the application scenario and data availability.*

## A.3 SUMMARY OF THEOREMS

In our paper, we had proposed multiple theorems, facts, propositions, and corollaries about linear forecasting models and our proposed methods. To provide a comprehensive idea of how each of them contributes to our analysis and what they mean in an intuitive sense, we summarize them in Table 5 shown below. For the detailed assumption and proofs, please refer to Appendix D.

Table 5: Summary of Key Theorems, Facts, Propositions, and Corollaries

| Theorem | Plain-Language Explanation | Role and Significance in the Paper |
|---|---|---|
| Fact 1 | A linear model can accurately represent a time series if the time series' characteristic roots are a subset of the model's characteristic roots. | 1. Defines core generalization logic: adaptation depends on root subset relationships. 2. Provides theoretical basis for fitting diverse time series. |
| Fact 2 | For linear models with longer forecasting horizons or extended lookback windows, their set of characteristic roots always includes the key roots of the system. | 1. Justifies independent modeling per forecast horizon. 2. Shows parameter redundancy enhances flexibility without harming core dynamics. |
| Proposition 1 (Informal version of Prop. 6) | For a zero-mean noise with finite second moments, convergence rate of OLS weights is $O(1/\sqrt{T})$. | 1. Reveals limitations of least-squares in noisy data. 2. Motivates structural regularization (RRR, Root Purge). |
| Proposition 2 (Informal version of Prop. 7) | Constraining weight matrix $W$ to low-rank is equivalent to projecting input and output into a shared low-dimensional subspace. | 1. Basis for RRR/DWRR methods. 2. Explains denoising via low-rank constraints. |
| Proposition 3 | Multi-channel time series with diagonal recurrence per channel can be transformed to a shared-parameter recurrence. | 1. Proves effectiveness of channel-independent modeling. 2. Shared weights improve robustness and reduce parameters. |
| Proposition 4 | Two exponential combinations with different characteristic roots that match at $L$ points must match everywhere. | 1. Strengthens Fact 1: root subset relation determines expressivity. 2. Provides generalization guarantee. |
| Proposition 5 | Characteristic polynomial of minimum-order recurrence divides higher-order recurrences. | 1. Deepens Fact 2 from polynomial theory. 2. High-order models only add redundant roots. |
| Proposition 6 (Formal version of Prop. 1) | For a zero-mean noise with finite second moments, convergence rate of OLS weights is $O(1/\sqrt{T})$. | 1. Reveals limitations of least-squares in noisy data. 2. Motivates structural regularization (RRR, Root Purge). |
| Proposition 7 (Formal version of Prop. 2) | Constraining weight matrix $W$ to low-rank is equivalent to projecting input and output into a shared low-dimensional subspace. | 1. Basis for RRR/DWRR methods. 2. Explains denoising via low-rank constraints. |
| Proposition 8 | DWRR's low-rank approximation of OLS is optimal (Frobenius norm), error = sum of discarded singular values squared. | 1. Proves DWRR optimality. 2. Guides engineering implementation. |
| Proposition 9 | RRR error is bounded by minimal singular values of input and output data. | 1. Quantifies RRR approximation accuracy. 2. Guides rank selection. |
| Proposition 10 | If true matrix is low-rank, truncated SVD error after noise is controlled by noise spectral norm. | 1. Provides matrix denoising interpretation. 2. Low-rank constraints robustly isolate noise. |
| Proposition 11 | Error of estimating true principal subspace decreases with sample size. | 1. Ensures statistical reliability of rank reduction. 2. Supports tuning rank $\rho$ with validation data. |
| Proposition 12 | OLS is a critical point of Root Purge objective iff residual and weights satisfy null-space condition. | 1. Shows relationship between Root Purge and OLS. 2. Guides regularization for noise suppression. |
| Proposition 13 | Deviation between Root Purge and OLS weights is bounded by minimal singular values and regularization $\lambda$. | 1. Quantifies correction magnitude. 2. Ensures stability of Root Purge. |

*Continued on next page*

Table 5 – *Continued from previous page*

| Theorem | Plain-Language Explanation | Role and Significance in the Paper |
| --- | --- | --- |
| Proposition 14 | Each singular value of Root Purge weights $\leq$ corresponding OLS weight singular value. | 1. Shows Root Purge implicitly enforces low-rankness. 2. Connects Root Purge to rank reduction. |
| Proposition 15 | Gradient descent with proper step size converges to a stationary point of Root Purge objective. | 1. Ensures feasibility of Root Purge optimization. 2. Justifies using OLS warm starts. |
| Proposition 16 | If weight matrix approximation error = $\varepsilon$, root perturbation = $O(\varepsilon)$. | 1. Connects weight approximation to root stability. 2. Ensures regularized model preserves core dynamics. |
| Corollary 1 | Sinusoids, polynomial, exponential, and combinations can be represented by linear difference equations. | 1. Shows linear models fit common temporal patterns. 2. Supports long-term forecasting. |
| Corollary 2 | A linear model represents a time series iff the time series' PV extension is a subextension of the model's PV extension. | 1. Uses differential Galois theory to characterize model expressivity. 2. Shows linear models' generality for complex dynamics. |

## B  BACKGROUND

### B.1  RELATED WORKS

**Time Series Forecasting.**  Time series forecasting (TSF) involves predicting future values by learning patterns such as trends and seasonality from historical data. Over time, it has given rise to a variety of modeling paradigms, each addressing the challenges of long-term forecasting from different perspectives (Kong et al., 2025). Transformer-based architectures have been a dominant line of research. For example, Nie et al. (2022) proposes segmenting time series into subseries-level patches, treating each patch as an input token while modeling channels independently. Similarly, Zhou et al. (2022) combines transformer networks with seasonal-trend decomposition, capturing broad temporal profiles via decomposition while leveraging transformers for detailed pattern learning. Meanwhile, Woo et al. (2023) introduces deep time-index models, explicitly conditioning predictions on future time indexes to better handle temporal variations and support both interpolation and extrapolation tasks. Beyond time-domain approaches, alternative methods explore frequency-domain representations. Xu et al. (2023) applies a low-pass filter followed by complex-valued linear transformations to strengthen long-term dependencies and smooth high-frequency noise. In parallel, model efficiency and minimalism have gained attention. Lin et al. (2024) demonstrates that strong long-term forecasting performance can be achieved with as few as 1,000 parameters through structured sparsity. Zeng et al. (2023) further questions the necessity of complex architectures, showing that simple linear models—by separately modeling trend and seasonal components—can outperform transformer-based models on long-term forecasting benchmarks. Finally, comprehensive evaluations (Brigato et al., 2025) suggest that no single model consistently outperforms others across all scenarios. This highlights the importance of diverse evaluation settings and context-aware model selection, given the heterogeneous nature of real-world time series data. Motivated by these developments, we aim to approach long-term time series forecasting from a theoretical perspective, seeking models that balance complexity, interpretability, and robustness across diverse temporal structures.

**Linear Prediction Theory.**  Linear prediction theory, rooted in the early 20th century, emerged from the study of time series analysis and signal processing, with foundational contributions from mathematicians like Norbert Wiener and Andrey Kolmogorov in the 1940s (Vaidyanathan, 2007). Wiener's work on optimal filtering (Wiener, 1942) and Kolmogorov's theory on extrapolation (Kolmogoroff, 1931) laid the theoretical groundwork, formalizing the idea that future values of a discrete-time signal can be estimated as a linear combination of past observations. The theory gained prominence in the 1960s and 1970s with the development of efficient algorithms, such as the Levinson-Durbin recursion (Durbin, 1960), which solved the Yule-Walker equations for autoregressive (AR) modeling (Burg, 1968). The connection between time series forecasting and linear prediction is fundamental, as both

disciplines aim to capture and exploit temporal dependencies in sequential data (Pourahmadi, 2001). Linear prediction provides a mathematically tractable framework for time series analysis by expressing future values as weighted linear combinations of past observations - an approach particularly well-suited for stationary processes (Hamilton, 2020). Starting with DLinear (Zeng et al., 2023), which extends the linear framework by separately modeling trend and seasonal components, recent studies (Li et al., 2023; Toner and Darlow, 2024) have further explored the properties of linear models, providing practical insights for improving forecasting accuracy. Additionally, the concept of scaling laws, as explored in Shi et al. (2024), demonstrates how performance improves with larger data sizes and model capacities, highlighting both the potential and limitations of scaling in linear forecasting models. Building on these developments, we aim to enhance linear prediction models to better handle the complexities of real-world time series data.

**Spectral Methods, Low Rank Structure and Koopman Theory.** Spectral methods, which are algorithms built on eigenvalues and eigenvectors or singular values and singular vectors of data dependent matrices, have become a widely used toolkit for extracting structure from large, noisy, and incomplete data (Kannan et al., 2009; Chen et al., 2021). They have been successfully applied in machine learning, signal processing, imaging science, econometrics and many other areas, both as stand alone estimators and as initializations for more sophisticated procedures (Abbe, 2018; Keshavan et al., 2010; Chen et al., 2019). In the time series domain, a particularly important instance of spectral methodology is singular spectrum analysis (SSA) (Hassani, 2007; Zhigljavsky, 2010). SSA forms a Hankel or trajectory matrix from lagged copies of the series, performs an SVD based decomposition, and reconstructs interpretable components such as trend, periodicities and noise for denoising, gap filling and forecasting.

Low-rank structure is a pervasive modeling assumption for high-dimensional dynamical systems Agarwal et al. (2018); Liu (2022). By positing an effective low-rank interaction matrix, one can capture large-scale behavior in a tractable form. Recent theoretical and empirical studies (Thibeault et al., 2024) probe this hypothesis by analyzing singular value decay in random graphs and real networks, quantifying effective ranks, and linking rapid singular value decay to accurate dimension reductions of nonlinear dynamics. These findings show that low-rank representations enable faithful reduced-order models of recurrent neural networks and other complex systems. Furthermore, rank reduction and low-rank recovery play a central role across applications (Kishore Kumar and Schneider, 2017; Hu et al., 2021). Beyond classical truncated SVD (hard thresholding), a rich set of algorithms—convex relaxations with the nuclear norm, nonconvex optimization, iteratively reweighted schemes, and modern SVD-free iterative methods—has emerged (Chen et al., 2021; Radhakrishnan et al., 2025). Recent advances (Krämer, 2025) revisit affine rank minimization and show that asymptotic log-determinant reweighted least squares can recover low-rank solutions under broad conditions, clarifying both computational and statistical trade-offs of low-rank modeling at scale. In the time-series domain, low-rank structure also provides a powerful inductive bias. For example, Agarwal et al. (2018) reformulates univariate time series into structured low-rank Page matrices so that denoising, imputation, and forecasting reduce to classical matrix estimation problems. More recently, Huang et al. (2025) shows that even ultra-lightweight networks with fewer than 0.4k parameters can achieve state-of-the-art long-term forecasting performance by exploiting low-rank periodic structures and orthogonality-constrained representations, drastically reducing model size and computation while maintaining accuracy.

An operator–theoretic perspective, in which nonlinear dynamics are represented by the infinite-dimensional linear Koopman operator acting on observables, has emerged as a powerful data-driven framework for the prediction, estimation, and control of nonlinear systems (Brunton et al., 2021; Budišić et al., 2012). In practice, this viewpoint is implemented through finite-dimensional approximations of the Koopman operator, most notably Dynamic Mode Decomposition (DMD) and its many extensions, including Extended DMD and kernel-based DMD (Schmid, 2010; Williams et al., 2015). An expanding body of research (Zhang et al., 2025; Liu et al., 2023) now leverages these Koopman-inspired methods to analyze and forecast time series, demonstrating their versatility in extracting low-dimensional linear representations from complex, high-dimensional dynamical data.

## B.2 TERMINOLOGY

**Instance Normalization**    Given a time series historical segment, $\mathbf{y}_{\text{his}} \in \mathbb{R}^L$, $\mu_{\text{his}}$ and $\sigma_{\text{his}}$ the mean and standard deviation of $\mathbf{y}_{\text{his}}$ respectively, and $\delta > 0$ a small constant for numerical stability, Instance normalization (IN) performs the following operations for forecasting:

$$\mathbf{y}'_{\text{his}} = (\mathbf{y}_{\text{his}} - \mu_{\text{his}}) / (\sigma_{\text{his}} + \delta) \implies \hat{\mathbf{y}}'_{\text{fut}} = f_\phi(\mathbf{y}'_{\text{his}}) \implies \hat{\mathbf{y}}_{\text{fut}} = \hat{\mathbf{y}}'_{\text{fut}} \cdot (\sigma_{\text{his}} + \delta) + \mu_{\text{his}}$$

It is also shown in Zeng et al. (2023) and Lin et al. (2024) that simpler operations, such as directly fixing $\sigma_{\text{his}} + \delta = 1$, also works.

All of the operations mentioned above have the same effect on $\mathbf{W}$ for a single-channel linear time series forecasting model, which forces the column sum of $\mathbf{W}$ to become 1 (Toner and Darlow, 2024).

**Channel Independent, Channel Dependent, and Individual Channel Modeling**    Channel Independent (CI), Channel Dependent (CD) (Nie et al., 2022), and Individual Channel Modeling (INC) are three means of handling multi-channel time series.

Consider a multi-channel time series $\{\mathbf{y}_t\}_{t=1}^T$, each of channel 1 to $m$ in this time series can be considered as a single-channel time series denoted as $\{y_t^{(1)}\}_{t=1}^T, \{y_t^{(2)}\}_{t=1}^T, \ldots, \{y_t^{(m)}\}_{t=1}^T$. The corresponding collection of history and future segments are denoted as $\mathbf{Y}_{\text{his}}^{(1)}, \mathbf{Y}_{\text{his}}^{(2)}, \ldots, \mathbf{Y}_{\text{his}}^{(m)} \in \mathbb{R}^{N \times L}$ and $\mathbf{Y}_{\text{fut}}^{(1)}, \mathbf{Y}_{\text{fut}}^{(2)}, \ldots, \mathbf{Y}_{\text{fut}}^{(m)} \in \mathbb{R}^{N \times H}$, respectively.

Firstly, we introduce CD, which has a general modeling target as follows:

$$\min_\phi \left\| \text{stack}\left( \mathbf{Y}_{\text{fut}}^{(1)}, \mathbf{Y}_{\text{fut}}^{(2)}, \ldots, \mathbf{Y}_{\text{fut}}^{(m)} \right) - f_\phi\left( \mathbf{Y}_{\text{his}}^{(1)}, \mathbf{Y}_{\text{his}}^{(2)}, \ldots, \mathbf{Y}_{\text{his}}^{(m)} \right) \right\|_F^2$$

CD is the most general target in time series. Potentially, CD allows errors that arise from any channel to impact predictions of *all* channels. INC can be viewed as a constrained version of CD. For INC, errors that arise from the $i$-th channel only impact predictions of the $i$-th channel, and *all interactions* between channels are dropped. The INC target is as follows, where $f_\phi^{(i)}$'s are *individual* models for each of the $i$-th channel:

$$\min_\phi \sum_{i=1}^m \left\| \mathbf{Y}_{\text{fut}}^{(i)} - f_\phi^{(i)}\left( \mathbf{Y}_{\text{his}}^{(i)} \right) \right\|_F^2$$

Finally, we give the CI target, which can be viewed as posing additional constraints on INC. Unlike INC, a *shared* model $f_\phi$ is used to minimize the following target:

$$\min_\phi \sum_{i=1}^m \left\| \mathbf{Y}_{\text{fut}}^{(i)} - f_\phi\left( \mathbf{Y}_{\text{his}}^{(i)} \right) \right\|_F^2.$$

Particularly, for linear time series models, the CD paradigm can be considered as fitting a matrix of shape $L \cdot m \times H \cdot m$, consisting $m \times m$ blocks of $\mathbf{W}_{i_1 i_2} \in \mathbb{R}^{L \times H}$ ($i_1$ and $i_2$ are two separate indexer over the channels). In this case, the linear CD target is as follows:

$$\min_\phi \sum_{i_2=1}^m \left\| \mathbf{Y}_{\text{fut}}^{(i_2)} - \left( \sum_{i_1=1}^m \mathbf{Y}_{\text{his}}^{(i_1)} \mathbf{W}_{i_1 i_2} \right) \right\|_F^2$$

Following previous logic, in linear time series models, CD pose no constraint on all $\mathbf{W}_{i_1 i_2}$, INC forced $\mathbf{W}_{i_1 i_2} = \mathbf{0}$ for all $i_1 \neq i_2$, and CI further forces $\mathbf{W}_{11} = \mathbf{W}_{22} = \cdots = \mathbf{W}_{mm}$. Their relationships are visualized in Figure 7.

## B.3 PRELIMINARY

**Time series Models and Difference Equations.**    Following the common practice of using CI, where the model operates on single-channel time series (Nie et al., 2022; Zeng et al., 2023; Das et al., 2024), we consider a single-channel noise-free discrete time series $\{y_t^*\}_{t=1}^T$. There are two main approaches to modeling such a time series:

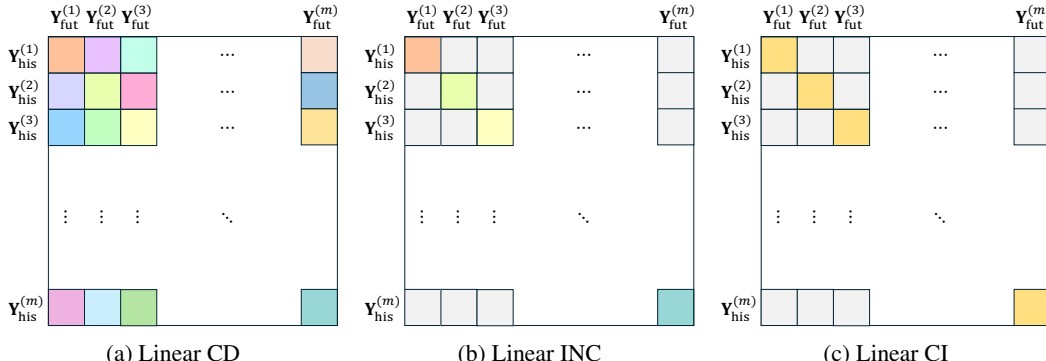

Figure 7: Visualization of CD, INC, and CI for linear time series models. We color **0** matrix blocks with light grey. Matrix blocks that are forced to be equal are colored the same.

1. **Time-based models:** These models directly map each time step $t$ to a predicted value using a function $f_\phi(t)$, such that
$$y_t^* = f_\phi(t).$$
While conceptually straightforward, this method can be difficult to scale or generalize, especially for long or complex sequences, because it treats time as the primary input without explicitly modeling the dynamics between observations.

2. **Structural models (difference equations):** Instead of predicting each value independently, this approach models dependencies between time steps using a function $f_\phi(\cdot)$. It imposes a constraint across consecutive values in the time series:
$$f_\phi(y_{t+1}^*, \underbrace{y_t^*, y_{t-1}^*, \ldots, y_{t-L+1}^*}_{\text{historical segment}}) = 0.$$

This form, known as a *difference equation*, is the discrete analogue of differential equations used in continuous systems.

Modern time series forecasting methods, particularly those based on deep learning, typically focus on learning such structural dependencies. While these models often achieve high predictive performance by implicitly capturing difference equations, their internal mechanisms are usually not interpretable.

**Homogeneous Linear Difference Equations.** We consider the class of homogeneous linear difference equations due to their analytical tractability and structural clarity in modeling temporal dependencies. These equations take the form
$$y_t^* + a_1 y_{t-1}^* + \cdots + a_p y_{t-p}^* = 0,$$
where $p$ is the order of the difference equation and $a_1, \ldots, a_p \in \mathbb{C}$.

More general linear difference equations may include external forcing or constant terms, known as *nonhomogeneous difference equations*. Nonhomogeneous difference equations can often be transformed into homogeneous equations. For instance, it is common to obtain a nonhomogeneous difference equation as follows:
$$y_t^* + a_1 y_{t-1}^* + \cdots + a_p y_{t-p}^* = c,$$
where $b$ is some constant bias. In this example with constant bias, conversion to the homogeneous form is done simply by taking algebraic reduction on the following:
$$y_t^* + a_1 y_{t-1}^* + \cdots + a_p y_{t-p}^* = c = y_{t-1}^* + a_1 y_{t-2}^* + \cdots + a_p y_{t-p-1}^*.$$

This justifies our focus on the homogeneous case. The solution to the homogeneous linear difference equation is obtained via the *characteristic equation*
$$r^p + a_1 r^{p-1} + \cdots + a_p = 0,$$

whose roots, called *characteristic roots* $\{r_i\}_{i=1}^p$ dictate the structure of the general solution. When the roots are distinct, the solution is expressed as

$$y_t^* = C_1 r_1^t + C_2 r_2^t + \cdots + C_L r_p^t,$$

where the constants $C_i$'s are determined by initial conditions. Repeated roots introduce polynomial factors of the form $(A_0 + A_1 t + ... + A_{k-1} t^{k-1}) r^t$, where $k$ is the multiplicity of root $r$. This method provides a closed-form description of the system's evolution, revealing a direct and interpretable link between the underlying structural formulation and explicit predictions across time steps. Such models form a foundational baseline for analyzing and approximating more intricate non-linear or learned temporal systems.

Another way of obtaining the linear difference equation comes from the linear recurrence relation with the general expression given below:

$$\underbrace{\begin{bmatrix} y_t \\ y_{t-1} \\ \vdots \\ y_{t-p+1} \end{bmatrix}}_{t \text{ step segment}} = \underbrace{\begin{bmatrix} c \\ 0 \\ \vdots \\ 0 \end{bmatrix}}_{\text{bias}} + \underbrace{\begin{bmatrix} a_1 & a_2 & \cdots & a_{p-1} & a_p \\ 1 & 0 & \cdots & 0 & 0 \\ \vdots & \vdots & \ddots & \vdots & \vdots \\ 0 & 0 & \cdots & 1 & 0 \end{bmatrix}}_{\mathbf{F} \text{ - the companion matrix}} \underbrace{\begin{bmatrix} y_{t-1} \\ y_{t-2} \\ \vdots \\ y_{t-p} \end{bmatrix}}_{t-1 \text{ step segment}} + \begin{bmatrix} \varepsilon_t \\ 0 \\ \vdots \\ 0 \end{bmatrix}. \tag{4}$$

Under the noise-free condition ($\varepsilon_t = 0$ for all $t$), the homogeneous form ($c = 0$) of Equation equation 4 can be further simplified. In this case, the higher-step future segment is directly obtained through the matrix power of the companion matrix $\mathbf{F}$. This observation naturally leads us to consider the eigenvalue decomposition of $\mathbf{F}$. Notably, the characteristic polynomial of $\mathbf{F}$ coincides with that of the underlying linear difference equation, which was given earlier in this section.

**The Multi-Channel Case.** Our discussion of homogeneous linear difference equations focuses mainly on single-channel time series. While it is common practice to model multi-channel time series by first converting them into multiple single-channel time series in recent works (Zeng et al., 2023; Nie et al., 2022; Eldele et al., 2024), we also cover *Vector Autoregression* (VAR), which is a fundamental multi-channel time series model that characterizes the joint dynamics of multiple variables. VAR without bias for an $m$-dimensional, noise-free time series $\{\mathbf{y}_t^*\}_{t=1}^T$ is a multivariate generalization of the homogeneous linear difference equation mentioned above:

$$\mathbf{y}_t^* = \sum_{i=1}^p \mathbf{A}_i \mathbf{y}_{t-i}^*,$$

where $\{A_i\}_{i=1}^p \in \mathbb{R}^{m \times m}$ are coefficient matrices. A more general form of a VAR($p$) model for an $m$-dimensional (not necessarily noise free) time series $\mathbf{y}_t = (y_t^{(1)}, y_t^{(2)} \ldots, y_t^{(m)})^\top$ is given by:

$$\mathbf{y}_t = \mathbf{c} + \sum_{i=1}^p \mathbf{A}_i \mathbf{y}_{t-i} + \boldsymbol{\varepsilon}_t, \tag{5}$$

where $\mathbf{c}$ is an $m \times 1$ vector of intercept terms, and $\boldsymbol{\varepsilon}_t \in \mathbb{R}^m$ is a white noise process with $\mathbb{E}[\boldsymbol{\varepsilon}_t] = \mathbf{0}$ and $\text{Cov}(\boldsymbol{\varepsilon}_t) = \boldsymbol{\Sigma}$.

For a homogeneous system ($\mathbf{c} = \mathbf{0}$) without noise ($\boldsymbol{\varepsilon}_t = \mathbf{0}$ for all $t$) described in Equation (5), the underlying system's dynamics is determined by the characteristic polynomial as follows:

$$\det \left( \mathbf{I}_m - \sum_{i=1}^p \mathbf{A}_i z^i \right) = 0. \tag{6}$$

The companion form representation for VAR is given as follows, which is also a multidimensional generalization of the companion matrix for the univariate case in Equation (4).

$$\underbrace{\begin{bmatrix} \mathbf{y}_t \\ \mathbf{y}_{t-1} \\ \vdots \\ \mathbf{y}_{t-p+1} \end{bmatrix}}_{t \text{ step segment}} = \underbrace{\begin{bmatrix} \mathbf{c} \\ \mathbf{0} \\ \vdots \\ \mathbf{0} \end{bmatrix}}_{\text{bias}} + \underbrace{\begin{bmatrix} \mathbf{A}_1 & \mathbf{A}_2 & \cdots & \mathbf{A}_{p-1} & \mathbf{A}_p \\ \mathbf{I}_m & \mathbf{0} & \cdots & \mathbf{0} & \mathbf{0} \\ \vdots & \vdots & \ddots & \vdots & \vdots \\ \mathbf{0} & \mathbf{0} & \cdots & \mathbf{I}_m & \mathbf{0} \end{bmatrix}}_{\mathbf{F} \text{ - the companion matrix}} \underbrace{\begin{bmatrix} \mathbf{y}_{t-1} \\ \mathbf{y}_{t-2} \\ \vdots \\ \mathbf{y}_{t-p} \end{bmatrix}}_{t-1 \text{ step segment}} + \begin{bmatrix} \boldsymbol{\varepsilon}_t \\ \mathbf{0} \\ \vdots \\ \mathbf{0} \end{bmatrix}, \tag{7}$$

where $\mathbf{F}$ is the $mp \times mp$ companion matrix. Notice solving for the characteristic roots by Equation (6) is equivalent to solving for the eigenvalues of the companion matrix in Equation (7). VAR models are widely employed in forecasting and structural analysis, with tools such as impulse response functions and variance decompositions providing insights into dynamic interactions. Extensions like structural VAR (SVAR) incorporate identifying restrictions for causal inference, while Bayesian VAR (BVAR) introduces shrinkage priors to improve estimation in high-dimensional settings.

**Stability of AR/VAR Models.** In classical time series analysis, the stability of AR and VAR models is usually an important consideration. Stability requires that the characteristic roots lie outside the unit circle, ensuring that the process is stationary and that forecasts do not diverge over time (Shumway and Stoffer, 2006). This condition matters when the model is used to describe the underlying data-generating process or when iterative forecasts are produced by repeatedly feeding predictions back into the model. However, in the context of direct multi-horizon forecasting, stability is not a strict requirement. Each forecast horizon is modeled and estimated independently, and the prediction for a given horizon is produced only once, without recursive dependence on earlier predictions (as shown in Equation (2)). As a result, even if the estimated AR/VAR coefficients correspond to an unstable system, this does not affect the validity of the forecasts at the specified horizons. The forecasts remain well-defined because they are not generated through repeated iteration of the unstable dynamics.

## C    REINTERPRETING DESIGN CHOICES IN LINEAR TIME SERIES MODELS

### C.1    BIAS-FREE LINEAR MODELS: JUSTIFICATION AND IMPLICATIONS

A common yet sometimes unexamined modeling decision is the omission of a bias term. We offer three principled reasons for this choice. First, from a theoretical standpoint, linear difference equations that include a constant bias term can be algebraically transformed into equivalent homogeneous forms; Second, standard normalization, especially instance normalization, centers input and target sequences, making bias terms negligible for sufficiently long sequences; Third, empirical results (Appendix E.6) show bias-free models match or outperform biased alternatives. This simplification preserves expressive power while streamlining the architecture.

### C.2    HIGHER-HORIZON FORECASTING AND EXTENDED LOOKBACK WINDOW

Recent time series models tend to directly give predictions for the entire forecasting horizon without recursive calls (Zhou et al., 2021; Zeng et al., 2023). This is because such practice may help alleviate cumulative errors in recursive predictions and improve performance. We thus turn our focus to the higher-horizon forecasting setting. Rather than predicting the next time step and applying iterative rollout, we model each forecast horizon independently. This results in $H$ distinct regressions, one for each future time step. Specifically, the predictive equation for the $j$-th horizon takes the following form when the model achieves zero training loss in an ideal noise-free situation:

$$y_{t+j} - \sum_{k=1}^{L} W_{kj} \cdot y_{t-k+1} = 0. \tag{8}$$

This formulation can be interpreted as a linear difference equation. Each such equation implicitly defines a set of characteristic roots that govern the dynamics at the corresponding forecast horizon. Notably, models for higher horizons tend to incorporate more temporal information. This leads to the following structural observation:

**Claim 1** (Part I of Fact 2). *The characteristic root set of a higher-horizon linear model contains, as a subset, the roots from basic dynamics.*

This claim suggests a hierarchical organization in temporal dynamics: models for longer-term forecasts preserve the essential roots needed for short-term accuracy while allowing more expressive representations. This supports the common practice of treating each horizon independently and further implies that higher-horizon models do not contradict, but rather generalize, the underlying system dynamics.

Another fundamental modeling choice is the length of the input (lookback) window $L$. When the underlying dynamics admit a minimal recurrence relation of order $K$, choosing $L > K$ introduces

redundancy. In such cases, the regression admits multiple parameterizations that yield identical predictive behavior:

**Claim 2** (Part II of Fact 2). *If the minimal recurrence order is $K$, then any lookback window of length $L > K$ yields non-unique representations that all preserve the characteristic roots of the underlying dynamics.*

While this non-uniqueness does not harm predictive performance, it underscores the flexibility of the model to encode equivalent dynamics in multiple ways.

The formal version and proof of Fact 2 are provided in Section D.2.

### C.3  TOY EXAMPLE

To illustrate how characteristic roots govern the forecasting behavior of linear time series models, we consider a simple noise-free example:
$$y(t) = 0.01t^2 + \sin t.$$
This system admits five characteristic roots, $\{e^i, -e^i, 1, 1, 1\}$, corresponding to the following general solution with coefficients $A, B, C, D, E$:
$$y_{\text{gen}}(t) = (At^2 + Bt + C) \cdot 1^t + D(\cos t + i \sin t) + E(\cos t - i \sin t).$$
This general solution can be exactly represented by a linear forecasting model with a lookback window of length 5 and a forecasting horizon of length 1:
$$y_t + a_1 y_{t-1} + \cdots + a_5 y_{t-5} = 0.$$
Following standard practice, we form the history and future matrices, $\mathbf{Y}_{\text{his}} \in \mathbb{R}^{N \times 5}$ and $\mathbf{Y}_{\text{fut}} \in \mathbb{R}^{N \times 1}$, and compute
$$\mathbf{W} = \left(\mathbf{Y}_{\text{his}}^\top \mathbf{Y}_{\text{his}}\right)^{-1} \mathbf{Y}_{\text{his}}^\top \mathbf{Y}_{\text{fut}}.$$
Empirically, we find that the characteristic polynomial of $\mathbf{W} \in \mathbb{R}^5$ yields the roots $\{e^i, e^{-i}, 1, 1, 1\}$. Since the roots $\{e^i, e^{-i}, 1, 1, 1\}$ are preserved, the model recovers the dynamics exactly, consistent with our observation of zero training and test errors.

Based on Fact 1, the role of roots also explains the model's generalization ability in the following two cases:

1. $\mathbf{W}$ can also perfectly forecast $x(t) = t + \cos t$, which has characteristic roots $\{e^i, -e^i, 1, 1\}$, a subset of $\{e^i, e^{-i}, 1, 1, 1\}$; see Figure 8, Left;

2. $\mathbf{W}$ *cannot* forecast $z(t) = \cos(1.1t)$, which has characteristic roots $\{e^{1.1i}, e^{-1.1i}\}$, not a subset of $\{e^i, e^{-i}, 1, 1, 1\}$; see Figure 8, Right.

More generally, for any lookback window length $\geq 5$ and arbitrary forecast horizon, the resulting matrix $\mathbf{W}_{\text{general}}$ introduces additional roots. However, these roots play no role in the forecasts, provided that $\{e^i, -e^i, 1, 1, 1\}$ remain among the characteristic roots. In this case, perfect forecasting is always guaranteed.

### C.4  INSTANCE NORMALIZATION AND CHANNEL INDEPENDENT MODELING

In this section, we discuss in detail how Instance Normalization (IN), Channel Independent (CI) Modeling, and other channel modeling methods (CD, INC) fits in our framework. You can refer to Appendix B.2 for more explanation of these terms.

**Instance Normalization (IN)**  As an application of our characteristic-root-based analysis in Section 3, instance normalization introduces a structural inductive bias aligned with this framework. By centering each input sequence, normalization implicitly enforces a unit root at $r = 1$, corresponding to level invariance in the dynamics. To see how we end up with this special root, we follow the proofs in Toner and Darlow (2024). For a linear model, the effect of IN on $\mathbf{W} \in \mathbb{R}^{L \times H}$ is that the column sum of $\mathbf{W}$ is 1, as follows:
$$\sum_{k=1}^{L} W_{kj} = 1, \quad \forall j = 1, 2, \ldots, H.$$

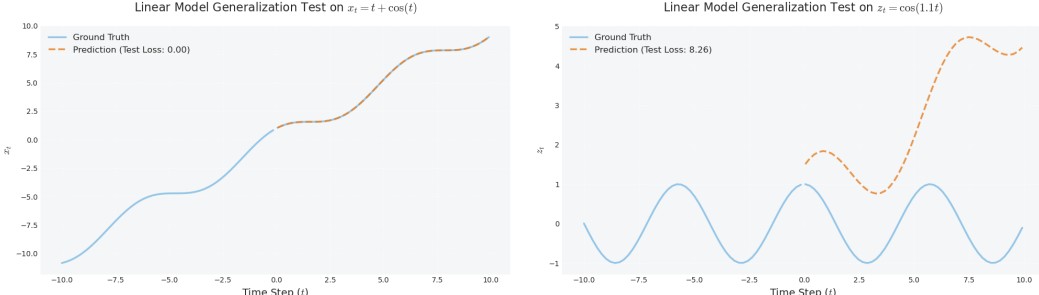

Figure 8: Generalization test of $\mathbf{W}$ forecasting the time series $y(t) = 0.01t^2 + \sin t$. Left: success of generalization to $x(t) = t + \cos t$. Right: failure of generalization to $z(t) = \cos(1.1t)$.

We then follow Equation (8) to establish the difference equation. It is important to notice that $y_{t+j}$ has a natural leading coefficient of 1, as a result, the leading coefficients of the characteristic polynomial sum to 0, suggesting the existence of a characteristic root of 1.

This special root enables the model to maintain consistency across sequences with varying mean shifts. This should be intuitive to see, as the general solution with a root being 1 is now the following:

$$y_t = C_1 r_1^t + C_2 r_2^t + \cdots + C_p 1^t = C_1 r_1^t + C_2 r_2^t + \cdots + C_p.$$

The undetermined $C_p$ in the general solution allows any mean shifts to become automatically generalizable.

**Channel Independent (CI) Modeling**   Our analysis in Section 3 also helps to explain the empirical success of channel-independent modeling in multi-channel time series. Although individual channels may exhibit distinct dynamics, modeling them in a shared fashion can still be effective. The key insight is that the overparameterization introduced by a sufficiently large $L$ provides enough representational capacity to accommodate the union of characteristic roots across channels, even without explicit cross-channel interactions. Without loss of generality, we assume that we have two channels, where the first channel has linear dynamics with roots $r_{1,1}, r_{1,2}, ..., r_{1,K_1}$ and the second channel has linear dynamics with roots $r_{2,1}, r_{2,2}, ..., r_{2,K_2}$ By Fact 1, as long as the model's characteristic root set $R$ contains the following set as a subset, the model can generalize to both the first and the second channel:

$$R_{\cup} = \{r_{1,1}, r_{1,2}, ..., r_{1,K_1}\} \cup \{r_{2,1}, r_{2,2}, ..., r_{2,K_2}\}$$

For instance, if the model is used to forecast the first channel now, it can be done by simply setting the $C$'s for roots in $R \setminus \{r_{1,1}, r_{1,2}, ..., r_{1,K_1}\}$ as 0. Similar things can also be done for the second channel to achieve generalization. When the roots are highly shared between two channels, CI is an efficient strategy as the cardinality of $R_{\cup}$ is much smaller than the capacity required to learn $\{r_{1,1}, r_{1,2}, ..., r_{1,K_1}\}$ and $\{r_{2,1}, r_{2,2}, ..., r_{2,K_2}\}$ separately.

The root sharing mechanism explains why CI models can use a shared weight $\mathbf{W}$ to model many channels, allowing better parameter efficiency. The performance improvements, on the other hand, is explained with the data scaling property as described in Proposition 1, as training the shared weight with more data improves its robustness.

**A VAR Perspective on CI**   Another valuable perspective on channel-independent modeling is through vectorized autoregressive (VAR) modeling, where the system's behavior can also be captured by the characteristic roots obtained from solving the characteristic equation (with the multivariate generalization as in Equation (6)). This approach is particularly interesting because the number of characteristic roots is directly related to both the order of the process, $p$, and the number of channels, $m$. Specifically, the total number of roots equals $p \times m$, reflecting the fact that the system evolves with a given order over multiple independent channels. In this framework, channel independence can be seen as a degenerate case where the autoregressive process is simplified by treating all matrices associated with the variables as diagonal matrices, with each diagonal element identical. More formally, this means that the coefficient matrices, denoted as $\mathbf{A}_i$, are of the form $\mathbf{A}_i = k_i \times \mathbf{I}$, where

$k_i$ is a scalar constant, and $\mathbf{I}$ is the identity matrix. This simplification implies that each channel behaves independently, with its evolution governed by the same recurrence parameter $k_i$, and no interactions between different channels occur. Furthermore, we have the following results:

**Proposition 3** (Equivalent Representation of Diagonal Recurrences). *Let $\{\mathbf{y}_t \in \mathbb{R}^m\}$ be a vector time series following a diagonal matrix recurrence of order $p$:*

$$\mathbf{y}_t = \sum_{i=1}^{p} \mathbf{D}_i \mathbf{y}_{t-i},$$

*where $\mathbf{D}_i = diag(d_i^{(1)}, \ldots, d_i^{(m)})$ are diagonal matrices. Then, there exists an equivalent order-$L$ recurrence (where $L \leq mp$) that can be written as:*

$$\mathbf{y}_t = \sum_{i=1}^{L} \mathbf{D}_i' \mathbf{y}_{t-i},$$

*where each $\mathbf{D}_i' = k_i \mathbf{I}_m$ is a scaled identity matrix, with $k_i \in \mathbb{R}$ and $\mathbf{I}_m$ being the identity matrix of size $m$.*

*Proof.* Let $\mathbf{y}_t = (y_t^{(1)}, \ldots, y_t^{(m)})^\top$. The given recurrence decomposes into $m$ independent scalar recurrences:

$$y_t^{(j)} = \sum_{i=1}^{p} d_i^{(j)} y_{t-i}^{(j)}, \quad \text{for } j = 1, \ldots, m.$$

Each scalar recurrence has an associated characteristic polynomial:

$$P_j(r) = r^p - \sum_{i=1}^{p} d_i^{(j)} r^{p-i}, \quad j = 1, \ldots, m.$$

Let $Q_j(r)$ be the minimal polynomial of $\{y_t^{(j)}\}$, which divides $P_j(r)$. The Least Common Multiple (LCM) of all $Q_j(r)$, denoted as $Q(r)$, is a monic polynomial of degree $L \leq mp$ that annihilates every $\{y_t^{(j)}\}$. Thus, all components satisfy:

$$y_t^{(j)} = \sum_{i=1}^{L} k_i y_{t-i}^{(j)}, \quad \text{for } j = 1, \ldots, m,$$

where $\{k_i\}_{i=1}^{L}$ are the coefficients of $-Q(r)$. In vector form, this becomes:

$$\mathbf{y}_t = \sum_{i=1}^{L} k_i \mathbf{y}_{t-i} = \sum_{i=1}^{L} \mathbf{D}_i' \mathbf{y}_{t-i},$$

where $\mathbf{D}_i' = k_i \mathbf{I}_m$. This proves the equivalence. $\square$

The equivalence between the original diagonal matrix recurrence and the transformed recurrence indicates that while the original system uses different diagonal matrices for each lag, the equivalent system employs a uniform scalar scaling (represented by $k_i \mathbf{I}_m$) across all channels at each time step. This simplification removes the structure of individual diagonal elements in favor of a single scaling factor for each lag, which effectively reduces the complexity of the model. The key takeaway is that both formulations describe the same system, but the latter offers a more compact and simplified representation. This result implies that a system with independent channels—each governed by a diagonal matrix with potentially different coefficients—can be equivalently represented by a system where the same scalar $k_i$ governs all channels at each time step. This reduction in complexity is particularly useful for simplifying the analysis of high-dimensional systems, as it shows how the system dynamics can be captured using fewer parameters, namely the scalars $k_i$, rather than a full set of diagonal matrices for each lag.

**Remark 3** (Technical Condition for General Matrices). *Considering a more general case of a matrix recurrence of the form*

$$\mathbf{y}_t = \sum_{i=1}^{p} \mathbf{A}_i \mathbf{y}_{t-i},$$

*where $\mathbf{A}_i \in \mathbb{R}^{m \times m}$ are arbitrary matrices (not necessarily diagonal), the components of the sequence $\mathbf{y}_t \in \mathbb{R}^m$ interact at each time step due to their dependence on multiple previous states $\mathbf{y}_{t-i}$. Unlike simpler cases, where recurrences can sometimes be expressed using scaled identity matrices, this general form does not have a similar proposition or a straightforward structure. The matrices $\mathbf{A}_i$ typically cannot be reduced to a form involving only scaled identity matrices unless they satisfy the restrictive condition of being simultaneously diagonalizable. In cases where the matrices are not simultaneously diagonalizable, the recurrence introduces coupled dynamics between the channels of $\mathbf{y}_t$, making the analysis significantly more complex. These coupled interactions prevent an easy separation or simplification of the system, which means that more advanced techniques are required to analyze the interdependencies and predict the behavior of the system over time.*

**Connections of VAR and Linear Channel Dependent Modeling**  Notice that the perspective of restricting VAR to model channel independence is very similar to the process in Figure 7. Indeed, VAR and linear CD models are deeply connected. Precisely, a linear CD model (with no bias) with a lookback window of length $L$ and a forecasting horizon $H$ is equivalent to a collection of $H$ VAR($L$) models as follows:

$$\begin{cases} \mathbf{y}_{t+1} = \sum_{i=0}^{L-1} \mathbf{A}_i^{(1)} \mathbf{y}_{t-i} + \boldsymbol{\varepsilon}_{t+1} \\ \mathbf{y}_{t+2} = \sum_{i=0}^{L-1} \mathbf{A}_i^{(2)} \mathbf{y}_{t-i} + \boldsymbol{\varepsilon}_{t+2} \\ \quad\vdots \\ \mathbf{y}_{t+H} = \sum_{i=0}^{L-1} \mathbf{A}_i^{(H)} \mathbf{y}_{t-i} + \boldsymbol{\varepsilon}_{t+H} \end{cases} \tag{9}$$

The parameter collection as described in Equation (9) aligns with those described in Appendix B.2 via basic tensor operations—we can stack all $\mathbf{A}_i^{(j)}$ appeared to make a 4-D tensor of shape $L \times H \times m \times m$. By permuting the axes orders and flattening it to make a matrix of shape $L \cdot m \times H \cdot m$, we can arrive at the CD learnable parameter set discussed before.

Following the convention in VAR, here we utilize the form of the parameter set described in Equation (5). However, it is important to see that it is closely connected to channel-dependent modeling used in the field of machine learning.

**CI, INC, and CD in Linear Time Series Models**  Building on our previous discussion of CI from both top-down and bottom-up views, we summarize our theoretical and empirical insights on using CI modeling, even in when the channels of a multivariate time series interact in complex ways. Furthermore, we go beyond from CI and discuss the practical potential of additional key paradigms—Individual Channel (INC), and Channel-Dependent (CD)—as defined in Appendix B.2.

- **Broad Use of CI is a Consequence:** Specifically, if each channel can be individually represented by a linear model (a much weaker and realistic assumption by Corollary 1 and Corollary 2), then the multivariate time series as a whole can be effectively modeled by applying the same linear operator across channels. We emphasize that *no assumption on channel relationships* is needed. This significantly broadens the applicability of CI modeling to real-world datasets, where channels may be correlated or interact in complex ways.

- **Benefit of CI in Practice:** Beyond expressiveness, Proposition 1 explains the robustness benefits of CI modeling. In particular, CI methods aggregate noise across channels, which effectively acts as implicit data augmentation. This makes CI models more robust under finite-sample conditions, especially when training data is noisy or limited.

- **Empirical Justification:** To address the challenge of limited data, we propose two inductive bias strategies — rank reduction and root purge — which are designed to improve model robustness by introducing structural regularization. The effectiveness of these strategies is reflected in our main experimental results (Table 1). In addition, we conducted controlled experiments comparing channel-independent (CI) and channel-individual (INC) modeling (see Table 3). These experiments reveal several key tradeoffs:

- CI models already benefit from shared information across channels, which reduces the marginal benefit from root purge.
- INC models, in contrast, have higher capacity and flexibility by design, which allows them to benefit more from root purge, especially under large-scale datasets such as ETTm and ETTh, where data sparsity is less of a concern.

Table 6: Comparison of common channel modeling approaches (CI, INC, and CD) in linear time series forecasting models.

| Modeling Choices | Attributes | |
|---|---|---|
| **CI** (*Channel-Independent*) | **Description** | All channels share the same linear representation |
| | **Capacity** | Low ($L \times H$) |
| | **Data Scaling** | Works well |
| | **Benefit from Our Methods** | Moderate |
| | **Practicality** | More practical |
| **INC** (*Individual Channel*) | **Description** | Each channel uses a separate linear model |
| | **Capacity** | High ($C \times H \times L$) |
| | **Data Scaling** | Suffers |
| | **Benefit from Our Methods** | Good |
| | **Practicality** | Feasible when the number of channels is small |
| **CD** (*Channel-Dependent*) | **Description** | All channels are jointly modeled using a linear model |
| | **Capacity** | High ($C \times C \times H \times L$) |
| | **Data Scaling** | Suffers |
| | **Benefit from Our Methods** | Moderate (difficult to optimize at scale) |
| | **Practicality** | Hard to implement due to memory overhead |

To finally clarify the distinction of CI, INC, and CD, we now compare these three primary channel modeling paradigms and summarize their key similarities and differences across several critical aspects in Table 6.

## C.5 Scaling in Pure Noise Series

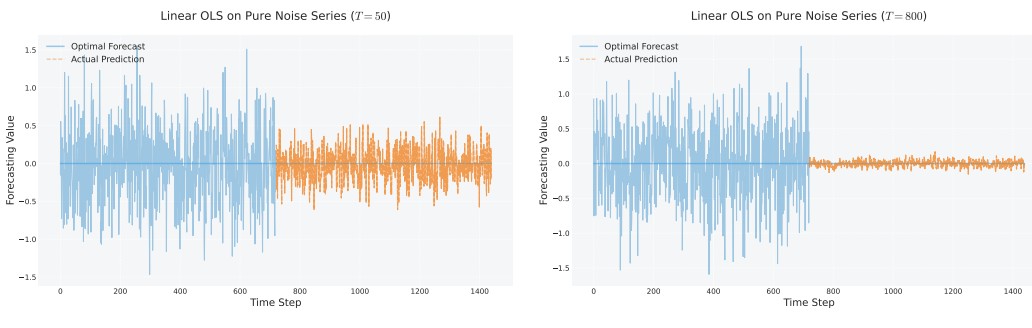

(a) OLS on a small scale of pure noise series        (b) OLS on a large scale of pure noise series

Figure 9: Qualitative visualization of an OLS model forecasting pure noise. Since noise is inherently unpredictable, the optimal forecast is a horizontal line at zero. (a) With limited data, the OLS fit shows significant deviation from this optimum. (b) As the dataset size increases, the model's forecast converges to the optimal solution, consistent with the theoretical scaling law in Proposition 1.

In practice, time series models will almost never encounter the noise-free case. Thus, a typical time series can be viewed as a superposition of two components: the true underlying signal and additive noise. Given the linearity of the forecasting model, the learned weight matrix $\mathbf{W}$ naturally acts on both components simultaneously, as follows:

$$\mathbb{E}\left[\|(\mathbf{y}_{\text{fut}}^* - \mathbf{W}^\top \mathbf{y}_{\text{his}}^*) + (\boldsymbol{\varepsilon}_{\text{fut}} - \mathbf{W}^\top \boldsymbol{\varepsilon}_{\text{his}})\|_2^2\right].$$

In the idealized analysis presented in Section 3.2, we separate the desired properties of an "ideal" linear model $\mathbf{W}$ into two parts:

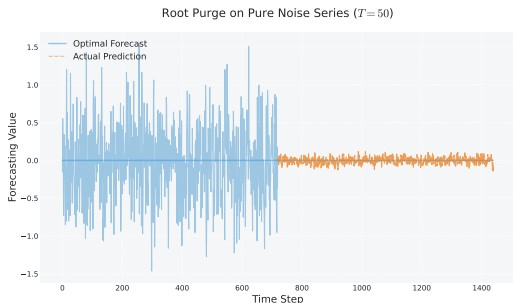 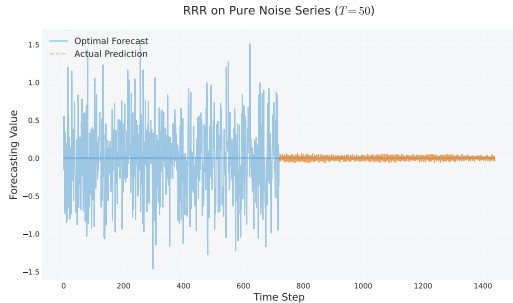

(a) Root Purge on a small scale of pure noise series      (b) RRR on a small scale of pure noise series

Figure 10: Qualitative demonstration of forecasting performance on pure noise. Models fitted with our proposed methods (RRR and Root Purge) generate predictions that converge significantly toward the optimal forecast (zero). Notably, their performance is comparable to an Ordinary Least Squares (OLS) model trained on a dataset 16 times larger.

1. It should perfectly recover and predict the signal dynamics (not including the noise term), i.e., map the historical signal part of the time series to the corresponding future signal values;

2. It should completely ignore the historical noise component, i.e., map it to zero, since noise is *unpredictable* by nature.

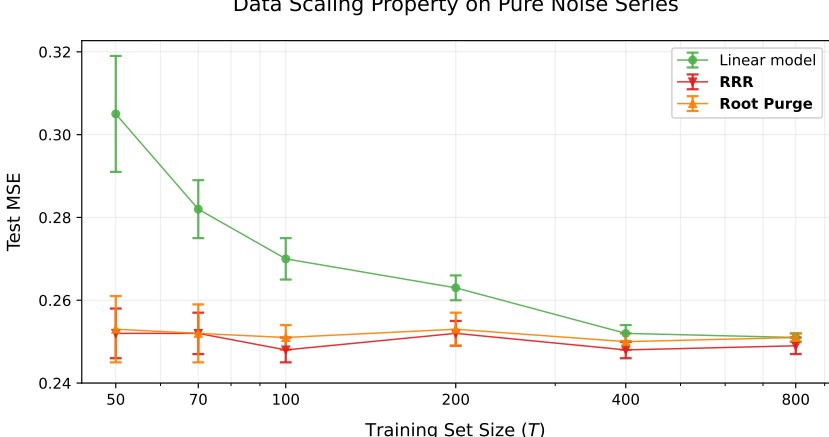

Figure 11: Mean Squared Error on pure noise series of varying sizes. Consistent with Proposition 1, OLS-optimized linear models require substantial data to converge to the optimal forecast, despite the fitting target being pure noise and the optimal prediction being trivially zero. Conversely, both RRR and Root Purge demonstrate robust performance, showing good performance even with limited data.

Both properties are essential for the forecasting performance: even if $\mathbf{W}$ can perfectly capture the signal dynamics, if it also maps noise in history to a biased quantity in the forecast, the prediction quality will suffer.

This effect can be better demonstrated with Figure 9 and Figure 10. We can imagine that even if a linear time series forecasting model can perfectly predict the noise-free dynamics, the output will still be corrupted by the term $\mathbf{W}^\top \varepsilon_{\mathrm{his}}$. If $\mathbf{W}^\top \varepsilon_{\mathrm{his}}$ is a large noise as shown in Figure 9a, this corruption will be strong, making the forecast suboptimal. On the contrary, if $\mathbf{W}^\top \varepsilon_{\mathrm{his}}$ produce output as shown in Figure 9a (by using lots of data), and Figures 10a and 10b (by using our methods), the forecasting performance can be significantly improved.

Notice that the previous analysis naturally establishes a **lower bound** for how well a linear time series forecasting model can perform by assuming that our model fits the noise-free dynamics perfectly. This

is similar to our analysis in Section 3.2, where we stated "In the idealized case where $\mathbf{W}$ perfectly recovers the signal dynamics, only the noise term remains." Studying this lower bound allows us to clearly see the fundamental limitation of learning a linear time series forecasting model with solely the MSE loss, as the lower bound of test MSE remains high, especially with limited training cases. This result is promised theoretically by Proposition 1 as well as empirically by Figure 11.

On the other hand, we demonstrate the superior scaling behavior of RRR and Root Purge in Figure 11 as they *actively* suppress noises in their optimization processes. Therefore, RRR and Root Purge unlock the potential of the linear time series forecasting model to perform better than those trained with solely MSE losses, and their empirical improvements are evidently shown in Table 1.

## D  FORMAL THEORIES AND PROOFS

### D.1  CHARACTERISTIC ROOTS GENERALIZATION

We give a detailed and formal version of Fact 1 and its proof as follows:

**Proposition 4** (Characteristic Roots Generalize up to Sequence Initial Conditions). *Let $r_1, \ldots, r_L \in \mathbb{C}$ be distinct complex numbers, and let $\{r_1, \ldots, r_p\} \subset \{r_1, \ldots, r_L\}$ for some $p < L$. Suppose $c_1, \ldots, c_p \in \mathbb{C}$ and $b_1, \ldots, b_L \in \mathbb{C}$ are such that*

$$\sum_{j=1}^{p} c_j r_j^{t_i} = \sum_{j=1}^{L} b_j r_j^{t_i}, \quad for \ i = 1, \ldots, L,$$

*where $t_1, \ldots, t_L \in \mathbb{Z}$ are distinct integers. Then, for all $t \in \mathbb{Z}$, it holds that*

$$\sum_{j=1}^{p} c_j r_j^{t} = \sum_{j=1}^{L} b_j r_j^{t}.$$

*Proof.* Define the function

$$h(t) := \sum_{j=1}^{p} c_j r_j^{t} - \sum_{j=1}^{L} b_j r_j^{t} = \sum_{j=1}^{L} d_j r_j^{t},$$

where

$$d_j = \begin{cases} c_j - b_j & \text{for } j = 1, \ldots, p, \\ -b_j & \text{for } j = p+1, \ldots, L. \end{cases}$$

Then $h(t_i) = 0$ for $i = 1, \ldots, L$ by assumption. Let $V \in \mathbb{C}^{L \times L}$ be the generalized Vandermonde matrix with entries $V_{i,j} = r_j^{t_i}$. Since the $r_j$ are distinct and the $t_i$ are distinct, $V$ is invertible. Hence, the equation

$$Vd = 0$$

implies $d = 0$, and thus all $d_j = 0$. Therefore,

$$\sum_{j=1}^{p} c_j r_j^{t} = \sum_{j=1}^{L} b_j r_j^{t} \quad \text{for all } t \in \mathbb{Z}.$$

$\square$

The proposition demonstrates a powerful uniqueness result: if a linear combination of exponential functions (with distinct complex bases $r_j$) agrees with another (potentially overcomplete) combination at $L$ distinct time points, then both expressions must be identical at all time points. This result can be extended to situations where roots are repeated. In such cases, the basis functions generalize to include terms of the form $(A_0 + A_1 t + \ldots + A_{k-1} t^{k-1}) r^t$, where $k$ accounts for the multiplicity of the root $r$. These generalized exponentials still form a linearly independent set under suitable conditions, and similar linear algebraic arguments (e.g., involving confluent Vandermonde matrices or the Wronskian) show that equality on a sufficient number of time points again guarantees equality everywhere.

This result directly implies that a linear model can represent any time series whose characteristic roots form a subset of the model's own roots. In other words, if a time series is generated by a set of exponentials associated with certain roots, then any linear model whose characteristic polynomial includes those roots, possibly along with additional ones, will still be able to reproduce the series exactly, provided the coefficients associated with the redundant roots are set to zero.

Based on the earlier proposition, we can formulate a corollary that highlights the expressivity of linear models:

**Corollary 1** (Expressivity of Linear Recurrence Models). *Let a time-dependent signal $y(t)$ be composed of functions of the following form:*

$$y(t) = \prod_{n \in \mathbb{N}} \left\{ \left[ \sum_{i \in I_n} (a_i \cos(\omega_{n,i} t) + b_i \sin(\omega_{n,i} t)) \right] + \left[ \sum_{p \in P_n} a_p t^p \right] + \left[ \sum_{r \in R_n} a_r e^{rt} \right] \right\},$$

*where $I_n, P_n, R_n$ are finite index sets for each $n \in \mathbb{N}$, and all coefficients $a_i, b_i, a_p, a_r \in \mathbb{C}$, frequencies $\omega_{n,i} \in \mathbb{R}$, and exponents $r \in \mathbb{C}$. Then any such function $y(t)$ can be perfectly represented as a solution to a linear difference equation with constant coefficients.*

This corollary highlights the expressive power of linear models with constant coefficients. Such models can perfectly represent signals composed of sinusoids, polynomials, exponentials, and their combinations — forms commonly found in real-world time series. Moreover, the exponential and sinusoidal functions form a complete basis in many function spaces. This means that even when exact representation is impossible, linear models can approximate complex signals with small error, making them highly effective for both modeling and prediction.

The expressivity of linear recurrence models can be understood more formally through the lens of *differential Galois theory* (Singer, 2009; Van der Put and Singer, 2012). In this framework, consider a time-dependent signal $y(t)$ that satisfies a linear differential (or difference) equation with constant coefficients[5]. Let $K$ denote the base field (e.g., rational functions in $t$), and let $L$ be the *Picard–Vessiot (PV) extension* generated by $y(t)$ over $K$. Then the set of all solutions to the associated linear system forms $L$, which is a minimal differential field extension closed under the operations required to solve the equation.

In this language, the linear recurrence model is *expressive* for a given signal $y(t)$ if and only if the Picard–Vessiot extension generated by $y(t)$ is contained in the extension generated by the model's characteristic roots. In other words, a linear model can exactly represent $y(t)$ precisely when the solution field $L$ of $y(t)$ is a *subextension* of the model's PV extension. This perspective allows a clean generalization: for any family of signals whose PV extensions are nested subextensions of a model's extension, a single linear recurrence system is guaranteed to represent all signals in the family. Conversely, if the PV extension of a target signal lies outside the model's extension, no linear model with the given characteristic roots can represent it exactly. Formally, we can restate Corollary 1 as follows:

**Corollary 2** (Differential Galois formulation). *Let $y(t)$ be a time-dependent signal and $K$ the base differential field. Let $L_y/K$ denote the Picard–Vessiot extension generated by $y(t)$, and let $L_M/K$ denote the PV extension corresponding to a linear recurrence model $M$ with characteristic roots $\{\lambda_i\}$. Then $y(t)$ can be represented by $M$ if and only if*

$$L_y \subseteq L_M.$$

*Equivalently, in terms of differential Galois groups,*

$$\mathrm{DGal}(L_M/K) \twoheadrightarrow \mathrm{DGal}(L_y/K),$$

*i.e. the differential Galois group of $L_y$ is a quotient of the differential Galois group of $L_M$.*

The differential Galois perspective can be further extended to the case where the coefficients of the linear recurrence or differential equation are not constant, but are functions of time or other variables. In this setting, the classical notion of characteristic roots does not directly apply. Nevertheless, one

---

[5]Unlike in differential Galois theory, the Picard–Vessiot extension in difference Galois theory is often only a ring (not a field) and one usually needs to localize it to handle zero divisors. In the sequel, we focus on the differential equation case for ease of exposition.

can still describe the expressive or generalization capacity of a model in terms of *Picard–Vessiot extensions*. Let the linear system now be

$$y^{(n)}(t) + a_{n-1}(t)y^{(n-1)}(t) + \cdots + a_0(t)y(t) = 0,$$

where the coefficients $a_i(t)$ are functions in the base differential field $K$. Denote by $L_y/K$ the Picard–Vessiot extension generated by a target signal $y(t)$, and by $L_M/K$ the PV extension generated by the linear model $M$ with functional coefficients. Then, as in the constant coefficient case, the model $M$ can exactly represent $y(t)$ if and only if $L_y \subseteq L_M$. This formulation shows that the PV extension framework naturally generalizes beyond constant coefficient systems. In particular, it provides a unified algebraic criterion for model expressivity and generalization: a linear model with functional coefficients can capture any signal whose solution field is contained within its PV extension, even when classical characteristic roots do not exist. Consequently, the concept of *subextension* remains the fundamental criterion for representability and generalization in the broader class of time-dependent linear systems.

## D.2 LINEAR MODELING PROPERTY UNDER NOISE-FREE SETTING

We give a detailed and formal version of Fact 2 and its proof as follows:

**Proposition 5** (Characteristic Polynomial Divisibility under Recurrence Extension). *Let a univariate linear dynamical system be governed by a minimal-order linear recurrence relation*

$$y_{t+1} = a_1 y_t + a_2 y_{t-1} + \cdots + a_p y_{t-p+1},$$

*with characteristic polynomial*

$$P(r) = r^p - a_1 r^{p-1} - \cdots - a_p.$$

*Suppose the same dynamics can also be expressed via a higher-order recurrence*

$$y_{t'+1} = b_1 y_{t'} + b_2 y_{t'-1} + \cdots + b_q y_{t'-q+1}, \quad q \geq p,$$

*with characteristic polynomial*

$$Q(r) = r^q - b_1 r^{q-1} - \cdots - b_q.$$

*Then $P(r)$ divides $Q(r)$; that is, there exists a polynomial $S(r)$ of degree $q - p$ such that*

$$Q(r) = P(r)S(r).$$

*Proof.* Let $P(r) \in \mathbb{R}[r]$ be the *minimal annihilating polynomial* for the sequence $\{y_t\}$. By definition, $P(r)$ is the monic polynomial of least degree such that the associated linear difference operator annihilates the sequence:

$$P(r) \cdot \{y_t\} = 0.$$

Now, suppose $Q(r) \in \mathbb{R}[r]$ is another characteristic polynomial corresponding to a recurrence relation that also annihilates $\{y_t\}$, i.e.,

$$Q(r) \cdot \{y_t\} = 0.$$

We claim that $P(r) \mid Q(r)$. Assume for contradiction that $P(r) \nmid Q(r)$. Then we can write:

$$Q(r) = P(r)S(r) + R(r),$$

where $R(r) \neq 0$ is the non-zero remainder polynomial with $\deg(R) < \deg(P) = p$. Applying both sides to the sequence $\{y_t\}$, we obtain:

$$Q(r) \cdot \{y_t\} = [P(r)S(r) + R(r)] \cdot \{y_t\} = P(r)S(r) \cdot \{y_t\} + R(r) \cdot \{y_t\}.$$

The first term vanishes by assumption, since $P(r) \cdot \{y_t\} = 0$. Thus:

$$Q(r) \cdot \{y_t\} = R(r) \cdot \{y_t\} = 0.$$

This implies that $R(r)$ also annihilates $\{y_t\}$, contradicting the minimality of $P(r)$, as $\deg(R) < \deg(P)$. Therefore, no such non-zero $R(r)$ can exist.

We conclude that there exists $S(r) \in \mathbb{R}[r]$ such that

$$Q(r) = P(r)S(r),$$

and hence $P(r) \mid Q(r)$.  □

The result implies that any higher-order recurrence relation that annihilates a given sequence must preserve the characteristic roots of the minimal recurrence, since its characteristic polynomial contains the minimal one as a factor. Two notable special cases of such higher-order recurrences are:

(i) the $j$-step ahead regressor, where $y_{t+j} = a_1 y_t + \cdots + a_p y_{t-p+1}$; and

(ii) extended lookback models of the form $y_{t+1} = a_1 y_t + \cdots + a_L y_{t-L+1}$ with $L > p$.

Both formulations represent higher-order linear relations that are consistent with the original dynamics, and therefore preserve the same characteristic roots. Importantly, increasing the order of the recurrence (i.e., enlarging the polynomial degree) introduces additional degrees of freedom. This can enable the model to generalize better, especially in noisy or non-stationary settings by capturing a richer set of dependencies while still being anchored to the original root structure.

**Remark 4.** *In the noise-free setting, the closed-form solution to the optimization problem of* Equation (1) *can always achieve zero error, confirming that a higher-order recurrence relation holds. However, if one uses the Moore–Penrose pseudoinverse to express this solution, the result will be the solution with minimal $\ell_2$ norm. In general, this solution does not correspond to the minimal-order recurrence and may include unnecessary higher-order terms that preserve the roots but do not reflect the most compact representation of the underlying dynamics.*

### D.3 Scaling Property on White Noise Series

We give a formal version of Proposition 1 and its proof as follows:

**Proposition 6** (Asymptotic Distribution of $\text{vec}(\mathbf{W})$ under Cross-Covariance). *Let $\mathbf{X} \in \mathbb{R}^{N \times L}$ and $\mathbf{Y} \in \mathbb{R}^{N \times H}$ be random matrices with the following moment conditions:*

$$\mathbb{E}\left[\begin{pmatrix} \text{vec}(\mathbf{X}) \\ \text{vec}(\mathbf{Y}) \end{pmatrix}\right] = \mathbf{0}, \quad \mathbb{E}\left[\begin{pmatrix} \text{vec}(\mathbf{X}) \\ \text{vec}(\mathbf{Y}) \end{pmatrix}\begin{pmatrix} \text{vec}(\mathbf{X}) \\ \text{vec}(\mathbf{Y}) \end{pmatrix}^\top\right] = \begin{pmatrix} \mathbf{\Sigma}_\mathbf{X} & \mathbf{\Sigma}_\mathbf{XY} \\ \mathbf{\Sigma}_\mathbf{XY}^\top & \mathbf{\Sigma}_\mathbf{Y} \end{pmatrix},$$

*where $\mathbf{\Sigma}_\mathbf{X} \in \mathbb{R}^{NL \times NL}$, $\mathbf{\Sigma}_\mathbf{Y} \in \mathbb{R}^{NH \times NH}$, and $\mathbf{\Sigma}_\mathbf{XY} \in \mathbb{R}^{NL \times NH}$. To fit the linear model $\mathbf{Y} = \mathbf{XW} + \mathbf{E}$, where $\mathbf{W} \in \mathbb{R}^{L \times H}$ is the weight matrix, and $\mathbf{E} \in \mathbb{R}^{N \times H}$ is independent noise, then, for large $N$, the least-squares estimator $\hat{\mathbf{W}} = (\mathbf{X}^\top \mathbf{X})^{-1} \mathbf{X}^\top \mathbf{Y}$ satisfies*

$$\sqrt{N}\left(\text{vec}(\hat{\mathbf{W}}) - \text{vec}(\mathbf{W}_\star)\right) \xrightarrow{d} \mathcal{N}(\mathbf{0}, \mathbf{\Gamma}),$$

*where:*

- $\mathbf{W}_\star = \tilde{\mathbf{\Sigma}}_\mathbf{X}^{-1} \tilde{\mathbf{\Sigma}}_\mathbf{XY}$ *is the population coefficient matrix with $\tilde{\mathbf{\Sigma}}_\mathbf{X} = \frac{1}{N}\mathbb{E}[\mathbf{X}^\top \mathbf{X}]$ and $\tilde{\mathbf{\Sigma}}_\mathbf{XY} = \frac{1}{N}\mathbb{E}[\mathbf{X}^\top \mathbf{Y}]$.*

- $\mathbf{\Gamma} = \mathbf{\Sigma}_{res}^{(0)} \otimes \tilde{\mathbf{\Sigma}}_\mathbf{X}^{-1}$, *with $\mathbf{\Sigma}_{res} = \mathbf{\Sigma}_\mathbf{Y} - \mathbf{\Sigma}_\mathbf{YX} \mathbf{\Sigma}_\mathbf{X}^{-1} \mathbf{\Sigma}_\mathbf{XY} \approx \mathbf{\Sigma}_{res}^{(0)} \otimes \mathbf{I}_N$, $\otimes$ denotes the Kronecker product, and $\mathbf{I}_N$ denotes the identity matrix of size $N$.*

*Proof.* First, the population coefficient matrix $\mathbf{W}_\star$ minimizes the expected squared error:

$$\mathbf{W}_\star = \arg\min_{\mathbf{W}} \mathbb{E}\left[\|\mathbf{Y} - \mathbf{XW}\|_F^2\right] = \tilde{\mathbf{\Sigma}}_\mathbf{X}^{-1} \tilde{\mathbf{\Sigma}}_\mathbf{XY}.$$

Substituting $\mathbf{Y} = \mathbf{XW}_\star + \mathbf{E}_\star$, the least-squares estimator can be written as:

$$\hat{\mathbf{W}} = (\mathbf{X}^\top \mathbf{X})^{-1} \mathbf{X}^\top \mathbf{Y} = (\mathbf{X}^\top \mathbf{X})^{-1} \mathbf{X}^\top (\mathbf{XW}_\star + \mathbf{E}_\star) = \mathbf{W}_\star + (\mathbf{X}^\top \mathbf{X})^{-1} \mathbf{X}^\top \mathbf{E}_\star.$$

Thus we have

$$\hat{\mathbf{W}} - \mathbf{W}_\star = (\mathbf{X}^\top \mathbf{X})^{-1} \mathbf{X}^\top \mathbf{E}_\star.$$

Vectorizing both sides and using the identity $\text{vec}(\mathbf{ABC}) = (\mathbf{C}^\top \otimes \mathbf{A})\text{vec}(\mathbf{B})$ gives:

$$\text{vec}(\hat{\mathbf{W}} - \mathbf{W}_\star) = \left(\mathbf{I}_H \otimes (\mathbf{X}^\top \mathbf{X})^{-1}\right)\text{vec}(\mathbf{X}^\top \mathbf{E}_\star).$$

To derive the limiting distribution, we study the asymptotic properties of $\mathbf{X}^\top \mathbf{X}$ and $\text{vec}(\mathbf{X}^\top \mathbf{E}_\star)$. By the law of large numbers, we have:

$$\frac{1}{N}\mathbf{X}^\top \mathbf{X} \xrightarrow{p} \tilde{\mathbf{\Sigma}}_\mathbf{X} \quad \Rightarrow \quad (\mathbf{X}^\top \mathbf{X})^{-1} \approx \frac{1}{N}\tilde{\mathbf{\Sigma}}_\mathbf{X}^{-1}.$$

Multiplying this asymptotic approximation gives

$$\sqrt{N}\,\text{vec}(\hat{\mathbf{W}} - \mathbf{W}_\star) \approx [\mathbf{I}_H \otimes \tilde{\mathbf{\Sigma}}_{\mathbf{X}}^{-1}]\left(\frac{1}{\sqrt{N}}\,\text{vec}(\mathbf{X}^\top \mathbf{E}_\star)\right).$$

Next, we apply the central limit theorem to the matrix product $\mathbf{X}^\top \mathbf{E}_\star$, which gives:

$$\frac{1}{\sqrt{N}}\,\text{vec}(\mathbf{X}^\top \mathbf{E}_\star) \xrightarrow{d} \mathcal{N}(\mathbf{0}, \mathbf{\Sigma}_{\text{res}}^{(0)} \otimes \tilde{\mathbf{\Sigma}}_{\mathbf{X}}).$$

Now applying Slutsky's theorem gives:

$$\sqrt{N}\left(\text{vec}(\hat{\mathbf{W}}) - \text{vec}(\mathbf{W}_\star)\right) \xrightarrow{d} \mathcal{N}(\mathbf{0}, \mathbf{\Gamma}),$$

where

$$\mathbf{\Gamma} = (\mathbf{I}_H \otimes \tilde{\mathbf{\Sigma}}_{\mathbf{X}}^{-1})(\mathbf{\Sigma}_{\text{res}}^{(0)} \otimes \tilde{\mathbf{\Sigma}}_{\mathbf{X}})(\mathbf{I}_H \otimes \tilde{\mathbf{\Sigma}}_{\mathbf{X}}^{-1})^\top = \mathbf{\Sigma}_{\text{res}}^{(0)} \otimes \tilde{\mathbf{\Sigma}}_{\mathbf{X}}^{-1}.$$

$\square$

The informal claim of Proposition 1 that the weights learned by a linear model forecasting a zero-mean noise with finite second moments converge at a rate proportional to $\mathcal{O}(1/\sqrt{T})$, where $T$ is the length of the observed time series, can be formally justified by the asymptotic normality theorem for least-squares estimators. In this context, each observation in time can be treated as an independent sample, allowing the time horizon $T$ to play the role of the sample size $N$ in the multivariate analysis. The theorem establishes that, under suitable regularity conditions, the least-squares estimator $\hat{\mathbf{W}}$ converges in distribution to a normal random variable centered at the true coefficient matrix $\mathbf{W}_\star$, with convergence rate $\mathcal{O}(1/\sqrt{N})$.

When the time series data consist of a zero-mean noise with finite second moments, the assumptions of the theorem are particularly well-suited. The proposition provides a rigorous statistical foundation for the empirical observation that forecasting models trained on longer time series tend to yield more stable and accurate parameter estimates. However, the convergence rate of the estimator is sublinear—specifically, on the order of $\mathcal{O}(1/\sqrt{T})$—which implies that a disproportionately large amount of data is required to significantly reduce the estimation variance. This highlights an inherent inefficiency in learning from high-variance signals like white noise: although increasing the sample size improves estimation accuracy, the marginal gains diminish. Therefore, to overcome the adverse effects introduced by noise, a substantially large dataset is often necessary.

### D.4 EFFECT OF MODEL PARAMETER RANK REDUCTION

We give a formal version of Proposition 2 and its proof as follows:

**Proposition 7** (Model Parameter Rank Reduction). *Let $\mathbf{W} \in \mathbb{R}^{L \times H}$ be a matrix with $\text{rank}(\mathbf{W}) \leq \rho \leq \min(L, H)$. Then there exist matrices $\mathbf{U}_\rho \in \mathbb{R}^{L \times \rho}$, $\mathbf{\Sigma}_\rho \in \mathbb{R}^{\rho \times \rho}$, and $\mathbf{V}_\rho \in \mathbb{R}^{H \times \rho}$ such that*

$$\mathbf{W} = \mathbf{U}_\rho \mathbf{\Sigma}_\rho \mathbf{V}_\rho^\top.$$

*Consequently, for any $\mathbf{x} \in \mathbb{R}^L$ and $\mathbf{y} \in \mathbb{R}^H$, the model $\mathbf{y} = \mathbf{W}^\top \mathbf{x}$ can be equivalently written as*

$$\mathbf{V}_\rho^\top \mathbf{y} = \mathbf{\Sigma}_\rho \mathbf{U}_\rho^\top \mathbf{x} \quad or \quad \mathbf{y} = \mathbf{V}_\rho \mathbf{\Sigma}_\rho \mathbf{U}_\rho^\top \mathbf{x}.$$

*Proof.* Since $\text{rank}(\mathbf{W}) \leq \rho \leq \min(L, H)$, the singular value decomposition (SVD) yields orthogonal matrices $\mathbf{U} \in \mathbb{R}^{L \times L}$, $\mathbf{V} \in \mathbb{R}^{H \times H}$ and a diagonal matrix $\mathbf{\Sigma} \in \mathbb{R}^{L \times H}$ with at most $\rho$ non-zero singular values, such that

$$\mathbf{W} = \mathbf{U}\mathbf{\Sigma}\mathbf{V}^\top.$$

Let $\mathbf{U}_\rho \in \mathbb{R}^{L \times \rho}$ and $\mathbf{V}_\rho \in \mathbb{R}^{H \times \rho}$ consist of the first $\rho$ columns of $\mathbf{U}$ and $\mathbf{V}$, respectively, and let $\mathbf{\Sigma}_\rho \in \mathbb{R}^{\rho \times \rho}$ contain the top $\rho$ singular values. Then,

$$\mathbf{W} = \mathbf{U}_\rho \mathbf{\Sigma}_\rho \mathbf{V}_\rho^\top.$$

For any $\mathbf{x} \in \mathbb{R}^L$, the model $\mathbf{y} = \mathbf{W}^\top \mathbf{x}$ becomes

$$\mathbf{y} = \mathbf{V}_\rho \mathbf{\Sigma}_\rho \mathbf{U}_\rho^\top \mathbf{x}.$$

Left-multiplying by $\mathbf{V}_\rho^\top$ and using the orthogonality property $\mathbf{V}_\rho^\top \mathbf{V}_\rho = \mathbf{I}_\rho$ gives the equivalent form

$$\mathbf{V}_\rho^\top \mathbf{y} = \mathbf{\Sigma}_\rho \mathbf{U}_\rho^\top \mathbf{x},$$

completing the proof. $\qquad\square$

From the Proposition 7, we see that fitting a rank-$\rho$ linear map is equivalent to:

- projecting $\mathbf{x}$ into an $\rho$-dimensional latent space via $\mathbf{U}_\rho^\top$,

- projecting $\mathbf{y}$ into the same latent space via $\mathbf{V}_\rho^\top$, and

- learning a linear map $\mathbf{\Sigma}_\rho$ between them.

This expresses the model as:

$$\text{(projected output)} = \text{linear map} \circ \text{(projected input)}.$$

So, constraining the rank of $\mathbf{W}$ is equivalent to simultaneously reducing the dimensionality of both inputs and outputs, and learning the relationship in that shared latent space.

Proposition 7 shows a *constructive duality* between two operations in time series forecasting:

- imposing a rank-$\rho$ constraint on the model weight matrix $\mathbf{W}$, and

- explicitly projecting both the input and output data into an $\rho$-dimensional latent space, and then learning a full-rank linear map between them.

In this work, we prefer the model-based rank constraint than data projection for the following reasons:

- Directly projecting data breaks the sequential manifold structure of the time series, potentially harming downstream performance;

- Model-based regularization enables cross-validation to efficiently tune the rank hyperparameter, while data pre-processing fixes it a priori;

- We empirically tested both strategies: constraining the model rank consistently outperformed pre-projected data. Other related approaches (e.g., Huang et al. (2025)) also use low-rank projections of data but similarly underperform our method.

### D.5 ANALYSIS OF RANK REDUCTION

We analyze rank reduction in the linear mapping $\mathbf{Y} = \mathbf{X}\mathbf{W}$, where $\mathbf{X} \in \mathbb{R}^{N \times L}$ and $\mathbf{Y} \in \mathbb{R}^{N \times H}$ The focus is on how Reduced-Rank Regression (RRR) and Direct Weight RRR (DWRR) alter the solution in terms of the Frobenius norm.

**Proposition 8** (Frobenius Norm Bound for DWRR Approximation)**.** *The Frobenius norm of the approximation error satisfies:*

$$\|\mathbf{W}_{DWRR} - \mathbf{W}_{OLS}\|_F = \sqrt{\sum_{i=\rho+1}^{\min(L,H)} \sigma_i^2}.$$

*where $\sigma_1 \geq \sigma_2 \geq \cdots \geq \sigma_{\min(L,H)} \geq 0$ are the singular values of $\mathbf{W}_{OLS}$.*

*Moreover, this is the best possible rank-$\rho$ approximation error in the Frobenius norm, meaning that for any rank-$\rho$ matrix $\mathbf{B}$,*

$$\|\mathbf{W}_{DWRR} - \mathbf{W}_{OLS}\|_F \leq \|\mathbf{B} - \mathbf{W}_{OLS}\|_F.$$

*Proof.* Let the full SVD of $\mathbf{W}_{\text{OLS}} \in \mathbb{R}^{L \times H}$ be

$$\mathbf{W}_{\text{OLS}} = \sum_{i=1}^{\min(L,H)} \sigma_i \mathbf{u}_i \mathbf{v}_i^\top,$$

where $\sigma_1 \geq \sigma_2 \geq \cdots \geq \sigma_{\min(L,H)} \geq 0$ are the singular values, and $\mathbf{u}_i$, $\mathbf{v}_i$ are the corresponding singular vectors. The rank-$\rho$ approximation $\mathbf{W}_{\text{DWRR}}$ retains the first $\rho$ terms:

$$\mathbf{W}_{\text{DWRR}} = \sum_{i=1}^{\rho} \sigma_i \mathbf{u}_i \mathbf{v}_i^\top.$$

The approximation error is then:

$$\mathbf{W}_{\text{DWRR}} - \mathbf{W}_{\text{OLS}} = - \sum_{i=\rho+1}^{\min(L,H)} \sigma_i \mathbf{u}_i \mathbf{v}_i^\top.$$

The Frobenius norm is the square root of the sum of the squared singular values. Hence,

$$\|\mathbf{W}_{\text{DWRR}} - \mathbf{W}_{\text{OLS}}\|_F = \sqrt{\sum_{i=\rho+1}^{\min(L,H)} \sigma_i^2}.$$

By the Eckart–Young–Mirsky theorem, the truncated SVD provides the best rank-$\rho$ approximation in Frobenius norm. That is, for any rank-$\rho$ matrix $\mathbf{B}$,

$$\|\mathbf{W}_{\text{DWRR}} - \mathbf{W}_{\text{OLS}}\|_F \leq \|\mathbf{B} - \mathbf{W}_{\text{OLS}}\|_F.$$

Therefore, $\mathbf{W}_{\text{DWRR}}$ is the optimal rank-$\rho$ approximation of $\mathbf{W}_{\text{OLS}}$ in the Frobenius norm.

$\square$

Similarly, we can derive analogous results for Reduced-Rank Regression (RRR).

**Proposition 9** (Frobenius Norm Bound for RRR vs. OLS)**.** *Assume $\mathbf{X} \in \mathbb{R}^{N \times L}$ has full column rank (i.e., $\operatorname{rank}(\mathbf{X}) = L$). Then, the Frobenius norm of the difference between $\mathbf{W}_{RRR}$ and $\mathbf{W}_{OLS}$ is bounded by:*

$$\|\mathbf{W}_{\text{RRR}} - \mathbf{W}_{\text{OLS}}\|_F \leq \frac{\sqrt{L}}{\sigma_{\min}(\mathbf{X})} \sqrt{\sum_{i=\rho+1}^{\min(N,H)} \sigma_i^2(\hat{\mathbf{Y}})},$$

*where $\sigma_{\min}(\mathbf{X})$ is the smallest singular value of $\mathbf{X}$, and $\sigma_i(\hat{\mathbf{Y}})$ are the singular values of $\hat{\mathbf{Y}} = \mathbf{X}\mathbf{W}_{OLS} \in \mathbb{R}^{N \times H}$.*

*Proof.* The OLS solution is

$$\mathbf{W}_{\text{OLS}} = (\mathbf{X}^\top\mathbf{X})^{-1}\mathbf{X}^\top\mathbf{X}\mathbf{W}_{\text{OLS}} = (\mathbf{X}^\top\mathbf{X})^{-1}\mathbf{X}^\top\hat{\mathbf{Y}} = (\mathbf{X}^\top\mathbf{X})^{-1}\mathbf{X}^\top\mathbf{U}\boldsymbol{\Sigma}\mathbf{V}^\top.$$

where $\hat{\mathbf{Y}} = \mathbf{U}\boldsymbol{\Sigma}\mathbf{V}^\top$ is the SVD of $\hat{\mathbf{Y}}$. RRR applies a rank constraint by projecting the solution onto the top $\rho$ right singular vectors of $\hat{\mathbf{Y}}$ (i.e. top $\rho$ singular vectors in $\mathbf{V}$), resulting in

$$\begin{aligned}
\mathbf{W}_{\text{RRR}} = \mathbf{W}_{\text{OLS}}\mathbf{V}_\rho\mathbf{V}_\rho^\top &= (\mathbf{X}^\top\mathbf{X})^{-1}\mathbf{X}^\top\mathbf{U}\boldsymbol{\Sigma}\mathbf{V}^\top\mathbf{V}_\rho\mathbf{V}_\rho^\top \\
&= (\mathbf{X}^\top\mathbf{X})^{-1}\mathbf{X}^\top \begin{pmatrix} \mathbf{U}_\rho & \mathbf{U}_{-\rho} \end{pmatrix} \begin{pmatrix} \boldsymbol{\Sigma}_\rho & \mathbf{0} \\ \mathbf{0} & \boldsymbol{\Sigma}_{-\rho} \end{pmatrix} \begin{pmatrix} \mathbf{V}_\rho^\top \\ \mathbf{V}_{-\rho}^\top \end{pmatrix} \mathbf{V}_\rho\mathbf{V}_\rho^\top \\
&= (\mathbf{X}^\top\mathbf{X})^{-1}\mathbf{X}^\top\mathbf{U}_\rho\boldsymbol{\Sigma}_\rho\mathbf{V}_\rho^\top,
\end{aligned}$$

where $\mathbf{U}_\rho \in \mathbb{R}^{N \times \rho}, \mathbf{V}_\rho \in \mathbb{R}^{H \times \rho}$ are the first $\rho$ columns of $\mathbf{U}$ and $\mathbf{V}$, respectively, $\mathbf{U}_{-\rho} \in \mathbb{R}^{N \times (L-\rho)}, \mathbf{V}_{-\rho} \in \mathbb{R}^{H \times (H-\rho)}$ are the remaining columns, $\boldsymbol{\Sigma}_\rho$ is the truncated diagonal matrix with the top $\rho$ singular values, and $\boldsymbol{\Sigma}_{-\rho}$ is the diagonal matrix with the remaining singular values.

The difference between the estimators is

$$\mathbf{W}_{\mathrm{RRR}} - \mathbf{W}_{\mathrm{OLS}} = (\mathbf{X}^\top \mathbf{X})^{-1}\mathbf{X}^\top(\mathbf{U}_\rho \mathbf{\Sigma}_\rho \mathbf{V}_\rho^\top - \mathbf{U}\mathbf{\Sigma}\mathbf{V}^\top)$$

$$= (\mathbf{X}^\top \mathbf{X})^{-1}\mathbf{X}^\top\left[\mathbf{U}_\rho \mathbf{\Sigma}_\rho \mathbf{V}_\rho^\top - (\mathbf{U}_\rho \quad \mathbf{U}_{-\rho})\begin{pmatrix}\mathbf{\Sigma}_\rho & \mathbf{0} \\ \mathbf{0} & \mathbf{\Sigma}_{-\rho}\end{pmatrix}\begin{pmatrix}\mathbf{V}_\rho^\top \\ \mathbf{V}_{-\rho}^\top\end{pmatrix}\right]$$

$$= -(\mathbf{X}^\top \mathbf{X})^{-1}\mathbf{X}^\top\mathbf{U}_{-\rho}\mathbf{\Sigma}_{-\rho}\mathbf{V}_{-\rho}^\top,$$

where $\mathbf{\Sigma}_{-\rho}$ retains only the lower $\min(N, H) - \rho$ singular values.

Using the submultiplicative property of the Frobenius norm and the orthogonality of $\mathbf{U}$ and $\mathbf{V}$, we have:

$$\|\mathbf{W}_{\mathrm{RRR}} - \mathbf{W}_{\mathrm{OLS}}\|_F \leq \|(\mathbf{X}^\top \mathbf{X})^{-1}\mathbf{X}^\top\|_F \cdot \|\mathbf{\Sigma}_{-\rho}\|_F.$$

Since

$$\|\mathbf{\Sigma}_{-\rho}\|_F = \sqrt{\sum_{i=\rho+1}^{\min(N,H)}\sigma_i^2(\hat{\mathbf{Y}})},$$

and

$$\|(\mathbf{X}^\top \mathbf{X})^{-1}\mathbf{X}^\top\|_F^2 = \mathrm{trace}(\mathbf{X}(\mathbf{X}^\top \mathbf{X})^{-2}\mathbf{X}^\top) \leq \frac{L}{\sigma_{\min}^2(\mathbf{X})},$$

we conclude that

$$\|\mathbf{W}_{\mathrm{RRR}} - \mathbf{W}_{\mathrm{OLS}}\|_F \leq \frac{\sqrt{L}}{\sigma_{\min}(\mathbf{X})} \cdot \sqrt{\sum_{i=\rho+1}^{\min(N,H)}\sigma_i^2(\hat{\mathbf{Y}})}.$$

$\square$

The Frobenius norm bound shows that when the discarded singular values are small, the RRR solution remains close to the OLS estimator. Thus, RRR can effectively suppress noise while preserving most of the signal, especially when the true underlying dynamics is of low rank.

## D.6 RANK REDUCTION AND NOISE ROBUSTNESS

Rank reduction (via truncated singular value decomposition or spectral projection) is a widely used and effective heuristic for denoising high-dimensional data. Intuitively, when the signal lives in a low-dimensional subspace and the noise is (approximately) isotropic or spread across many directions, projecting the noisy data onto the dominant rank-$\rho$ subspace "concentrates" signal energy while discarding much of the high-dimensional noise.

Below we formalize two complementary theoretical explanations for why rank reduction reduces noise:

- **Result I (Matrix Denoising).** Treat the (symmetric) data / weight matrix itself as the object to denoise: if $\mathbf{M}^\star$ is low-rank and we observe $\mathbf{M} = \mathbf{M}^\star + \mathbf{E}$, then the rank-$\rho$ spectral truncation of $\mathbf{M}$ approximates $\mathbf{M}^\star$ with error controlled by the noise spectral norm $\|\mathbf{E}\|_2$ [6].

- **Result II (Linear Factor Model).** When each observation arises from a linear low-rank model $\mathbf{y_i} = \mathbf{L}^\star\mathbf{x_i} + \epsilon_\mathbf{i}$ (latent factors $\mathbf{x_i}$ and idiosyncratic noise $\epsilon_\mathbf{i}$)[7], the sample covariance's top-$\rho$ eigenspace estimates the population principal subspace with an error that decreases with sample size $n$; projecting observations onto this estimated low-rank subspace reduces variance of the fitted signals and yields improved reconstruction.

The statements below make these ideas precise and give high-probability error bounds.

---

[6]Note that $\|\cdot\|_2$ with a *matrix* inside is known as the spectral norm of a matrix. Specifically, for some matrix $\mathbf{M}$, $\|\mathbf{M}\|_2 = \sigma_{\max}(\mathbf{M})$, where $\sigma_{\max}(\mathbf{M})$ is the maximum singular value of $\mathbf{M}$.

[7]This classical linear factor model should not be confused with our linear forecast model; it aims at recovering a low-rank factor structure from independent samples, whereas ours focuses on forecasting future values from past observations.

**Proposition 10** (Low-rank Matrix Estimation Error). *Let $\mathbf{M}^\star \in \mathbb{R}^{n \times n}$ be a rank-$\rho$ symmetric matrix with eigen-decomposition*

$$\mathbf{M}^\star = \mathbf{U}^\star \mathbf{\Lambda}^\star \mathbf{U}^{\star\top},$$

*where $\mathbf{U}^\star \in \mathbb{R}^{n \times \rho}$ has orthonormal columns, and $\mathbf{\Lambda}^\star = \mathrm{diag}(\lambda_1^\star, \ldots, \lambda_\rho^\star)$ with $|\lambda_1^\star| \geq \cdots \geq |\lambda_\rho^\star| > 0$. Suppose we observe*

$$\mathbf{M} = \mathbf{M}^\star + \mathbf{E},$$

*where $\mathbf{E}$ is a symmetric noise matrix with entries $\{E_{ij}\}_{i \geq j}$ independently distributed as $E_{ij} \overset{i.i.d.}{\sim} N(0, \sigma^2)$. Let $\mathbf{U} = [\mathbf{u_1}, \ldots, \mathbf{u_\rho}]$ and $\mathbf{\Lambda} = \mathrm{diag}(\lambda_1, \ldots, \lambda_\rho)$ be the top-$\rho$ eigenvectors and eigenvalues of $\mathbf{M}$ (in descending order of absolute value), and define*

$$\widehat{\mathbf{M}} := \mathbf{U}\mathbf{\Lambda}\mathbf{U}^\top.$$

*If the noise level satisfies*

$$\sigma\sqrt{n} \; \leq \; \frac{1 - \frac{1}{\sqrt{2}}}{5} \, |\lambda_\rho^\star|,$$

*then with probability at least $1 - O(n^{-8})$ we have:*

$$\|\widehat{\mathbf{M}} - \mathbf{M}^\star\|_2 \leq 10\,\sigma\sqrt{n}, \quad \|\widehat{\mathbf{M}} - \mathbf{M}^\star\|_F \leq 10\,\sigma\sqrt{2\rho n}.$$

*Sketch of the proof.* We outline the main steps of the proof.

**Step 1: Truncating the noise matrix.** Since the Gaussian entries of the noise matrix $\mathbf{E}$ are unbounded, we introduce a truncated version

$$\widetilde{E}_{ij} := E_{ij} \, \mathbf{1}\big\{|E_{ij}| \leq 5\sigma\sqrt{\log n}\big\}, \quad 1 \leq i, j \leq n.$$

By Gaussian tail bounds and the union bound,

$$\mathbb{P}\{\mathbf{E} = \widetilde{\mathbf{E}}\} \geq 1 - n^{-10}.$$

Hence, with overwhelming probability, we may replace $\mathbf{E}$ by $\widetilde{\mathbf{E}}$.

**Step 2: Bounding the spectral norm of the noise.** All entries of $\widetilde{\mathbf{E}}$ satisfy $|\widetilde{E}_{ij}| \leq B := 5\sigma\sqrt{\log n}$ and $\mathbb{E}[\widetilde{E}_{ij}^2] \leq \sigma^2$. A matrix Bernstein/Wigner-type inequality (Wainwright, 2019) shows that

$$\|\widetilde{\mathbf{E}}\|_2 \; \leq \; 4\sigma\sqrt{n} + C_1 B \log n$$

with high probability. For sufficiently large $n$, the second term is negligible, leading to

$$\|\mathbf{E}\|_2 = \|\widetilde{\mathbf{E}}\|_2 \; \leq \; 5\sigma\sqrt{n}$$

with high probability.

**Step 3: Controlling the subspace estimation error.** Since the target matrix $\mathbf{M}^\star$ has rank $\rho$ with eigenvalue gap $|\lambda_\rho^\star|$, then under the signal-to-noise condition $\|\mathbf{E}\|_2 \leq (1 - 1/\sqrt{2})|\lambda_\rho^\star|$, the Davis–Kahan theorem (Davis and Kahan, 1970) gives

$$\mathrm{dist}(\mathbf{U}, \mathbf{U}^\star) \; \leq \; \frac{2\|\mathbf{E}\|_2}{|\lambda_\rho^\star|}.$$

**Step 4: Bounds on eigenvalues and low-rank reconstruction.** Weyl's inequality (Weyl, 1912) implies that all spurious eigenvalues beyond rank $\rho$ are bounded by $\|\mathbf{E}\|_2$. For the low-rank reconstruction $\widehat{\mathbf{M}} := \mathbf{U}\mathbf{\Lambda}\mathbf{U}^\top$, the triangle inequality yields

$$\|\widehat{\mathbf{M}} - \mathbf{M}^\star\|_2 \; \leq \; 2\|\mathbf{E}\|_2.$$

Since the difference has rank at most $2\rho$, we further obtain

$$\|\widehat{\mathbf{M}} - \mathbf{M}^\star\|_F \; \leq \; 2\sqrt{2\rho}\,\|\mathbf{E}\|_2.$$

$\square$

The above result also extends to rectangular (asymmetric) matrices with independent entries via the standard *symmetric dilation trick* (Tropp et al., 2015). Specifically, for $\mathbf{X} \in \mathbb{R}^{n_1 \times n_2}$, define its symmetric dilation

$$S(\mathbf{X}) := \begin{bmatrix} 0 & \mathbf{X} \\ \mathbf{X}^\top & 0 \end{bmatrix} \in \mathbb{R}^{(n_1+n_2) \times (n_1+n_2)},$$

which satisfies $\|\mathbf{X}\|_2 = \|S(\mathbf{X})\|_2$. Applying the theorem to $S(\mathbf{X})$ yields the same type of bounds for $\mathbf{X}$ itself. For conciseness, we may sometimes apply the theorem directly to asymmetric matrices without explicitly invoking the dilation.

**Proposition 11** (Estimation Error of the Principal Subspace). *Suppose we observe $n$ independent samples $\{\mathbf{y_i}\}_{i=1}^n \subset \mathbb{R}^p$ obeying*

$$\mathbf{y_i} = \mathbf{L}^\star \mathbf{x_i} + \epsilon_\mathbf{i}, \qquad 1 \le i \le n,$$

*where $\mathbf{x_i} \overset{i.i.d.}{\sim} \mathcal{N}(0, \mathbf{I}_\rho)$ are latent factors, $\epsilon_\mathbf{i} \overset{i.i.d.}{\sim} \mathcal{N}(0, \sigma^2 \mathbf{I_p})$ are idiosyncratic noises, and $\mathbf{L}^\star = \mathbf{U}^\star (\mathbf{\Lambda}^\star)^{1/2}$. Here $\mathbf{U}^\star \in \mathbb{R}^{p \times \rho}$ has orthonormal columns and $\mathbf{\Lambda}^\star = \mathrm{diag}(\lambda_1^\star, \ldots, \lambda_\rho^\star)$ satisfies $\lambda_1^\star \ge \cdots \ge \lambda_\rho^\star > 0$. Define the condition number $\kappa := \lambda_1^\star / \lambda_\rho^\star$.*

*Let the sample covariance matrix be*

$$\mathbf{M} := \frac{1}{n} \sum_{i=1}^n \mathbf{y_i} \mathbf{y_i}^\top,$$

*and denote by $\lambda_1 \ge \cdots \ge \lambda_\rho$ its top-$\rho$ eigenvalues with associated eigenvectors $\mathbf{u_1}, \ldots, \mathbf{u}_\rho$. Let $\mathbf{U} = [\mathbf{u_1}, \ldots, \mathbf{u}_\rho] \in \mathbb{R}^{p \times \rho}$ be the spectral estimator of the principal subspace $\mathbf{U}^\star$.*

*If the sample size satisfies*

$$n \ge C \left( \kappa^2 \rho + \rho \log^2(n+p) + \frac{\kappa \sigma^2 p}{\lambda_\rho^\star} + \frac{\sigma^4 p}{(\lambda_\rho^\star)^2} \right) \log^3(n+p)$$

*for some sufficiently large constant $C > 0$, then with probability at least $1 - O((n+p)^{-10})$, one has*

$$\mathrm{dist}(\mathbf{U}, \mathbf{U}^\star) \lesssim \left( \frac{\sigma}{\sqrt{\lambda_\rho^\star}} \sqrt{\frac{\kappa p}{n}} + \frac{\sigma^2}{\lambda_\rho^\star} \sqrt{\frac{p}{n}} + \kappa \sqrt{\frac{\rho}{n}} \right) \log^{1/2}(n+p),$$

*where*

$$\mathrm{dist}(\mathbf{U}, \mathbf{U}^\star) := \min_{R \in \mathcal{O}_{\rho \times \rho}} \|\mathbf{U}R - \mathbf{U}^\star\|.$$

Please refer to Chen et al. (2021) for the proof of the Proposition 11.

Note that rank reduction approach, or truncated singular value decomposition (TSVD) corresponds to a "hard-thresholding" operation on the singular values: retaining the top $\rho$ singular values and setting the rest to zero. A more desirable approach is to perform singular value shrinkage, that is, to apply a nonlinear function $\eta(\sigma_i)$ to each singular value $\sigma_i$, yielding the estimator

$$\hat{\mathbf{X}}_\eta = \sum_i \eta(\sigma_i) \mathbf{u_i} \mathbf{v_i}^\top.$$

This compensates for the "inflation" effect of noise on the singular values and thus improves estimation accuracy. Within an asymptotic framework provided by random matrix theory (Tao, 2012) (as $m, n \to \infty$ with aspect ratio $\beta$ fixed), the authors (Gavish and Donoho, 2017) show that for a given loss function (such as the Frobenius-norm MSE), there exists a unique asymptotically optimal singular value shrinkage function whose asymptotic risk is no worse than that of any other shrinkage rule across all low-rank models. This implies that in this framework, once the loss function is specified, the optimal form of singular value shrinkage is uniquely and rationally determined.

### D.7 ANALYSIS OF ROOT PURGE

When $\mathbf{X}$ and $\mathbf{Y}$ share the same dimensions (i.e., $\mathbf{X}, \mathbf{Y} \in \mathbb{R}^{N \times L}$), the modified objective function of Root Purge:

$$J(\mathbf{W}) = \|\mathbf{Y} - \mathbf{X}\mathbf{W}\|_F^2 + \lambda \|(\mathbf{Y} - \mathbf{X}\mathbf{W})\mathbf{W}\|_F^2 = \|\mathbf{Y} - \mathbf{X}\mathbf{W}\|_F^2 + \lambda \|\mathbf{R}\mathbf{W}\|_F^2,$$

admits a special structure. Here, $\mathbf{W} \in \mathbb{R}^{L \times L}$ is now a square matrix. In practice, the optimization of $J(\mathbf{W})$ can also be carried out using a stop-gradient strategy, where the residual in the root purge term $\mathbf{R} = \mathbf{Y} - \mathbf{X}\mathbf{W}$ is treated as constant when updating $\mathbf{W}$. Since the stop-gradient case tends to be easier, we mainly analyze the full-gradient version of root purge and give conclusions of stop-gradient version in the remarks.

**Proposition 12** (Equivalence Condition). *The ordinary least squares solution* $\mathbf{W}_{OLS} = (\mathbf{X}^\top \mathbf{X})^{-1}\mathbf{X}^\top \mathbf{Y}$ *is a critical point of the modified objective* $J(\mathbf{W})$ *if and only if the residuals* $\mathbf{E} = \mathbf{Y} - \mathbf{X}\mathbf{W}_{OLS}$ *satisfy:*

$$\mathbf{E}^\top \mathbf{E} \mathbf{W}_{OLS} = \mathbf{0}.$$

*Proof.* The gradient of $J(\mathbf{W})$ with respect to $\mathbf{W}$ is given by:

$$\nabla_{\mathbf{W}} J(\mathbf{W}) = -2\mathbf{X}^\top \mathbf{R} + 2\lambda(\mathbf{R}^\top \mathbf{R}\mathbf{W} - \mathbf{X}^\top \mathbf{R}\mathbf{W}\mathbf{W}^\top).$$

where $\mathbf{R} = \mathbf{Y} - \mathbf{X}\mathbf{W}$ denotes the residue. Consider the least squares solution $\mathbf{W}_{\text{OLS}} = (\mathbf{X}^\top \mathbf{X})^{-1}\mathbf{X}^\top \mathbf{Y}$, and define the corresponding residual as $\mathbf{E} = \mathbf{Y} - \mathbf{X}\mathbf{W}_{\text{OLS}}$. Since $\mathbf{W}_{\text{OLS}}$ minimizes the squared loss $\|\mathbf{Y} - \mathbf{X}\mathbf{W}\|_F^2$, the residual is orthogonal to the column space of $\mathbf{X}$, which implies:

$$\mathbf{X}^\top \mathbf{E} = \mathbf{0}.$$

Substituting $\mathbf{W} = \mathbf{W}_{\text{OLS}}$ into the gradient expression yields:

$$\nabla_{\mathbf{W}} J(\mathbf{W}_{\text{OLS}}) = -2\mathbf{X}^\top \mathbf{E} + 2\lambda \mathbf{E}^\top \mathbf{E}\mathbf{W}_{\text{OLS}} - 2\lambda \mathbf{X}^\top \mathbf{E}\mathbf{W}_{\text{OLS}}\mathbf{W}_{\text{OLS}}^\top = 2\lambda \mathbf{E}^\top \mathbf{E}\mathbf{W}_{\text{OLS}}.$$

Therefore, $\nabla_{\mathbf{W}} J(\mathbf{W}_{\text{OLS}}) = \mathbf{0}$ if and only if

$$\mathbf{E}^\top \mathbf{E} \, \mathbf{W}_{\text{OLS}} = \mathbf{0}.$$

$\square$

This proposition highlights a key geometric insight about when the ordinary least squares (OLS) solution remains optimal under a modified loss that penalizes residuals in the direction of the solution itself. Specifically, the condition $\mathbf{E}^\top \mathbf{E}\mathbf{W}_{\text{OLS}} = \mathbf{0}$ implies that $\mathbf{W}_{\text{OLS}}$ must lie in the null space of the residual matrix $\mathbf{E}$. In other words, the prediction directions encoded in $\mathbf{W}_{\text{OLS}}$ must be orthogonal to the residual errors. This condition ensures that the additional penalty term, which measures how much the residuals are aligned with the predictions, has no effect at $\mathbf{W}_{\text{OLS}}$. Geometrically, it means that the model's learned directions do not amplify or interact with the error left over after fitting, preserving OLS as a critical point of the new objective. This equivalence gives a deeper understanding of how structured residuals or model alignment can impact regularized learning. Moreover, we have the following result:

**Proposition 13** (Bounded Deviation of Root Purge). *For the minimizer* $\mathbf{W}^*$ *of* $J(\mathbf{W})$[8]*, its deviation from* $\mathbf{W}_{OLS}$ *is bounded by:*

$$\|\mathbf{W}^* - \mathbf{W}_{OLS}\|_F \leq \frac{\|\mathbf{E}^\top \mathbf{E}\mathbf{W}_{OLS}\|_F}{\sigma_{\min}(\mathbf{X}^\top \mathbf{X})(1/\lambda + \sigma_{\min}^2(\mathbf{W}_{OLS})) + \sigma_{\min}(\mathbf{E}^\top \mathbf{E})},$$

*where* $\sigma_{\min}(\cdot)$ *denotes the smallest singular value of a matrix.*

---

[8]Our analysis focuses on the local behavior around $\mathbf{W}_{\text{OLS}}$, as it preserves the most informative dynamics of the system. This restricts $\mathbf{W}^*$ to a neighborhood of $\mathbf{W}_{\text{OLS}}$, allowing us to neglect higher-order terms in $\boldsymbol{\Delta}$. If it deviates too far from this neighborhood, the root-seeking term may incur a large loss, potentially compromising the system's integrity.

*Proof.* The gradient of $J(\mathbf{W})$ with respect to $\mathbf{W}$ is given by:

$$\nabla_{\mathbf{W}} J(\mathbf{W}) = -2\mathbf{X}^\top \mathbf{R} + 2\lambda(\mathbf{R}^\top \mathbf{R}\mathbf{W} - \mathbf{X}^\top \mathbf{R}\mathbf{W}\mathbf{W}^\top).$$

where $\mathbf{R} = \mathbf{Y} - \mathbf{X}\mathbf{W}$ denotes the residue. Setting $\nabla_{\mathbf{W}} J(\mathbf{W}^*) = 0$ gives:

$$-\mathbf{X}^\top \mathbf{R}^* + \lambda(\mathbf{R}^{*\top}\mathbf{R}^*\mathbf{W}^* - \mathbf{X}^\top \mathbf{R}^*\mathbf{W}^*\mathbf{W}^{*\top}) = \mathbf{0}.$$

Then from $\mathbf{E} = \mathbf{Y} - \mathbf{X}\mathbf{W}_{\text{OLS}}$, we have $\mathbf{R}^* = \mathbf{Y} - \mathbf{X}\mathbf{W}^* = \mathbf{E} + \mathbf{X}\mathbf{W}_{\text{OLS}} - \mathbf{X}\mathbf{W}^* = \mathbf{E} - \mathbf{X}(\mathbf{W}^* - \mathbf{W}_{\text{OLS}})$. Substituting this into the above equation gives:

$$\begin{aligned}
-\,&\mathbf{X}^\top \left(\mathbf{E} - \mathbf{X}(\mathbf{W}^* - \mathbf{W}_{\text{OLS}})\right) \\
&+ \lambda \left[ \left(\mathbf{E} - \mathbf{X}(\mathbf{W}^* - \mathbf{W}_{\text{OLS}})\right)^\top \left(\mathbf{E} - \mathbf{X}(\mathbf{W}^* - \mathbf{W}_{\text{OLS}})\right)\mathbf{W}^* \right. \\
&\left. - \mathbf{X}^\top \left(\mathbf{E} - \mathbf{X}(\mathbf{W}^* - \mathbf{W}_{\text{OLS}})\right)\mathbf{W}^*\mathbf{W}^{*\top} \right] = \mathbf{0},
\end{aligned}$$

Applying $\mathbf{X}^\top \mathbf{E} = \mathbf{0}$ and dropping the higher-order terms gives:

$$\mathbf{X}^\top \mathbf{X}\boldsymbol{\Delta} + \lambda \mathbf{E}^\top \mathbf{E}\mathbf{W}_{\text{OLS}} + \lambda \mathbf{E}^\top \mathbf{E}\boldsymbol{\Delta} + \lambda \mathbf{X}^\top \mathbf{X}\boldsymbol{\Delta}\mathbf{W}_{\text{OLS}}\mathbf{W}_{\text{OLS}}^\top = \mathbf{0},$$

where $\boldsymbol{\Delta} = \mathbf{W}^* - \mathbf{W}_{\text{OLS}}$. Vectorize this equation using $\text{vec}(\cdot)$ and the Kronecker product $\otimes$:

$$\left[(\mathbf{I} + \lambda \mathbf{W}_{\text{OLS}}\mathbf{W}_{\text{OLS}}^\top) \otimes \mathbf{X}^\top \mathbf{X} + \lambda \mathbf{I} \otimes \mathbf{E}^\top \mathbf{E}\right] \text{vec}(\boldsymbol{\Delta}) = -\lambda \text{vec}(\mathbf{E}^\top \mathbf{E}\mathbf{W}_{\text{OLS}}).$$

Let $\mathbf{A} = (\mathbf{I} + \lambda \mathbf{W}_{\text{OLS}}\mathbf{W}_{\text{OLS}}^\top) \otimes \mathbf{X}^\top \mathbf{X} + \lambda \mathbf{I} \otimes \mathbf{E}^\top \mathbf{E}$. Then we have:

$$\|\boldsymbol{\Delta}\|_F = \|\text{vec}(\boldsymbol{\Delta})\|_2 \le \|\mathbf{A}^{-1}\|_2 \|\text{vec}(\lambda \mathbf{E}^\top \mathbf{E}\mathbf{W}_{\text{OLS}})\|_2 = \lambda \|\mathbf{A}^{-1}\|_2 \|\mathbf{E}^\top \mathbf{E}\mathbf{W}_{\text{OLS}}\|_F \tag{10}$$

The singular values satisfy:

$$\begin{aligned}
\sigma_{\min}(\mathbf{A}) &\ge \sigma_{\min}(\mathbf{X}^\top \mathbf{X})\sigma_{\min}(\mathbf{I} + \lambda \mathbf{W}_{\text{OLS}}\mathbf{W}_{\text{OLS}}^\top) + \lambda \sigma_{\min}(\mathbf{E}^\top \mathbf{E}) \\
&= \sigma_{\min}(\mathbf{X}^\top \mathbf{X})(1 + \lambda \sigma_{\min}^2(\mathbf{W}_{\text{OLS}})) + \lambda \sigma_{\min}(\mathbf{E}^\top \mathbf{E})
\end{aligned}$$

where we used $\sigma_{\min}(\mathbf{I} + \lambda \mathbf{W}_{\text{OLS}}\mathbf{W}_{\text{OLS}}^\top) = 1 + \lambda \sigma_{\min}^2(\mathbf{W}_{\text{OLS}})$. Thus we have:

$$\|\mathbf{A}^{-1}\|_2 \le \frac{1}{\sigma_{\min}(\mathbf{X}^\top \mathbf{X})(1 + \lambda \sigma_{\min}^2(\mathbf{W}_{\text{OLS}})) + \lambda \sigma_{\min}(\mathbf{E}^\top \mathbf{E})}.$$

Substituting this into Equation (10) yields the final bound:

$$\|\mathbf{W}^* - \mathbf{W}_{\text{OLS}}\|_F \le \frac{\|\mathbf{E}^\top \mathbf{E}\mathbf{W}_{\text{OLS}}\|_F}{\sigma_{\min}(\mathbf{X}^\top \mathbf{X})(1/\lambda + \sigma_{\min}^2(\mathbf{W}_{\text{OLS}})) + \sigma_{\min}(\mathbf{E}^\top \mathbf{E})}.$$

$\square$

This proposition provides a meaningful upper bound on the deviation between the minimizer $\mathbf{W}^*$ of the modified objective $J(\mathbf{W})$ and the ordinary least squares (OLS) solution $\mathbf{W}_{\text{OLS}}$. Several important insights emerge from this result:

1. **Bounded Deviation:**
   This bound quantifies how much $\mathbf{W}^*$ can move away from $\mathbf{W}_{\text{OLS}}$ due to the influence of noise and regularization. A smaller $\lambda$, larger singular values (i.e., better-conditioned matrices), or well-structured $\mathbf{W}_{\text{OLS}}$ all help tighten the bound.

2. **Bias vs. Robustness Trade-off:**
   Although $\mathbf{W}^*$ may not be an unbiased estimator—as it deviates from the OLS solution—the bound shows that this deviation is explicitly controlled by the regularization parameter $\lambda$, the norm of $\mathbf{W}_{\text{OLS}}$, and the spectral properties of $\mathbf{X}^\top \mathbf{X}$ and $\mathbf{E}^\top \mathbf{E}$. This introduces robustness: rather than fitting noise in the data (as OLS can), the solution remains stable under perturbations, especially when the noise matrix $\mathbf{E}$ is significant.

3. **Optimization Warm Start:**
   Since the bound suggests that $\mathbf{W}^*$ lies close to $\mathbf{W}_{\text{OLS}}$, a practical implication is to initialize iterative optimization algorithms at $\mathbf{W}_{\text{OLS}}$. This warm start can lead to faster convergence by starting closer to the final solution.

In summary, while the estimator $\mathbf{W}^*$ is biased relative to $\mathbf{W}_{\mathrm{OLS}}$, the trade-off results in enhanced robustness to noise, controlled deviation, and practical optimization benefits, making this formulation useful in settings where noise and overfitting are concerns. Combining Proposition 8, Proposition 9, and Proposition 13 with the triangle inequality allows us to bound the Frobenius norm errors $\|\mathbf{W}^* - \mathbf{W}_{\mathrm{RRR}}\|_F$ and $\|\mathbf{W}^* - \mathbf{W}_{\mathrm{DWRR}}\|_F$, suggesting that Root Purge and rank reduction techniques can have similar effects on controlling model complexity and noise sensitivity. Therefore, these methods provide alternative yet related strategies for regularization in multi-channel forecasting problems.

**Remark 5** (Equivalence Condition of Stop-Gradient Root Purge). *For the stop-gradient objective*

$$\widetilde{J}(\mathbf{W}) = \|\mathbf{Y} - \mathbf{XW}\|_F^2 + \lambda \|\mathbf{RW}\|_F^2,$$

*where the residual matrix $\mathbf{R}$ is treated as constant during optimization, the same conclusion from Proposition 12 holds.*

**Remark 6** (Bounded Deviation of Stop-Gradient Root Purge). *For the stop-gradient objective*

$$\widetilde{J}(\mathbf{W}) = \|\mathbf{Y} - \mathbf{XW}\|_F^2 + \lambda \|\mathbf{RW}\|_F^2,$$

*where the residual matrix $\mathbf{R}$ is treated as constant during optimization, the gradient term become $\nabla \widetilde{J}(\mathbf{W}) = -2\mathbf{X}^\top \mathbf{R} + 2\lambda \mathbf{R}^\top \mathbf{RW}$, and the bound as stated in Proposition 13 becomes*

$$\|\mathbf{W}^* - \mathbf{W}_{OLS}\|_F \leq \frac{\|\mathbf{E}^\top \mathbf{E} \mathbf{W}_{OLS}\|_F}{1/\lambda \cdot \sigma_{\min}(\mathbf{X}^\top \mathbf{X}) + \sigma_{\min}(\mathbf{E}^\top \mathbf{E})}.$$

To further illustrate the relationship between Root Purge and Rank Reduction, we have the following results:

**Proposition 14** (Anisotropic Shrinkage and Conditional Singular Value Shrinkage at the Stationary Point). *Assume that $\mathbf{X}^\top \mathbf{X} \succ 0$, and $\mathbf{W}^*$ satisfies the orthogonality condition*

$$\mathbf{X}^\top (\mathbf{Y} - \mathbf{XW}^*)\mathbf{W}^* = \mathbf{0}.$$

*Then there exists a symmetric positive semidefinite matrix $\mathbf{S} \preceq \mathbf{I}$ such that*

$$\mathbf{W}^* = \mathbf{S}\,\mathbf{W}_{\mathrm{OLS}},$$

*where*

$$\mathbf{S} = \left(\mathbf{I} + (\mathbf{X}^\top \mathbf{X})^{-1}\lambda \mathbf{R}^{*\top}\mathbf{R}^*\right)^{-1}, \qquad \mathbf{R}^* = \mathbf{Y} - \mathbf{XW}^*.$$

*If we further assume that $\mathbf{X}^\top \mathbf{X}$ and $\mathbf{R}^\top \mathbf{R}$ are simultaneously diagonalizable. Then the unique minimizer $\mathbf{W}^*$ satisfies*

$$\sigma_i(\mathbf{W}^*) \leq \sigma_i(\mathbf{W}_{\mathrm{OLS}}), \quad \forall i.$$

*Proof.* Let $\mathbf{R} = \mathbf{Y} - \mathbf{XW}$. Using standard matrix calculus, the gradient of $J$ is given by

$$\nabla J(\mathbf{W}) = -2\mathbf{X}^\top \mathbf{R} + 2\lambda \left(\mathbf{R}^\top \mathbf{RW} - \mathbf{X}^\top \mathbf{RWW}^\top\right).$$

At a stationary point $\mathbf{W}^*$, $\nabla J(\mathbf{W}^*) = \mathbf{0}$, which yields

$$\mathbf{X}^\top \mathbf{R}^* + \lambda \mathbf{X}^\top \mathbf{R}^*\mathbf{W}^*\mathbf{W}^{*\top} = \lambda \mathbf{R}^{*\top}\mathbf{R}^*\mathbf{W}^*.$$

Under the assumed orthogonality condition $\mathbf{X}^\top \mathbf{R}^*\mathbf{W}^* = \mathbf{0}$, the second term on the left-hand side vanishes, and the above equation reduces to

$$(\mathbf{X}^\top \mathbf{X} + \lambda \mathbf{R}^{*\top}\mathbf{R}^*)\mathbf{W}^* = \mathbf{X}^\top \mathbf{Y}.$$

Let $\mathbf{A} = \mathbf{X}^\top \mathbf{X}$, $\mathbf{B} = \lambda \mathbf{R}^{*\top}\mathbf{R}^* \succeq 0$, and $\mathbf{C} = \mathbf{X}^\top \mathbf{Y}$. Then

$$\mathbf{W}^* = (\mathbf{A} + \mathbf{B})^{-1}\mathbf{C}, \qquad \mathbf{W}_{\mathrm{OLS}} = \mathbf{A}^{-1}\mathbf{C}.$$

Using the identity

$$(\mathbf{A} + \mathbf{B})^{-1} = (\mathbf{I} + \mathbf{A}^{-1}\mathbf{B})^{-1}\mathbf{A}^{-1},$$

we obtain

$$\mathbf{W}^* = \underbrace{(\mathbf{I} + \mathbf{A}^{-1}\mathbf{B})^{-1}}_{\mathbf{S}} \mathbf{W}_{\mathrm{OLS}}.$$

Since $\mathbf{A}^{-1}\mathbf{B} \succeq 0$, it follows that $\mathbf{0} \preceq \mathbf{S} \preceq \mathbf{I}$, completing the proof of our first conclusion. Additionally, by further assumptions, $\mathbf{A}$ and $\mathbf{B}$ are simultaneously diagonalizable. Hence there exists an orthogonal matrix $\mathbf{U}$ such that

$$\mathbf{A} = \mathbf{U}\,\mathrm{diag}(a_i)\mathbf{U}^\top, \qquad \mathbf{B} = \mathbf{U}\,\mathrm{diag}(b_i)\mathbf{U}^\top, \quad b_i \geq 0.$$

Therefore,

$$(\mathbf{A}+\mathbf{B})^{-1} = \mathbf{U}\,\mathrm{diag}\left(\frac{1}{a_i+b_i}\right)\mathbf{U}^\top, \qquad \mathbf{A}^{-1} = \mathbf{U}\,\mathrm{diag}\left(\frac{1}{a_i}\right)\mathbf{U}^\top.$$

Let $\widetilde{\mathbf{C}} = \mathbf{U}^\top\mathbf{C}$. Then

$$\mathbf{W}^* = \mathbf{U}\,\mathrm{diag}\left(\frac{1}{a_i+b_i}\right)\widetilde{\mathbf{C}}, \qquad \mathbf{W}_{\mathrm{OLS}} = \mathbf{U}\,\mathrm{diag}\left(\frac{1}{a_i}\right)\widetilde{\mathbf{C}}.$$

Since $a_i + b_i \geq a_i$ for all $i$, it follows that

$$\frac{1}{a_i+b_i} \leq \frac{1}{a_i}, \quad \forall i.$$

Consequently, each singular value of $\mathbf{W}^*$ is bounded above by the corresponding singular value of $\mathbf{W}_{\mathrm{OLS}}$, which completes the proof. $\qquad\square$

The first part of Proposition 14 establishes that, under a mild orthogonality condition, the stationary solution $\mathbf{W}^*$ can be expressed as a linear transformation of the OLS estimator via a contraction operator $\mathbf{S} \preceq \mathbf{I}$. This result reveals an *anisotropic* shrinkage mechanism, where different directions in parameter space are suppressed to different degrees depending on the residual covariance. With further assumption in the second part, we can additionally perform a component-wise comparison of singular values between $\mathbf{W}^*$ and $\mathbf{W}_{\mathrm{OLS}}$.

The simultaneous diagonalizability of $\mathbf{X}^\top\mathbf{X}$ and $\mathbf{R}^\top\mathbf{R}$ should be interpreted as an analytical choice of coordinate system rather than a restrictive modeling assumption. Indeed, the result characterizes shrinkage behavior in the common eigenbasis of the data covariance and the residual covariance. In particular, directions associated with large residual energy experience stronger suppression, while directions well explained by the data are preserved. As the regularization parameter $\lambda$ increases, this selective attenuation progressively eliminates low-energy singular modes, effectively reducing the rank of the solution. This theoretical behavior is clearly reflected in our empirical results, as shown in Figure 2 and Figure 14, where increasing $\lambda$ progressively attenuates smaller singular values. This supports the interpretation that Root Purge acts similarly to rank reduction techniques by selectively suppressing spurious or noisy components (roots) in the solution. Hence, Root Purge can be seen as a principled mechanism for regularizing overparameterized models by promoting low-rank structure.

**Remark 7** (Anisotropic vs. isotropic shrinkage). *Unlike ridge regression, which applies uniform (isotropic) shrinkage across all directions, the shrinkage operator $\mathbf{S}$ depends on the residual covariance $\mathbf{R}^{*\top}\mathbf{R}^*$ and therefore induces a data-dependent, directionally selective contraction of the OLS estimator.*

**Remark 8** (Norm contraction). *As a direct consequence of $\mathbf{0} \preceq \mathbf{S} \preceq \mathbf{I}$, we have*

$$\|\mathbf{W}^*\|_2 \leq \|\mathbf{W}_{\mathrm{OLS}}\|_2, \qquad \|\mathbf{W}^*\|_F \leq \|\mathbf{W}_{\mathrm{OLS}}\|_F.$$

**Remark 9** (Shrinkage of Stop-Gradient Root Purge). *Consider the stop-gradient objective*

$$\widetilde{J}(\mathbf{W}) = \|\mathbf{Y} - \mathbf{X}\mathbf{W}\|_F^2 + \lambda\|\mathbf{R}\mathbf{W}\|_F^2,$$

*where the residual matrix $\mathbf{R}$ is treated as constant during optimization. We can drop the orthogonality condition that $\mathbf{X}^\top(\mathbf{Y} - \mathbf{X}\mathbf{W}^*)\mathbf{W}^* = \mathbf{0}$. We still arrive at the conclusion of Proposition 14 given the rest of assumptions are satisfied.*

Finally, to support the implementation and theoretical analysis of our algorithm, we present a classical convergence result for gradient descent in the non-convex setting. Although our objective function is non-convex, the following proposition guarantees that gradient descent with a properly chosen step size converges to a stationary point. This forms the basis for understanding the optimization behavior in our setting.

**Proposition 15** (Convergence of Gradient Descent Algorithm). *Let $J(\mathbf{W})$ be a differentiable function with L-Lipschitz continuous gradient:*

$$\|\nabla J(\mathbf{W}_1) - \nabla J(\mathbf{W}_2)\|_F \leq L\|\mathbf{W}_1 - \mathbf{W}_2\|_F, \quad \forall\, \mathbf{W}_1, \mathbf{W}_2.$$

*Assume $J(\mathbf{W})$ is bounded below by $J^*$. Consider the gradient descent update:*

$$\mathbf{W}_{k+1} = \mathbf{W}_k - \eta \nabla J(\mathbf{W}_k), \quad with \ \eta = \frac{1}{L}.$$

*Then, after $K$ iterations, the minimum gradient norm satisfies:*

$$\min_{0 \leq k < K} \|\nabla J(\mathbf{W}_k)\|_F^2 \leq \frac{2L(J(\mathbf{W}_0) - J^*)}{K}.$$

*Proof.* By Lipschitz continuity of $\nabla J(\mathbf{W})$, for any $\mathbf{W}$ and $\mathbf{W}' = \mathbf{W} - \eta \nabla J(\mathbf{W})$:

$$J(\mathbf{W}') \leq J(\mathbf{W}) + \langle \nabla J(\mathbf{W}), \mathbf{W}' - \mathbf{W} \rangle + \frac{L}{2}\|\mathbf{W}' - \mathbf{W}\|_F^2.$$

Substituting $\mathbf{W}' - \mathbf{W} = -\eta \nabla J(\mathbf{W})$ gives:

$$J(\mathbf{W}') \leq J(\mathbf{W}) - \eta \|\nabla J(\mathbf{W})\|_F^2 + \frac{L\eta^2}{2}\|\nabla J(\mathbf{W})\|_F^2.$$

With $\eta = \frac{1}{L}$, we have:

$$J(\mathbf{W}_{k+1}) \leq J(\mathbf{W}_k) - \frac{1}{2L}\|\nabla J(\mathbf{W}_k)\|_F^2.$$

Summing over $k = 0, \ldots, K-1$ gives:

$$\sum_{k=0}^{K-1} \frac{1}{2L}\|\nabla J(\mathbf{W}_k)\|_F^2 \leq \sum_{k=0}^{K-1} [J(\mathbf{W}_k) - J(\mathbf{W}_{k+1})] = J(\mathbf{W}_0) - J(\mathbf{W}_K).$$

Since $J(\mathbf{W}_K) \geq J^*$, we get:

$$\sum_{k=0}^{K-1} \|\nabla J(\mathbf{W}_k)\|_F^2 \leq 2L(J(\mathbf{W}_0) - J^*).$$

Taking the average gives:

$$\frac{1}{K}\sum_{k=0}^{K-1} \|\nabla J(\mathbf{W}_k)\|_F^2 \leq \frac{2L(J(\mathbf{W}_0) - J^*)}{K},$$

implying:

$$\min_{0 \leq k < K} \|\nabla J(\mathbf{W}_k)\|_F^2 \leq \frac{2L(J(\mathbf{W}_0) - J^*)}{K}.$$

$\square$

As a consequence of this proposition, we see that gradient descent reaches an $\epsilon$-stationary point ($\|\nabla J(\mathbf{W})\|_F^2 \leq \epsilon$) in $\mathcal{O}(1/\epsilon)$ iterations. While recursively solving least square may provides faster convergence through its second-order approximation, it requires solving a linear system with an additional computational cost of $\mathcal{O}(L^3)$ per iteration due to the matrix inversion of $\mathbf{X}^\top \mathbf{X} + \lambda \mathbf{R}_k^\top \mathbf{R}_k$ (computational costs in matrix multiplications are not counted as it is also required for other algorithms). In contrast, gradient descent avoids this cubic scaling, making it preferable for extended lookback window problems where $L \gg 1$. Note that this convergence result applies to any objective function satisfying the Lipschitz gradient condition. In particular, it applies to our specific optimization problem:

**Remark 10.** *The gradient $\nabla J(\mathbf{W})$ is Lipschitz continuous with constant $L$ satisfying[9]:*

$$L \leq 2\|\mathbf{X}\|_2^2 + 2\lambda(6M^2\|\mathbf{X}\|_2^2 + 6M\|\mathbf{X}\|_2\|\mathbf{Y}\|_2 + \|\mathbf{Y}\|_2^2) \tag{11}$$

*where:*

- $M = \max_{\mathbf{W}\in\mathcal{W}} \|\mathbf{W}\|_F$ *for* $\mathcal{W} = \{\mathbf{W}|J(\mathbf{W}) \leq J(\mathbf{W}_0)\}$;

- $\|\mathbf{X}\|_2$ *and* $\|\mathbf{Y}\|_2$ *are spectral norms.*

**Remark 11.** *For the stop-gradient objective*

$$\widetilde{J}(\mathbf{W}) = \|\mathbf{Y} - \mathbf{X}\mathbf{W}\|_F^2 + \lambda\|\mathbf{R}\mathbf{W}\|_F^2,$$

*where the residual matrix $\mathbf{R}$ is treated as constant during optimization, the gradient $\nabla\widetilde{J}(\mathbf{W})$ is Lipschitz continuous with constant $L$ satisfying:*

$$\widetilde{L} \leq 2\|\mathbf{X}\|_2^2 + 2\lambda(3M^2\|\mathbf{X}\|_2^2 + 4M\|\mathbf{X}\|_2\|\mathbf{Y}\|_2 + \|\mathbf{Y}\|_2^2) \tag{12}$$

### D.8 ANALYSIS OF CHARACTERISTIC ROOTS

To this point, we have not explicitly formalized the connection between rank reduction and the resulting restructuring of polynomial roots. In the following proposition, we establish this relationship rigorously, highlighting how the structure of the underlying system governs the interplay between low-rank approximations and root perturbations—an idea central to the focus of this paper.

**Proposition 16** (Root Perturbation under Low-Rank Approximation). *Let $\mathbf{W} \in \mathbb{R}^{L\times H}$ be a coefficient matrix where each column $\mathbf{w}_j$ defines a monic polynomial:*

$$P_j(r) = r^{L+j-1} - \sum_{i=1}^{L} W_{i,j}r^{L-i}$$

*with distinct roots $\{r_1^{(j)}, \ldots, r_{L+j-1}^{(j)}\}$ for each $j = 1, \ldots, H$. Let $\tilde{\mathbf{W}}$ be approximated coefficient matrix with error bound $\|\mathbf{W} - \tilde{\mathbf{W}}\|_F \leq \epsilon$. Then for each column polynomial $\tilde{P}_j(r)$ of $\tilde{\mathbf{W}}$ with roots $\{\tilde{r}_1^{(j)}, \ldots, \tilde{r}_{L+j-1}^{(j)}\}$, the following root perturbation bound holds:*

$$\max_{1\leq i\leq L+j-1} \min_{1\leq l\leq L+j-1} |\tilde{r}_i^{(j)} - r_l^{(j)}| \leq \kappa_j \cdot \epsilon$$

*where $\kappa_j$ is the condition number of the root-finding problem for $P_j(r)$, dependent on the minimal separation between roots.*

*Proof.* Given $\|\mathbf{W} - \tilde{\mathbf{W}}\|_F \leq \epsilon$, the perturbation in each column satisfies: $\|\mathbf{w}_j - \tilde{\mathbf{w}}_j\|_2 \leq \epsilon$. For each polynomial $P_j(r)$, consider its companion matrix:

$$\mathbf{C}_j = [\,\mathbf{S} \mid \mathbf{w}_j'\,] \in \mathbb{R}^{(L+j-1)\times(L+j-1)}$$

where:

- $\mathbf{w}_j' = \begin{bmatrix} \mathbf{0}_{j-1} \\ \mathbf{w}_j \end{bmatrix}$ (zero-padded coefficient vector);

- $\mathbf{S}$ is the shift matrix:

$$\mathbf{S} = \begin{bmatrix} 0 & & & \\ 1 & 0 & & \\ & 1 & \ddots & \\ & & \ddots & 0 \\ & & & 1 \end{bmatrix} \in \mathbb{R}^{(L+j-1)\times(L+j-2)}$$

with ones on the first subdiagonal and zeros elsewhere.

---

[9]The Lipschitz constant $L$ for $\nabla J(\mathbf{W})$ can be estimated as two parts: root-seeking term $\|\mathbf{Y} - \mathbf{X}\mathbf{W}\|_F^2$ contributes to $2\|\mathbf{X}\|_2^2$, and root-purging term $\|(\mathbf{Y} - \mathbf{X}\mathbf{W})\mathbf{W}\|_F^2$ has two gradient components: $2\lambda\mathbf{R}^\top\mathbf{R}\mathbf{W}$ and $2\lambda\mathbf{X}^\top\mathbf{R}\mathbf{W}\mathbf{W}^\top$. The bound of $L$ can be found by looking at each component separately. The first component give us the bound of $2\lambda\left[2\|\mathbf{X}\|_2(\|\mathbf{Y}\|_2 + \|\mathbf{X}\|_2M)M + (\|\mathbf{Y}\|_2 + \|\mathbf{X}\|_2M)^2\right]$ and the later give us the bound of $2\lambda\left[2\|\mathbf{X}\|_2(\|\mathbf{Y}\|_2 + \|\mathbf{X}\|_2M)M + \|\mathbf{X}\|_2^2M^2\right]$ via triangle inequalities and submultiplicativity.

Note that the eigenvalues of $\mathbf{C}_j$ are exactly $\{r_l^{(j)}\}_{l=1}^{L+j-1}$. Under perturbation $\boldsymbol{\Delta}_j = \mathbf{C}_j - \tilde{\mathbf{C}}_j$, where $\tilde{\mathbf{C}}_j$ is the companion matrix for $\tilde{P}_j(r)$, we have:

$$\|\boldsymbol{\Delta}_j\|_2 = \|\mathbf{w}_j - \tilde{\mathbf{w}}_j\|_2 \leq \epsilon$$

By assumption $\{r_l^{(j)}\}_{l=1}^{L+j-1}$ from $\mathbf{C}_j$ are all distinct, applying Bauer-Fike theorem to a eigenvalues $\tilde{r}_i^{(j)}$ from $\tilde{\mathbf{C}}_j$, we have the following:

$$\min_l |\tilde{r}_i^{(j)} - r_l^{(j)}| \leq \kappa(\mathbf{V}_j)\|\boldsymbol{\Delta}_j\|_2$$

where $\mathbf{V}_j$ is the Vandermonde matrix that diagonalizes $\mathbf{C}_j$ as $\mathbf{C}_j = \mathbf{V}_j^{-1}\boldsymbol{\Lambda}_j\mathbf{V}_j$, and $\kappa(\mathbf{V}_j) = \kappa(\mathbf{V}_j^{-1}) = \|\mathbf{V}_j\|_2\|\mathbf{V}_j^{-1}\|_2$ is the condition number of the eigenvector matrix for $\mathbf{C}_j$. For distinct roots, the following form controls the upper bound of the conditioning (Gautschi, 1962):

$$\kappa(\mathbf{V}_j) \leq (L+j-1) \cdot \underbrace{\left(\max_l \sum_{m=1}^{L+j-1} \left|r_l^{(j)}\right|^{m-1}\right)}_{\text{directly evaluating } \|\mathbf{V}_j\|_\infty} \cdot \underbrace{\max_l \prod_{m \neq l} \frac{1 + |r_m^{(j)}|}{|r_l^{(j)} - r_m^{(j)}|}}_{\text{Gautschi's bound of } \|\mathbf{V}_j^{-1}\|_\infty} =: \kappa_j$$

The leading term of the above equation is a result of the inequality between $\|\cdot\|_2$ and $\|\cdot\|_\infty$ controlled solely by matrix shapes. In this case we have $\|\mathbf{V}_j\|_2 \leq \sqrt{L+j-1}\|\mathbf{V}_j\|_\infty$ as $\mathbf{V}_j \in \mathbb{R}^{(L+j-1)\times(L+j-1)}$. Combining these results yields:

$$\max_{1 \leq i \leq L+j-1} \min_{1 \leq l \leq L+j-1} |\tilde{r}_i^{(j)} - r_l^{(j)}| \leq \kappa_j \cdot \epsilon$$

$\square$

From this proposition, we see that the bound on root perturbation is directly controlled by the Frobenius norm of the approximation error $\|\mathbf{W} - \tilde{\mathbf{W}}\|_F$. Since we have explicitly derived Frobenius norm bounds for each method, including Root Purge, RRR, and DWRR, the proposition ensures that the resulting perturbations to the root structure are similarly bounded. Consequently, the leading roots are preserved, while less significant roots may shift or collapse. This behavior reflects the structural influence imposed by the rank constraint and reinforces the central theme of the paper: that rank reduction reshapes the root landscape in a principled and quantifiable manner.

# E    EXPERIMENTS

## E.1    DATASET

We provide a summary of the dataset we used in Table 7. For ETT (Zhou et al., 2021) data, we follow previous works (Lin et al., 2024; Xu et al., 2023; Wu et al., 2022) and divide the dataset into train, val, and test sets with a $6 : 2 : 2$ ratio. For the rest of the datasets, they are divided in to train, validation, and test sets with a $7 : 1 : 2$ ratio. The characteristics of these datasets are further illustrated by representative samples in Figure 12.

For the synthetic dataset we used in Section 5.3 to study data scaling property, noise robustness, and root distribution, the train, validation, and test sets are split with a $5 : 2.5 : 2.5$ ratio.

Table 7: Detailed information of the datasets we used in this paper.

| Datasets | ETTh1 | ETTh2 | ETTm1 | ETTm2 | Exchange | Weather | Traffic | Electricity |
|---|---|---|---|---|---|---|---|---|
| No. of Channels | 7 | 7 | 7 | 7 | 8 | 21 | 862 | 321 |
| Timesteps | 17420 | 17420 | 69680 | 69680 | 7588 | 52696 | 17544 | 26304 |
| Frequency | Hourly | Hourly | 15min | 15min | Daily | 10min | Hourly | Hourly |
| Application Domain | Electricity | Electricity | Electricity | Electricity | Economy | Weather | Traffic | Electricity |

Additionally, we also use M4 (Makridakis et al., 2020) dataset to test the robustness of our methods under short lookback window as well as their performances in short-term forecasting tasks. The M4

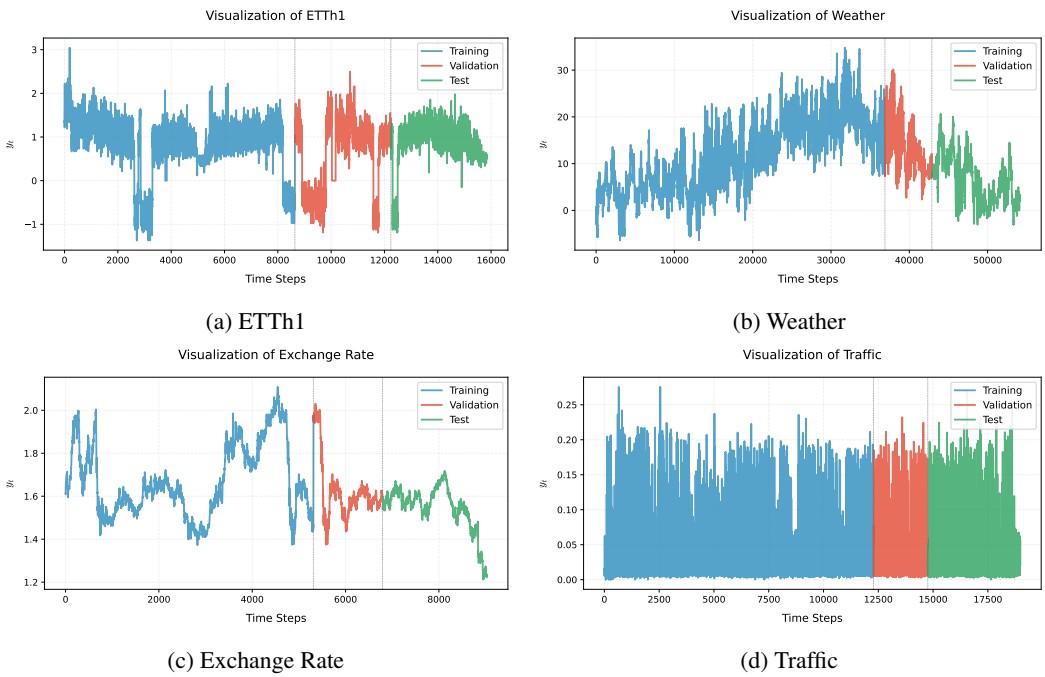

Figure 12: Representative samples from the ETTh1, weather, exchange rate, and Traffic datasets, illustrating the complex, non-stationary trends characteristic of real-world time series data.

dataset consists of 100,000 real-world time series from diverse domains such as finance, economics, demographics, and industry. Each subset of M4 is collected in yearly, quarterly, monthly, weekly, daily, and hourly frequencies, with a mixture of data from the domains mentioned before. For the M4 dataset, the lookback window $L$ is set to $2 \times H$, where $H$ is the forecasting horizon, which differs from long-term forecasting datasets. The details of M4 dataset are listed in Tabel 8.

Table 8: Detailed information of the M4 dataset.

| Datasets | Yearly | Quarterly | Monthly | Weekly | Daily | Hourly |
|---|---|---|---|---|---|---|
| No. of Channels | 1 | 1 | 1 | 1 | 1 | 1 |
| Forecasting Horizon ($H$) | 6 | 8 | 18 | 13 | 14 | 48 |
| Training/Test Set Size | 23000 | 24000 | 48000 | 359 | 4227 | 414 |

### E.2 IMPLEMENTATION DETAILS

**Main Experiments.** In experiments for both RRR and Root Purge, the length of lookback window is chosen to be 720. The major reason is that the lookback window determines the number of roots a linear model can capture, which is the key to the generalization behavior of linear time series models. This setting is also consistent with prior works in linear time series models (Xu et al., 2023; Lin et al., 2024). For experiments with RRR, we use singular value decomposition (SVD) to directly solve linear regression for the least squares (OLS) in the time domain on the CPU. Each experiment for RRR is only done once, as the results are produced in close form and therefore static.

We adopt a frequency-domain linear layer for experiments with Root Purge for its better training stability (Xu et al., 2023; Yi et al., 2023). Due to the linearity of the Fourier Transform and the inverse Fourier Transform, learning a complex-valued linear layer in the frequency domain still results in an overall linear mapping, on which the Root Purge algorithm has theoretical guarantees for the effectiveness of the rank-nullity trade-off. We provide a comparison of linear models in the time domain and the frequency domain in Table 12. Further, we provide the results of performing Root Purge on the time domain in Table 13.

**Transforming Frequency-Domain Weights to the Time Domain.**    We explain how to transform a learned weight matrix in the frequency domain, denoted $\mathbf{W}_F$, into its corresponding time-domain representation $\mathbf{W}_T$. Consider a general setting where the model defines a linear operator $\mathcal{G}_{\mathbf{W}}(\cdot)$ (which may involve a composition of multiple linear transforms). Our goal is to recover a single matrix $\tilde{\mathbf{W}}$ equivalent to $\mathcal{G}(\cdot)$, i.e. $\mathcal{G}_{\mathbf{W}}(\mathbf{X}) = \mathbf{X}\tilde{\mathbf{W}}$ for any input $\mathbf{X}$. A convenient way to obtain $\tilde{\mathbf{W}}$ is by evaluating $\mathcal{G}_{\mathbf{W}}$ on the identity matrix $\mathbf{I}$:

$$\tilde{\mathbf{W}} = \mathcal{G}_{\mathbf{W}}(\mathbf{I}).$$

Since $\mathcal{G}_{\mathbf{W}}$ represents the trained model, we can compute $\tilde{\mathbf{W}}$ directly by feeding the identity matrix into the model. This method is general and applies to any linear operator (so that we must have $\tilde{\mathbf{W}} = \mathbf{W}_T$), whether defined in the time or frequency domain. Specifically, when the model is implemented as a frequency-domain transform,

$$\mathcal{G}_{\mathbf{W}} = \mathcal{F}^{-1} \circ \mathbf{W}_F \circ \mathcal{F},$$

which yields the full time-domain weight matrix $\mathbf{W}_T$ associated with the learned operator.

A similar approach can be used to approximate a nonlinear operator $\mathcal{G}$ with a linear one. Evaluating $\mathcal{G}$ on the identity matrix yields: $\tilde{\mathbf{W}} \approx \mathcal{G}(\mathbf{I})$, which defines a linear surrogate $\tilde{\mathbf{W}}$ such that: $\mathcal{G}(\mathbf{X}) \approx \tilde{\mathbf{W}}\mathbf{X}$. This is particularly useful when $\mathcal{G}$ is locally linear, offering a tractable way to analyze or approximate nonlinear models, including neural networks or nonlinear filters.

**Computing Resource.**    All experiments of Root Purge are implemented using Pytorch (Paszke et al., 2017) and conducted on a single NVIDIA V100 32GB GPU. We employ MSE (Mean Squared Error) as both our loss function and the evaluation metrics, following prior works in linear time series forecasting and our theoretical analysis. Each of the experiments for Root Purge is repeated 5 times to remove run-to-run differences.

For more details on the running settings of Root Purge, please refer to our code and running scripts in the supplementary material[10].

### E.3   Details of Baseline Results and Comparing with Additional Baselines

We provide details for obtaining our baseline results. For many previous works, the results listed are impacted by a long-standing bug[11], mistakenly setting `droplast=True` for the test and validation dataset. For this reason, we either cite results with this bug fixed, or reproduce the results with the official open source code with this bug fixed. The details of our baseline results are as follows:

**FEDformer** (Zhou et al., 2022) Transformer-based model with sparse attention. We cite the results with this bug fixed from Xu et al. (2023) and reproduce missing results with the open-source implementation at `https://github.com/thuml/Time-Series-Library`.

**PatchTST** (Nie et al., 2022) Transformer-based model considering a short time series patch as a single token.    We cite the results with this bug fixed from Xu et al. (2023) and reproduce missing results with the open-source implementation at `https://github.com/thuml/Time-Series-Library`.

**TimesNet** (Wu et al., 2023) Convolution-based model applied on time series that is converted to 2D. We cite the results with this bug fixed from Xu et al. (2023) and reproduce missing results with the open-source implementation at `https://github.com/thuml/Time-Series-Library`.

**TSLANet** (Eldele et al., 2024) Convolution-based model. The official implementation is affected by the bug mentioned before. Thus, we reproduce the result using the open-source implementation at `https://github.com/emadeldeen24/TSLANet/`. For a fair comparison with our results, we use a fixed lookback window of 720. The rest of the setup follows the script provided in the repository.

**Plain FilterNet** (Yi et al., 2024) Convolution based model with a 2-layer MLP head. The official implementation is affected by the bug mentioned before. Thus, we reproduce the result using the

---

[10]Our code is available at https://github.com/Wangzzzzzzzz/RootPurge.
[11]See more details in `https://github.com/VEWOXIC/FITS/`.

open-source implementation at `https://github.com/aikunyi/FilterNet/`. For a fair comparison with our results, we use a fixed lookback window of 720. The rest of the setup follows the script provided in the repository. When no official shell script is provided, we use the default settings in the code.

**FITS** (Xu et al., 2023) Linear model on the frequency domain. FITS takes lookback window as a hyperparameter. Thus, for a fair comparison with our results, we use a fixed lookback window of 720 and reproduce the result using the open-source implementation at `https://github.com/VEWOXIC/FITS/`. The rest of the hyperparameters follow the official implementation that yields the best results.

**SparseTSF** (Lin et al., 2024) Linear model with convolution backbones. We cite the result from the original paper, as it already contains the result after fixing the bug.

**DLinear** (Liu et al., 2024) Linear model with moving average. We cite the result with the bug fixed from Xu et al. (2023) and reproduce missing results with the open-source implementation at `https://github.com/vivva/DLinear/`.

### E.4 Comparing on Traffic and Electricity Dataset

Due to data scaling property in Proposition 1, we expect RRR and Root Purge to show more significant improvements on small-scale datasets. Under CI settings, large datasets such as Electricity and Traffic constitute a huge amount of data for training. It is clear in Table 9 that complex non-linear models achieve better results than linear models, taking advantage of larger model capacity.

For linear models, these large-scale datasets allow any linear model to approach the optimum via only the root-seeking loss. In Appendix E.8, we reaffirm that although the numerical analysis from RRR and DWRR still shows that the optimal rank of $\mathbf{W}$ is *not* full, the performance of full-rank models can come close to that of their rank-truncated counterparts.

As shown in Table 9, the performance of all linear models is very close on these two datasets. RRR and Root Purge either match or outperform other linear models (i.e., DLinear, FITS, and SparseTSF). Notably, for Root Purge, while data scaling has already pushed the model close to its optimality, performance remains robust with a relatively large regularization coefficient ($\lambda = 0.125$). This reaffirms the robustness of our methods.

Table 9: Forecasting result for ECL and Traffic with horizon $H \in \{96, 192, 336, 720\}$ with lookback window of length $L = 720$. Following the main experiment settings, for RRR, we tune the rank on the validation set, select the top three with lowest validation MSE, and report the best test result. For Root Purge, we select the best test MSE from a hyperparameter search over $\lambda \in [0.125, 0.25, 0.5]$. The best results are highlighted in **red**, and the second-best in **blue**. Methods we proposed in this paper are **bolded** in this table.

| Dataset | $H$ | FEDformer | FilterNet | TSLANet | TimesNet | PatchTST | DLinear | SparseTSF | FITS | **RRR** | **Root Purge** |
|---|---|---|---|---|---|---|---|---|---|---|---|
| ECL | 96 | 0.188 | 0.136 | **0.129** | 0.168 | **0.129** | 0.140 | 0.138 | 0.135 | 0.133 | 0.134 |
| | 192 | 0.197 | 0.153 | **0.147** | 0.184 | 0.149 | 0.153 | 0.151 | 0.149 | 0.148 | **0.148** |
| | 336 | 0.212 | 0.168 | **0.162** | 0.198 | 0.166 | 0.169 | 0.166 | 0.165 | **0.164** | 0.165 |
| | 720 | 0.244 | 0.207 | **0.196** | 0.220 | 0.210 | 0.204 | 0.205 | 0.205 | **0.203** | 0.205 |
| Traffic | 96 | 0.573 | 0.393 | **0.362** | 0.593 | **0.366** | 0.413 | 0.389 | 0.386 | 0.385 | 0.386 |
| | 192 | 0.611 | 0.408 | **0.377** | 0.617 | **0.388** | 0.423 | 0.398 | 0.397 | 0.396 | 0.397 |
| | 336 | 0.621 | 0.424 | **0.391** | 0.629 | **0.398** | 0.437 | 0.411 | 0.411 | 0.410 | 0.411 |
| | 720 | 0.630 | 0.462 | **0.430** | 0.640 | 0.457 | 0.466 | 0.448 | 0.449 | **0.448** | 0.450 |

### E.5 Computational Cost & Complexity Analysis

While the focuses of RRR, DWRR, and Root Purge are not maximizing computational costs, we provide the following analysis of the computational complexity.

**RRR and DWRR** For inference, under the most general setting and within a channel-independent framework, if we consider a dataset with $c$ channels, the computational complexity for all linear models is $\mathcal{O}(c \cdot L \cdot H)$, where $L$ is the input sequence length and $H$ is the output sequence length. However, due to the rank limitation we impose on the intermediate weight matrix, the complexity

in our methods (RRR and DWRR) can be significantly reduced to $\mathcal{O}(c \cdot \rho \cdot (H + L))$, where $\rho \ll \min(H, L)$ denotes the rank of the matrix.

For training, the computational complexity of the RRR algorithm can be broken down into four steps:

- Estimating $\mathbf{W}_{\text{OLS}}$: $\mathcal{O}(NL^2 + NLH + L^3 + L^2H)$
- Calculating the estimation: $\mathcal{O}(NLH)$
- Performing SVD: $\mathcal{O}(\min(NH^2, N^2H))$
- Rank truncation: $\mathcal{O}(H\rho^2 + LH\rho)$

Overall, given that the sample size $N$ is generally much larger than both the input and output lengths, and due to the low-rank property of the weight matrix, the total computational complexity can be approximated as $\mathcal{O}(NL^2 + NLH)$.

For the DWRR algorithm, the training complexity involves three steps:

- Estimating $\mathbf{W}_{\text{OLS}}$: $\mathcal{O}(NL^2 + NLH + L^3 + L^2H)$
- Performing SVD: $\mathcal{O}(\min(LH^2, L^2H))$
- Rank truncation: $\mathcal{O}(LH\rho)$

Similarly, under general conditions, the total computational complexity is approximately $\mathcal{O}(NL^2 + NLH)$.

**Root Purge** For the Root Purge algorithm, since its training process relies on gradient-based optimization, we measure the training memory consumption and iteration speed on an NVIDIA V100 GPU. In addition, we also measure the inference time and parameter size. We summarize the results on the ETTh2 and ECL datasets in Table 10 and Table 11 respectively.

Table 10: Computational measurements based on ETTh2 dataset with batch 8. Lookback window length is set to $L = 720$ and forecasting horizon is set to $H = 192$.

| Model | Train GPU Memory (MB ↓) | Training Time (ms/iter ↓) | Parameter Counts (↓) | Inference Time (ms ↓) |
|---|---|---|---|---|
| SparseTSF | 424 | 1.5 | 8.6K | 0.28 |
| DLinear | 410 | 1.5 | 415.3K | 0.30 |
| FITS | 414 | 1.9 | 48.9K | 0.53 |
| FilterNet | 434 | 4.0 | 235.3K | 0.55 |
| TSLANet | 604 | 3.7 | 605.4K | 1.25 |
| Koopa | 478 | 25.1 | 252.8K | 2.79 |
| PatchTST | 1028 | 13.9 | 2,611.8K | 3.03 |
| **Root Purge** | 432 | 1.7 | 165.0K | 0.29 |

Table 11: Computational measurements results based on ECL dataset with batch 8. Lookback window length is set to $L = 720$ and forecasting horizon is set to $H = 192$.

| Model | Train GPU Memory (MB ↓) | Training Time (ms/iter ↓) | Parameter Counts (↓) | Inference Time (ms ↓) |
|---|---|---|---|---|
| SparseTSF | 488 | 8.3 | 8.6K | 0.42 |
| DLinear | 472 | 5.2 | 415.3K | 0.49 |
| FITS | 538 | 13.5 | 130.0K | 5.45 |
| FilterNet | 550 | 6.2 | 235.3K | 0.81 |
| TSLANet | 4382 | 91.5 | 1,581.1K | 2.57 |
| Koopa | 1920 | 56.9 | 63,777.2K | 50.89 |
| PatchTST | 15266 | 264.0 | 2,611.8K | 3.73 |
| **Root Purge** | 534 | 6.1 | 165.0K | 0.47 |

As you can see, Root Purge is able to achieve SOTA performances while using close-to-minimum training/inference resources.

E.6 LINEAR FORECASTER WITH BIAS AND WITHOUT BIAS

We list our experimental results comparing the setting of with/without bias in Table 12. The experiments are done for both the time-domain linear block and the frequency domain linear block.

We repeat our experiments 5 times and include the error bars in the table, and the experimental settings in this section follow our main experiments.

Although the gap tends to be larger when fitting a time domain model, for both frequency and time domain models, the one without bias is performing better than/equally well as the model with bias.

The result from this section can also serve as a non-regularized baseline for Root Purge, where we use $\lambda = 0$.

Table 12: Comparison of the time/frequency-domain linear model with bias and without bias. The model without bias tends to perform better/equally well (better results in **red**) compared to the model with bias in both time and frequency domains. Overall, the time domain model without bias performs similarly to the frequency domain model without bias, with only minor degradations.

| Dataset | Horizon | Time Domain | | Frequency Domain | |
|---|---|---|---|---|---|
| | | With Bias | Without Bias | With Bias | Without Bias |
| ETTh1 | 96 | $0.379 \pm 0.001$ | $\textbf{0.378} \pm 0.001$ | $0.375 \pm 0.001$ | $\textbf{0.374} \pm 0.001$ |
| | 192 | $0.415 \pm 0.000$ | $\textbf{0.415} \pm 0.001$ | $0.410 \pm 0.000$ | $\textbf{0.410} \pm 0.000$ |
| | 336 | $0.448 \pm 0.002$ | $\textbf{0.443} \pm 0.001$ | $0.432 \pm 0.001$ | $\textbf{0.431} \pm 0.001$ |
| | 720 | $0.459 \pm 0.001$ | $\textbf{0.435} \pm 0.002$ | $0.428 \pm 0.000$ | $\textbf{0.427} \pm 0.000$ |
| ETTh2 | 96 | $\textbf{0.275} \pm 0.001$ | $0.276 \pm 0.001$ | $\textbf{0.272} \pm 0.000$ | $0.273 \pm 0.000$ |
| | 192 | $0.334 \pm 0.000$ | $\textbf{0.334} \pm 0.001$ | $0.332 \pm 0.000$ | $\textbf{0.332} \pm 0.000$ |
| | 336 | $0.361 \pm 0.002$ | $\textbf{0.360} \pm 0.002$ | $0.355 \pm 0.001$ | $\textbf{0.354} \pm 0.001$ |
| | 720 | $0.396 \pm 0.001$ | $\textbf{0.384} \pm 0.003$ | $0.379 \pm 0.000$ | $\textbf{0.378} \pm 0.000$ |
| ETTm1 | 96 | $0.311 \pm 0.002$ | $\textbf{0.310} \pm 0.001$ | $0.310 \pm 0.001$ | $\textbf{0.308} \pm 0.002$ |
| | 192 | $0.343 \pm 0.003$ | $\textbf{0.340} \pm 0.001$ | $0.339 \pm 0.000$ | $\textbf{0.337} \pm 0.000$ |
| | 336 | $0.369 \pm 0.001$ | $\textbf{0.369} \pm 0.001$ | $0.367 \pm 0.000$ | $\textbf{0.366} \pm 0.000$ |
| | 720 | $0.418 \pm 0.002$ | $\textbf{0.417} \pm 0.001$ | $0.416 \pm 0.000$ | $\textbf{0.415} \pm 0.000$ |
| ETTm2 | 96 | $0.163 \pm 0.001$ | $\textbf{0.163} \pm 0.001$ | $0.164 \pm 0.000$ | $\textbf{0.164} \pm 0.000$ |
| | 192 | $0.217 \pm 0.000$ | $\textbf{0.217} \pm 0.000$ | $\textbf{0.218} \pm 0.000$ | $0.219 \pm 0.000$ |
| | 336 | $0.270 \pm 0.001$ | $\textbf{0.270} \pm 0.001$ | $0.269 \pm 0.000$ | $\textbf{0.269} \pm 0.000$ |
| | 720 | $0.354 \pm 0.001$ | $\textbf{0.353} \pm 0.001$ | $0.350 \pm 0.000$ | $\textbf{0.350} \pm 0.000$ |
| Weather | 96 | $0.143 \pm 0.000$ | $\textbf{0.143} \pm 0.001$ | $0.145 \pm 0.000$ | $\textbf{0.145} \pm 0.000$ |
| | 192 | $0.186 \pm 0.000$ | $\textbf{0.186} \pm 0.001$ | $0.188 \pm 0.001$ | $\textbf{0.187} \pm 0.000$ |
| | 336 | $0.237 \pm 0.001$ | $\textbf{0.237} \pm 0.001$ | $0.238 \pm 0.000$ | $\textbf{0.238} \pm 0.000$ |
| | 720 | $0.310 \pm 0.000$ | $\textbf{0.308} \pm 0.001$ | $0.309 \pm 0.000$ | $\textbf{0.309} \pm 0.000$ |
| Exchange | 96 | $0.091 \pm 0.001$ | $\textbf{0.087} \pm 0.000$ | $0.088 \pm 0.001$ | $\textbf{0.084} \pm 0.000$ |
| | 192 | $0.192 \pm 0.001$ | $\textbf{0.178} \pm 0.001$ | $0.186 \pm 0.000$ | $\textbf{0.176} \pm 0.000$ |
| | 336 | $0.370 \pm 0.001$ | $\textbf{0.328} \pm 0.003$ | $0.346 \pm 0.001$ | $\textbf{0.329} \pm 0.001$ |
| | 720 | $1.084 \pm 0.022$ | $\textbf{0.925} \pm 0.011$ | $0.961 \pm 0.003$ | $\textbf{0.935} \pm 0.002$ |
| ECL | 96 | $0.134 \pm 0.000$ | $\textbf{0.134} \pm 0.000$ | $0.135 \pm 0.000$ | $\textbf{0.135} \pm 0.001$ |
| | 192 | $0.149 \pm 0.000$ | $\textbf{0.149} \pm 0.000$ | $0.149 \pm 0.000$ | $\textbf{0.149} \pm 0.000$ |
| | 336 | $0.165 \pm 0.000$ | $\textbf{0.165} \pm 0.000$ | $0.165 \pm 0.000$ | $\textbf{0.165} \pm 0.000$ |
| | 720 | $0.205 \pm 0.000$ | $\textbf{0.204} \pm 0.000$ | $0.205 \pm 0.000$ | $\textbf{0.205} \pm 0.000$ |
| Traffic | 96 | $0.386 \pm 0.001$ | $\textbf{0.386} \pm 0.000$ | $0.385 \pm 0.000$ | $\textbf{0.385} \pm 0.000$ |
| | 192 | $0.397 \pm 0.000$ | $\textbf{0.397} \pm 0.000$ | $0.397 \pm 0.000$ | $\textbf{0.397} \pm 0.000$ |
| | 336 | $0.411 \pm 0.001$ | $\textbf{0.411} \pm 0.001$ | $0.411 \pm 0.001$ | $\textbf{0.410} \pm 0.000$ |
| | 720 | $0.449 \pm 0.000$ | $\textbf{0.449} \pm 0.000$ | $0.449 \pm 0.000$ | $\textbf{0.449} \pm 0.000$ |

### E.7 ADDITIONAL EMPIRICAL RESULTS FOR ROOT PURGE

**Root Purge on Time Domain** In our main experiment (Section 5), we use a frequency-domain linear block following Yi et al. (2023) and Xu et al. (2023). In this section, we show that Root Purge also works well on the time-domain linear models. Following the experimental setting in the main experiment, we perform hyperparameter selection with $\lambda \in [0.125, 0.25, 0.5]$ to obtain our results. The full experimental results for each choice of hyperparameter can be found in Table 15, and we give a summary table for the result with the best hyperparameter in Table 13.

Hyperparameter sensitivity and singular values are also studied for time-domain linear models with Root Purge. We provide hyperparameter sensitivity in Figure 13 and singular value spectrum in Figure 14. Overall, the hyperparameter sensitivities and singular value trend for Root Purge in the time domain are similar to those in the frequency domain.

**Full Experimental Results of the Impact of Root Purge Hyperparameter** We conduct a broader hyperparameter search with $\lambda \in [1/32, 1/16, 1/8, 1/4, 1/2, 1, 2]$ for ETTh1 and ETTm1; we show a complete result for each of $H = \{96, 192, 336, 720\}$ in Table 14. Overall, a large range of $\lambda$ will yield improved results. Still, there are some differences in the hyperparameter sensitivity across

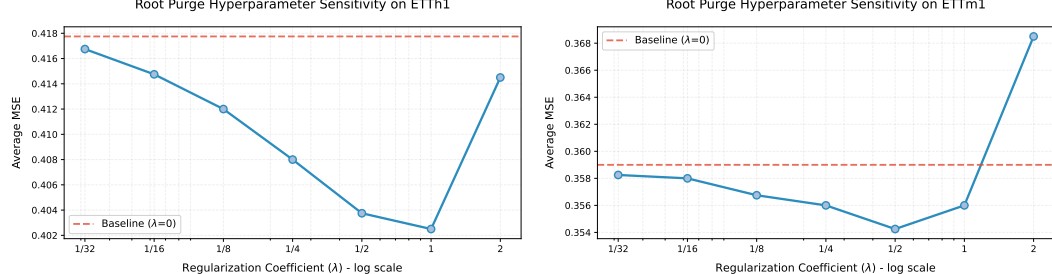

Figure 13: (**Time Domain**) Average forecasting MSE on ETTh1 and ETTm1 across horizons $H = \{96, 192, 336, 720\}$ for different values of $\lambda$. The hyperparameter sensitivity of the time-domain linear model is similar to that of the frequency-domain linear model. As for the frequency-domain linear model, a break-down table for each horizon for the time-domain linear model is in Table 14.

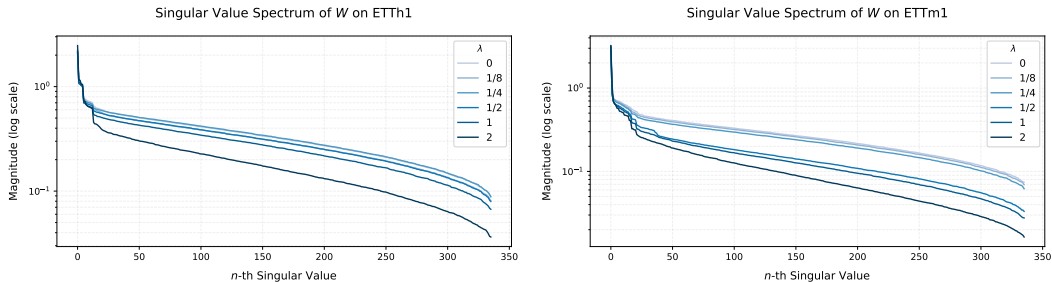

Figure 14: (**Time Domain**) First 336 singular value magnitudes on ETTh1 and ETTm1 under different values of $\lambda$ (log scale) with $\mathbf{W}$. As $\lambda$ increases, Root Purge pushes the weight matrix $\mathbf{W}$ to have more smaller singular values, while the significant singular values remain largely unaffected. The overall effect of $\lambda$ on singular values is consistent with what we found in the main text, where $\mathbf{W}$ is learned in the frequency domain.

Table 13: Root Purge result on time-domain linear model. This shows that Root Purge is also effective in time-domain linear models. The error bars are omitted for simplicity. A full set of results is also shown in Table 15 with error bars.

| Horizon \ Dataset | ETTh1 | ETTh2 | ETTm1 | ETTm2 | Weather | Exchange | ECL | Traffic |
|---|---|---|---|---|---|---|---|---|
| 96 | 0.362 | 0.271 | 0.305 | 0.161 | 0.142 | 0.085 | 0.133 | 0.385 |
| 192 | 0.397 | 0.330 | 0.335 | 0.216 | 0.185 | 0.175 | 0.148 | 0.397 |
| 336 | 0.432 | 0.359 | 0.361 | 0.268 | 0.235 | 0.324 | 0.165 | 0.411 |
| 720 | 0.423 | 0.381 | 0.415 | 0.352 | 0.308 | 0.932 | 0.205 | 0.450 |

forecasting horizons. For instance, $H = 720$ is relatively more sensitive to large regularization, and for smaller $H$, performance tends to be more robust.

Furthermore, since the Root Purge method involves some hyperparameter tuning, we also provide empirical results with $\lambda \in [0.125, 0.25, 0.5]$ across all datasets and aforementioned forecasting horizons. The results are shown in Table 15.

Table 14: Break-down results for Root Purge hyperparameter sensitivity testing on ETTh1 and ETTm1. Overall, the trend for the frequency domain model and the time domain model is similar. While a large range of $\lambda$ will yield improved results, across different forecasting horizons, $H = 720$ is relatively more sensitive to large regularization. For smaller $H$, performance tends to be robust even when a large $\lambda$ is used.

| | Dataset | $H$ | $\lambda$ | | | | | | | |
|---|---|---|---|---|---|---|---|---|---|---|
| | | | 0 | $1/32$ | $1/16$ | $1/8$ | $1/4$ | $1/2$ | 1 | 2 |
| Frequency Domain | ETTh1 | 96 | 0.374 | 0.373 | 0.371 | 0.368 | 0.364 | 0.359 | 0.354 | 0.351 |
| | | 192 | 0.410 | 0.408 | 0.406 | 0.403 | 0.399 | 0.394 | 0.389 | 0.388 |
| | | 336 | 0.431 | 0.431 | 0.430 | 0.427 | 0.425 | 0.423 | 0.423 | 0.424 |
| | | 720 | 0.427 | 0.425 | 0.424 | 0.423 | 0.421 | 0.423 | 0.434 | 0.485 |
| | ETTm1 | 96 | 0.308 | 0.307 | 0.307 | 0.306 | 0.306 | 0.305 | 0.305 | 0.310 |
| | | 192 | 0.337 | 0.336 | 0.336 | 0.335 | 0.334 | 0.333 | 0.333 | 0.338 |
| | | 336 | 0.366 | 0.365 | 0.364 | 0.363 | 0.361 | 0.360 | 0.361 | 0.368 |
| | | 720 | 0.415 | 0.414 | 0.414 | 0.413 | 0.412 | 0.413 | 0.420 | 0.456 |
| Time Domain | ETTh1 | 96 | 0.378 | 0.378 | 0.376 | 0.373 | 0.368 | 0.362 | 0.356 | 0.354 |
| | | 192 | 0.415 | 0.413 | 0.411 | 0.408 | 0.403 | 0.398 | 0.392 | 0.390 |
| | | 336 | 0.443 | 0.442 | 0.440 | 0.438 | 0.435 | 0.432 | 0.430 | 0.430 |
| | | 720 | 0.435 | 0.434 | 0.432 | 0.429 | 0.426 | 0.423 | 0.432 | 0.484 |
| | ETTm1 | 96 | 0.310 | 0.309 | 0.310 | 0.309 | 0.308 | 0.306 | 0.306 | 0.311 |
| | | 192 | 0.340 | 0.340 | 0.339 | 0.337 | 0.337 | 0.335 | 0.335 | 0.339 |
| | | 336 | 0.369 | 0.368 | 0.368 | 0.366 | 0.364 | 0.361 | 0.362 | 0.368 |
| | | 720 | 0.417 | 0.416 | 0.415 | 0.415 | 0.415 | 0.415 | 0.421 | 0.456 |

Table 15: Full experimental results with Root Purge. The hyperparameter sets, following the main experiment, are $\lambda \in [0.125, 0.25, 0.5]$.

| Dataset | Horizon | Frequency Domain Linear | | | Time Domain Linear | | |
|---|---|---|---|---|---|---|---|
| | | $\lambda = 0.125$ | $\lambda = 0.25$ | $\lambda = 0.5$ | $\lambda = 0.125$ | $\lambda = 0.25$ | $\lambda = 0.5$ |
| ETTh1 | 96 | $0.368 \pm 0.000$ | $0.364 \pm 0.000$ | $0.359 \pm 0.000$ | $0.373 \pm 0.002$ | $0.367 \pm 0.002$ | $0.362 \pm 0.000$ |
| | 192 | $0.403 \pm 0.000$ | $0.399 \pm 0.000$ | $0.394 \pm 0.000$ | $0.408 \pm 0.001$ | $0.403 \pm 0.001$ | $0.397 \pm 0.001$ |
| | 336 | $0.427 \pm 0.000$ | $0.425 \pm 0.000$ | $0.423 \pm 0.000$ | $0.438 \pm 0.001$ | $0.435 \pm 0.001$ | $0.432 \pm 0.001$ |
| | 720 | $0.423 \pm 0.000$ | $0.421 \pm 0.000$ | $0.423 \pm 0.000$ | $0.429 \pm 0.001$ | $0.426 \pm 0.001$ | $0.423 \pm 0.000$ |
| ETTh2 | 96 | $0.270 \pm 0.000$ | $0.269 \pm 0.000$ | $0.268 \pm 0.000$ | $0.274 \pm 0.001$ | $0.273 \pm 0.001$ | $0.271 \pm 0.001$ |
| | 192 | $0.330 \pm 0.000$ | $0.329 \pm 0.000$ | $0.328 \pm 0.000$ | $0.333 \pm 0.001$ | $0.332 \pm 0.001$ | $0.330 \pm 0.001$ |
| | 336 | $0.354 \pm 0.000$ | $0.355 \pm 0.000$ | $0.356 \pm 0.000$ | $0.360 \pm 0.002$ | $0.359 \pm 0.001$ | $0.360 \pm 0.001$ |
| | 720 | $0.377 \pm 0.000$ | $0.377 \pm 0.000$ | $0.379 \pm 0.000$ | $0.383 \pm 0.003$ | $0.382 \pm 0.002$ | $0.381 \pm 0.000$ |
| ETTm1 | 96 | $0.306 \pm 0.000$ | $0.306 \pm 0.000$ | $0.305 \pm 0.000$ | $0.309 \pm 0.001$ | $0.308 \pm 0.001$ | $0.305 \pm 0.001$ |
| | 192 | $0.335 \pm 0.000$ | $0.334 \pm 0.000$ | $0.333 \pm 0.000$ | $0.337 \pm 0.001$ | $0.337 \pm 0.001$ | $0.335 \pm 0.002$ |
| | 336 | $0.363 \pm 0.000$ | $0.361 \pm 0.000$ | $0.360 \pm 0.000$ | $0.366 \pm 0.001$ | $0.364 \pm 0.001$ | $0.361 \pm 0.001$ |
| | 720 | $0.413 \pm 0.000$ | $0.412 \pm 0.000$ | $0.413 \pm 0.000$ | $0.415 \pm 0.000$ | $0.415 \pm 0.001$ | $0.415 \pm 0.001$ |
| ETTm2 | 96 | $0.162 \pm 0.000$ | $0.161 \pm 0.000$ | $0.161 \pm 0.000$ | $0.161 \pm 0.000$ | $0.161 \pm 0.001$ | $0.161 \pm 0.001$ |
| | 192 | $0.217 \pm 0.000$ | $0.217 \pm 0.000$ | $0.216 \pm 0.000$ | $0.217 \pm 0.001$ | $0.216 \pm 0.001$ | $0.216 \pm 0.001$ |
| | 336 | $0.269 \pm 0.000$ | $0.269 \pm 0.000$ | $0.269 \pm 0.000$ | $0.269 \pm 0.001$ | $0.268 \pm 0.001$ | $0.268 \pm 0.001$ |
| | 720 | $0.350 \pm 0.000$ | $0.350 \pm 0.000$ | $0.355 \pm 0.000$ | $0.352 \pm 0.001$ | $0.352 \pm 0.001$ | $0.353 \pm 0.001$ |
| Weather | 96 | $0.143 \pm 0.000$ | $0.142 \pm 0.000$ | $0.142 \pm 0.000$ | $0.143 \pm 0.000$ | $0.143 \pm 0.000$ | $0.142 \pm 0.000$ |
| | 192 | $0.187 \pm 0.000$ | $0.186 \pm 0.000$ | $0.186 \pm 0.000$ | $0.186 \pm 0.000$ | $0.185 \pm 0.000$ | $0.185 \pm 0.001$ |
| | 336 | $0.238 \pm 0.000$ | $0.238 \pm 0.000$ | $0.238 \pm 0.000$ | $0.236 \pm 0.000$ | $0.236 \pm 0.001$ | $0.235 \pm 0.001$ |
| | 720 | $0.310 \pm 0.000$ | $0.310 \pm 0.000$ | $0.315 \pm 0.000$ | $0.308 \pm 0.001$ | $0.310 \pm 0.000$ | $0.313 \pm 0.000$ |
| Exchange | 96 | $0.083 \pm 0.000$ | $0.083 \pm 0.000$ | $0.082 \pm 0.000$ | $0.086 \pm 0.000$ | $0.085 \pm 0.001$ | $0.085 \pm 0.000$ |
| | 192 | $0.172 \pm 0.001$ | $0.172 \pm 0.000$ | $0.172 \pm 0.000$ | $0.176 \pm 0.001$ | $0.176 \pm 0.000$ | $0.175 \pm 0.000$ |
| | 336 | $0.324 \pm 0.001$ | $0.324 \pm 0.001$ | $0.324 \pm 0.001$ | $0.325 \pm 0.001$ | $0.324 \pm 0.001$ | $0.324 \pm 0.002$ |
| | 720 | $0.941 \pm 0.002$ | $0.962 \pm 0.002$ | $1.010 \pm 0.006$ | $0.932 \pm 0.009$ | $0.954 \pm 0.008$ | $1.002 \pm 0.009$ |
| ECL | 96 | $0.134 \pm 0.000$ | $0.133 \pm 0.000$ | $0.134 \pm 0.000$ | $0.133 \pm 0.000$ | $0.134 \pm 0.000$ | $0.134 \pm 0.001$ |
| | 192 | $0.148 \pm 0.000$ | $0.148 \pm 0.000$ | $0.149 \pm 0.000$ | $0.149 \pm 0.000$ | $0.148 \pm 0.000$ | $0.149 \pm 0.000$ |
| | 336 | $0.165 \pm 0.000$ | $0.166 \pm 0.000$ | $0.167 \pm 0.000$ | $0.165 \pm 0.000$ | $0.166 \pm 0.000$ | $0.167 \pm 0.000$ |
| | 720 | $0.205 \pm 0.000$ | $0.207 \pm 0.000$ | $0.210 \pm 0.000$ | $0.205 \pm 0.000$ | $0.207 \pm 0.000$ | $0.211 \pm 0.000$ |
| Traffic | 96 | $0.386 \pm 0.000$ | $0.386 \pm 0.000$ | $0.387 \pm 0.000$ | $0.385 \pm 0.000$ | $0.386 \pm 0.000$ | $0.386 \pm 0.000$ |
| | 192 | $0.397 \pm 0.000$ | $0.397 \pm 0.000$ | $0.399 \pm 0.000$ | $0.397 \pm 0.000$ | $0.397 \pm 0.000$ | $0.398 \pm 0.000$ |
| | 336 | $0.411 \pm 0.000$ | $0.412 \pm 0.000$ | $0.416 \pm 0.000$ | $0.411 \pm 0.000$ | $0.413 \pm 0.000$ | $0.416 \pm 0.001$ |
| | 720 | $0.450 \pm 0.000$ | $0.452 \pm 0.000$ | $0.457 \pm 0.000$ | $0.450 \pm 0.000$ | $0.452 \pm 0.000$ | $0.457 \pm 0.000$ |

### E.8 Additional Empirical Results for Rank Reduction Methods

**Performing SVD Directly on W (DWRR)** As mentioned in Section 4, it is also straightforward to directly perform rank reduction on the **W** matrix. In this section, we summarize the results of DWRR in Table 16. The experimental setting of DWRR is exactly the same as RRR. We tune the rank on the validation set, select the top three ranks with the lowest validation MSE, and report the best test MSE achieved using these ranks. Despite the DWRR's strong empirical performance, we argue against its robustness and recommend readers with RRR. An example is shown in Figure 16, with a full set of results at the end of the Appendix. It's obvious that RRR's validation and test curve show much higher consistency.

Table 16: DWRR performance. In comparison to RRR shown in the main text, DWRR can frequently perform better, except for ETTh2, where DWRR shows degradation.

| Horizon \ Dataset | ETTh1 | ETTh2 | ETTm1 | ETTm2 | Weather | Exchange | ECL | Traffic |
|---|---|---|---|---|---|---|---|---|
| 96 | 0.365 | 0.270 | 0.306 | 0.162 | 0.140 | 0.084 | 0.133 | 0.385 |
| 192 | 0.399 | 0.331 | 0.332 | 0.215 | 0.182 | 0.173 | 0.148 | 0.396 |
| 336 | 0.426 | 0.355 | 0.365 | 0.268 | 0.232 | 0.323 | 0.164 | 0.410 |
| 720 | 0.427 | 0.384 | 0.414 | 0.349 | 0.304 | 0.911 | 0.203 | 0.448 |

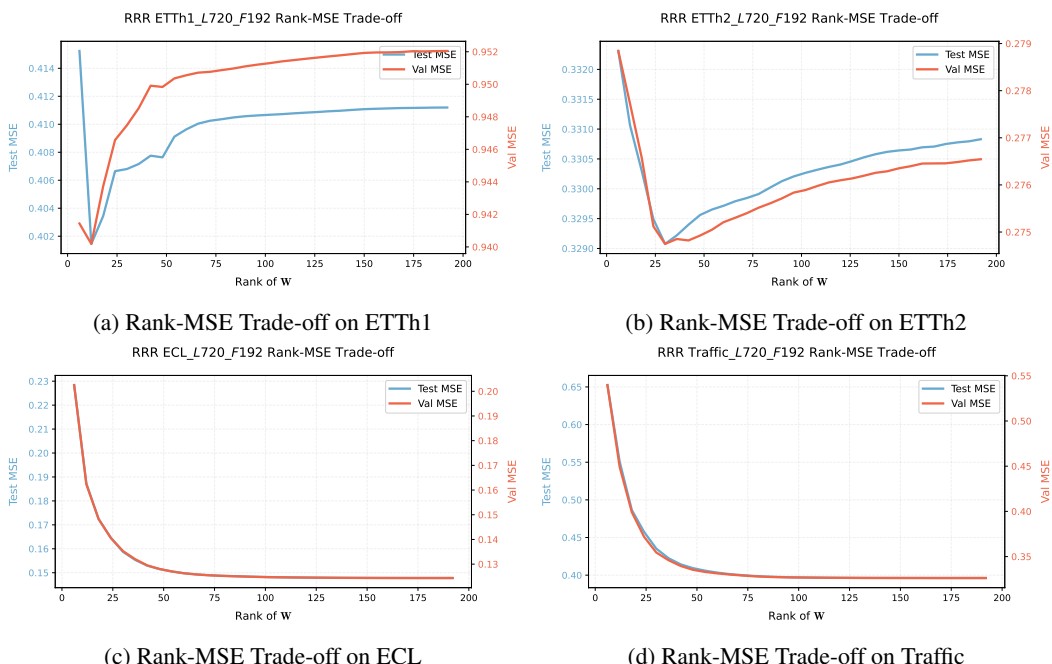

(a) Rank-MSE Trade-off on ETTh1  (b) Rank-MSE Trade-off on ETTh2

(c) Rank-MSE Trade-off on ECL  (d) Rank-MSE Trade-off on Traffic

Figure 15: RRR Rank-MSE Trade-off on 4 example datasets. (a) & (b): On smaller datasets, there are clear benefits in truncating the rank of **W**. (c) & (d): On larger datasets (Electricity, Traffic), the performance curve is robust to the rank of **W**, and a full-rank matrix does not cause significant performance degradation—a result of data scaling, where sufficient data helps suppress spurious roots. However, although the curves for these larger datasets appear flat at this scale, numerical analysis reveals that the optimal model is *not* achieved at full rank, indicating that slight rank truncation remains beneficial despite data scaling.

**RRR & DWRR Rank-MSE Trade-off Curves** While no retraining is needed for RRR and DWRR, both methods incur repetitive evaluation of the model performance on the validation data in order to determine the optimal rank to use for **W**. For a better reading experience, we showcase some representative examples.

In Figure 15, it can be seen that for smaller datasets such as ETTh1 and ETTh2, it is usually beneficial to truncate the rank, suggesting the existence of spurious roots as described in our theoretical analysis.

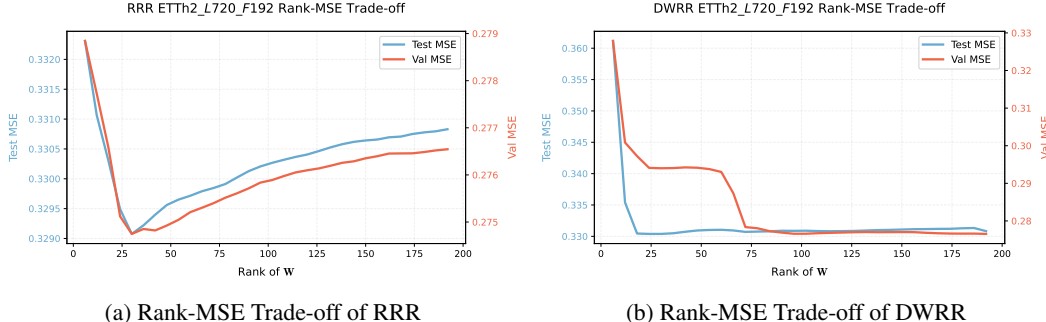

(a) Rank-MSE Trade-off of RRR  (b) Rank-MSE Trade-off of DWRR

Figure 16: RRR Rank-MSE Trade-off compared to DWRR on ETTh2. The validation trade-off curve of RRR highly matches the test trade-off curve. For DWRR, we found that there can be significant gaps for some datasets, and an example (ETTh2) is shown here. Overall, RRR tends to generate more consistent trade-off curves, suggesting better robustness than DWRR.

Conversely, for the larger and more complex Electricity and Traffic datasets, a full-rank $\mathbf{W}$ does not significantly degrade performance, which aligns with the data scaling effect in Proposition 1: sufficient data suppresses spurious roots. That said, our experiments with RRR and DWRR reveal that even for the Electricity and Traffic datasets, the optimal model is *not* at full rank; slight truncation remains beneficial despite data scaling. This indicates that a low-rank structure is relatively common in time series forecasting. Consequently, applying regularization by default can be a generally beneficial practice for time series forecasting tasks.

In Figure 16, we observe that DWRR has less consistent rank-MSE trade-off curves for validation and test datasets. This might suggest DWRR—although showing better performance in many datasets and is inherently computationally cheaper—can be less robust than RRR.

We leave the full plot with all datasets and all forecasting horizons $H = \{96, 192, 336, 720\}$ at the end of the Appendix, in Figures 19, 20, 21, 22, 23, 24, 25, and 26.

### E.9 RRR AND ROOT PURGE ON DIFFERENT LOOKBACK WINDOW/FORECASTING HORIZON

The commonly used lookback windows of previous time series forecasting models are of lengths 96, 192, 336, and 720. We perform a thorough study on the performance of RRR and Root Purge on all of these lookback windows and summarize the results in Table 18. Since we use Root Purge on a frequency-domain linear model in the main experiment, we continue to use these settings in this section. Each experiment for Root Purge, following the setting of the main experiment, is repeated 5 times. Overall, there is a clear trend that with a longer lookback window, the performance of both RRR and Root Purge improves. Still, the performance of RRR and Root Purge remains competitive even when the lookback window length is set to shorter values.

It is important to notice that reducing the lookback window is a much more inefficient method in characteristic root selection than RRR and Root Purge. For instance, for ETTh1, as shown in Appendix E.8, the optimal rank obtained by RRR is much less than 96, and decreasing the rank will result in improvements. However, if we use a lookback window of 96 directly to keep our root set small, the result is much worse.

In addition, we notice that previous works (Liu et al., 2023; Zhang et al., 2025) has also been using the forecasting horizon of 48 and 144. Thus, we summary RRR and Root Purge performances for these forecasting tasks in Table 17 for better comparing with methods using these settings.

Table 17: Performance of Root Purge and RRR models at forecasting horizon of 48 and 144. The lookback window is still set to 720, aligning with the setting in our main experiments.

| Dataset | ETTh1 | | ETTh2 | | ETTm1 | | ETTm2 | | Weather | | Exchange | | Traffic | | ECL | |
|---|---|---|---|---|---|---|---|---|---|---|---|---|---|---|---|---|
| Horizon | 48 | 144 | 48 | 144 | 48 | 144 | 48 | 144 | 48 | 144 | 48 | 144 | 48 | 144 | 48 | 144 |
| Root Purge | 0.332 | 0.379 | 0.213 | 0.301 | 0.282 | 0.325 | 0.124 | 0.192 | 0.111 | 0.165 | 0.042 | 0.124 | 0.369 | 0.391 | 0.118 | 0.143 |
| RRR | 0.344 | 0.386 | 0.215 | 0.302 | 0.280 | 0.326 | 0.123 | 0.192 | 0.110 | 0.162 | 0.044 | 0.127 | 0.368 | 0.391 | 0.117 | 0.142 |

Table 18: Impact of lookback window length $L$ on the performance of RRR and Root Purge. Given that the lookback window length directly impacts the number of characteristic roots of the corresponding linear difference equation, the lookback window alone can serve as a strong regularization. Thus, we expect RRR and Root Purge to perform better with higher lookback windows. The error bars are omitted for RRR as it produces static results.

| Dataset | Horizon | RRR | | | | Root Purge (Frequency Domain) | | | |
|---|---|---|---|---|---|---|---|---|---|
| | | $L = 96$ | $L = 192$ | $L = 336$ | $L = 720$ | $L = 96$ | $L = 192$ | $L = 336$ | $L = 720$ |
| ETTh1 | 96 | 0.383 | 0.376 | 0.369 | 0.367 | $0.383 \pm 0.000$ | $0.373 \pm 0.000$ | $0.364 \pm 0.000$ | $0.359 \pm 0.000$ |
| | 192 | 0.433 | 0.420 | 0.402 | 0.401 | $0.433 \pm 0.001$ | $0.419 \pm 0.000$ | $0.397 \pm 0.000$ | $0.394 \pm 0.000$ |
| | 336 | 0.474 | 0.448 | 0.424 | 0.430 | $0.473 \pm 0.000$ | $0.446 \pm 0.000$ | $0.421 \pm 0.000$ | $0.423 \pm 0.000$ |
| | 720 | 0.461 | 0.431 | 0.418 | 0.425 | $0.461 \pm 0.003$ | $0.433 \pm 0.000$ | $0.421 \pm 0.001$ | $0.421 \pm 0.000$ |
| ETTh2 | 96 | 0.289 | 0.282 | 0.272 | 0.268 | $0.293 \pm 0.001$ | $0.288 \pm 0.000$ | $0.277 \pm 0.000$ | $0.268 \pm 0.000$ |
| | 192 | 0.374 | 0.354 | 0.334 | 0.329 | $0.378 \pm 0.000$ | $0.360 \pm 0.000$ | $0.336 \pm 0.000$ | $0.328 \pm 0.000$ |
| | 336 | 0.414 | 0.382 | 0.358 | 0.352 | $0.416 \pm 0.000$ | $0.386 \pm 0.000$ | $0.355 \pm 0.000$ | $0.355 \pm 0.000$ |
| | 720 | 0.416 | 0.399 | 0.384 | 0.376 | $0.419 \pm 0.001$ | $0.403 \pm 0.000$ | $0.387 \pm 0.000$ | $0.377 \pm 0.000$ |
| ETTm1 | 96 | 0.351 | 0.308 | 0.301 | 0.306 | $0.359 \pm 0.000$ | $0.311 \pm 0.000$ | $0.300 \pm 0.000$ | $0.305 \pm 0.000$ |
| | 192 | 0.389 | 0.343 | 0.336 | 0.336 | $0.396 \pm 0.000$ | $0.346 \pm 0.000$ | $0.336 \pm 0.000$ | $0.333 \pm 0.000$ |
| | 336 | 0.422 | 0.380 | 0.370 | 0.365 | $0.428 \pm 0.000$ | $0.381 \pm 0.000$ | $0.370 \pm 0.000$ | $0.360 \pm 0.000$ |
| | 720 | 0.483 | 0.439 | 0.426 | 0.414 | $0.487 \pm 0.000$ | $0.439 \pm 0.000$ | $0.425 \pm 0.000$ | $0.412 \pm 0.000$ |
| ETTm2 | 96 | 0.182 | 0.172 | 0.165 | 0.161 | $0.184 \pm 0.000$ | $0.174 \pm 0.000$ | $0.165 \pm 0.000$ | $0.161 \pm 0.000$ |
| | 192 | 0.246 | 0.231 | 0.220 | 0.216 | $0.248 \pm 0.000$ | $0.233 \pm 0.000$ | $0.221 \pm 0.000$ | $0.216 \pm 0.000$ |
| | 336 | 0.307 | 0.284 | 0.273 | 0.268 | $0.307 \pm 0.000$ | $0.285 \pm 0.000$ | $0.274 \pm 0.000$ | $0.269 \pm 0.000$ |
| | 720 | 0.407 | 0.382 | 0.367 | 0.348 | $0.408 \pm 0.000$ | $0.384 \pm 0.000$ | $0.368 \pm 0.000$ | $0.350 \pm 0.000$ |
| Weather | 96 | 0.164 | 0.150 | 0.143 | 0.140 | $0.168 \pm 0.000$ | $0.155 \pm 0.000$ | $0.146 \pm 0.000$ | $0.142 \pm 0.000$ |
| | 192 | 0.211 | 0.194 | 0.186 | 0.182 | $0.217 \pm 0.000$ | $0.202 \pm 0.000$ | $0.191 \pm 0.000$ | $0.186 \pm 0.000$ |
| | 336 | 0.267 | 0.248 | 0.239 | 0.232 | $0.274 \pm 0.000$ | $0.257 \pm 0.000$ | $0.246 \pm 0.000$ | $0.238 \pm 0.000$ |
| | 720 | 0.348 | 0.328 | 0.316 | 0.304 | $0.353 \pm 0.000$ | $0.334 \pm 0.000$ | $0.321 \pm 0.000$ | $0.310 \pm 0.000$ |
| Exchange | 96 | 0.082 | 0.082 | 0.083 | 0.084 | $0.082 \pm 0.000$ | $0.083 \pm 0.000$ | $0.083 \pm 0.000$ | $0.082 \pm 0.000$ |
| | 192 | 0.169 | 0.170 | 0.171 | 0.174 | $0.170 \pm 0.000$ | $0.173 \pm 0.000$ | $0.171 \pm 0.000$ | $0.172 \pm 0.001$ |
| | 336 | 0.311 | 0.312 | 0.312 | 0.324 | $0.313 \pm 0.000$ | $0.319 \pm 0.000$ | $0.321 \pm 0.001$ | $0.324 \pm 0.001$ |
| | 720 | 0.813 | 0.816 | 0.843 | 0.915 | $0.821 \pm 0.000$ | $0.829 \pm 0.000$ | $0.892 \pm 0.004$ | $0.941 \pm 0.002$ |
| Average | | 0.350 | 0.330 | **0.321** | **0.321** | 0.353 | 0.334 | **0.325** | **0.322** |

## E.10 SHORT-TERM FORECASTING–AN EXTREME CASE

The performance of linear models on the M4 dataset is often limited (Xu et al., 2023; Liu et al., 2023; Wu et al., 2023). Our theoretical analysis in Section 3 and Appendix C attributes this primarily to two factors:

1. The restricted lookback window sizes, which range from only 12 to 96.

2. The requirement for a single model to fit a large number of time series from diverse domains, each with potentially different dynamics.

For example, fitting a linear model to the M4 yearly data is equivalent to capturing *all* dynamics across 23,000 different series using *only* 12 characteristic roots. The severe constraint on the number of roots, combined with the diversity of dynamics to be captured, implies that a linear model will significantly *underfit* the M4 dataset, explaining its subpar performance.

Root Purge and RRR leverage rank and nullity manipulation to eliminate spurious roots. However, when a linear model is already significantly underfitting, no spurious roots are present. Consequently, we expect little to no performance gain from applying Root Purge or RRR to linear models on M4. On the other hand, our approaches—especially Root Purge—do not over-regularize the model even in these extreme cases (as shown in Table 19), suggesting their potential as a default regularizer for linear time series forecasting models.

## E.11 HIGHER-ORDER ROOT PURGE

The root purge we considered in the main text has the following target, where $\mathcal{G}_{\mathbf{W}}(\cdot)$ defines a linear transformation:

$$\min_{\mathbf{W}} \underbrace{\|\mathbf{Y}_{\text{fut}} - \mathcal{G}_{\mathbf{W}}(\mathbf{Y}_{\text{his}})\|_F^2}_{\text{root-seeking}} + \lambda \underbrace{\|\mathcal{G}_{\mathbf{W}} \circ \mathcal{P}\left(\mathbf{Y}_{\text{fut}} - \mathcal{G}_{\mathbf{W}}(\mathbf{Y}_{\text{his}})\right)\|_F^2}_{\text{root-purging}}$$

We now define the *order* of root purge as 1, forcing the noise to be contained in the null-space of $\mathcal{G}_{\mathbf{W}}(\cdot)$. To extend from order-1 Root Purge, it seems also reasonable to perform Root Purge on

Table 19: Forecasting performance on the M4 benchmark for linear models with and without Root Purge. The suboptimal performance of the baseline linear model is consistent with our theoretical observations in Section 3, a consequence of its limited capacity. Crucially, these results demonstrate the robustness of Root Purge. Even in these extreme cases of underfitting, the regularizer maintains stability and introduces negligible negative effects, affirming its utility as a default regularization.

| Metric | M4 Subset | Frequency Domain Linear Model | | Time Domain Linear Model | |
|---|---|---|---|---|---|
| | | Baseline | + Root Purge | Baseline | + Root Purge |
| sMAPE | Yearly | 14.914 | 14.911 | 14.871 | 14.880 |
| | Quarterly | 10.977 | 10.977 | 10.860 | 10.860 |
| | Monthly | 13.402 | 13.404 | 13.354 | 13.351 |
| | Others | 4.861 | 4.856 | 4.913 | 4.911 |
| | Average | 12.742 | 12.741 | 12.682 | 12.683 |
| MASE | Yearly | 3.170 | 3.171 | 3.044 | 3.044 |
| | Quarterly | 1.349 | 1.347 | 1.322 | 1.323 |
| | Monthly | 1.022 | 1.022 | 1.010 | 1.010 |
| | Others | 3.313 | 3.313 | 3.343 | 3.344 |
| | Average | 1.709 | 1.709 | 1.670 | 1.670 |
| OWA | Yearly | 0.855 | 0.855 | 0.838 | 0.838 |
| | Quarterly | 0.990 | 0.990 | 0.975 | 0.975 |
| | Monthly | 0.945 | 0.945 | 0.938 | 0.938 |
| | Others | 1.034 | 1.033 | 1.044 | 1.044 |
| | Average | 0.916 | 0.916 | 0.904 | 0.904 |

higher-order composition of $\mathcal{G}_{\mathbf{W}} \circ \mathcal{P}(\cdot)$, as follows:

$$\left(\mathcal{G}_{\mathbf{W}} \circ \mathcal{P}\right)^{k}(\cdot) = \underbrace{\mathcal{G}_{\mathbf{W}} \circ \mathcal{P}(\mathcal{G}_{\mathbf{W}} \circ \mathcal{P}(\cdots \mathcal{G}_{\mathbf{W}} \circ \mathcal{P}(\cdot)))}_{k \text{ such composition}}$$

In this section, we investigate this general form of Root Purge and conclude that Root Purge of order 1 works best. As summarized in Table 20, higher-order Root Purge has attenuated effects on regularization, and the performance tends to degrade as order increases. The order-1 Root Purge is able to attain the best result in almost all of the benchmarks. To rule out run-to-run variance, we repeat each experiment 5 times and report the mean with standard deviations in our table.

Table 20: Root Purge results across different orders. The best results are highlighted in **red**.

| Dataset | $H$ | Order-1 Root Purge (Original) | Order-2 Root Purge | Order-3 Root Purge | Order-4 Root Purge |
|---|---|---|---|---|---|
| ETTh1 | 96 | **0.359** ± 0.000 | 0.365 ± 0.000 | 0.371 ± 0.000 | 0.373 ± 0.000 |
| | 192 | **0.394** ± 0.000 | 0.401 ± 0.000 | 0.406 ± 0.000 | 0.408 ± 0.000 |
| | 336 | **0.423** ± 0.000 | 0.431 ± 0.000 | 0.432 ± 0.000 | 0.432 ± 0.000 |
| | 720 | **0.421** ± 0.000 | 0.427 ± 0.000 | 0.428 ± 0.000 | 0.428 ± 0.000 |
| ETTh2 | 96 | **0.268** ± 0.000 | 0.270 ± 0.000 | 0.271 ± 0.000 | 0.271 ± 0.000 |
| | 192 | **0.328** ± 0.000 | 0.330 ± 0.000 | 0.331 ± 0.000 | 0.331 ± 0.000 |
| | 336 | 0.355 ± 0.000 | **0.354** ± 0.000 | 0.354 ± 0.000 | 0.354 ± 0.000 |
| | 720 | **0.377** ± 0.000 | 0.377 ± 0.000 | 0.377 ± 0.000 | 0.377 ± 0.000 |
| ETTm1 | 96 | **0.305** ± 0.000 | 0.306 ± 0.000 | 0.307 ± 0.000 | 0.308 ± 0.000 |
| | 192 | **0.333** ± 0.000 | 0.335 ± 0.000 | 0.336 ± 0.000 | 0.337 ± 0.000 |
| | 336 | **0.360** ± 0.000 | 0.361 ± 0.000 | 0.362 ± 0.000 | 0.364 ± 0.000 |
| | 720 | **0.412** ± 0.000 | 0.414 ± 0.000 | 0.415 ± 0.000 | 0.415 ± 0.000 |
| ETTm2 | 96 | **0.161** ± 0.000 | 0.161 ± 0.000 | 0.163 ± 0.000 | 0.163 ± 0.000 |
| | 192 | **0.216** ± 0.000 | 0.217 ± 0.000 | 0.217 ± 0.000 | 0.217 ± 0.000 |
| | 336 | **0.269** ± 0.000 | 0.269 ± 0.000 | 0.269 ± 0.000 | 0.269 ± 0.000 |
| | 720 | **0.350** ± 0.000 | 0.350 ± 0.000 | 0.350 ± 0.000 | 0.350 ± 0.000 |
| Weather | 96 | **0.142** ± 0.000 | 0.143 ± 0.000 | 0.143 ± 0.000 | 0.144 ± 0.000 |
| | 192 | **0.186** ± 0.000 | 0.187 ± 0.000 | 0.188 ± 0.000 | 0.188 ± 0.000 |
| | 336 | **0.238** ± 0.000 | 0.238 ± 0.000 | 0.238 ± 0.000 | 0.238 ± 0.000 |
| | 720 | 0.310 ± 0.000 | **0.309** ± 0.000 | 0.309 ± 0.000 | 0.309 ± 0.000 |
| Exchange | 96 | **0.082** ± 0.000 | 0.083 ± 0.000 | 0.084 ± 0.000 | 0.087 ± 0.002 |
| | 192 | **0.172** ± 0.001 | 0.173 ± 0.000 | 0.174 ± 0.000 | 0.174 ± 0.000 |
| | 336 | **0.324** ± 0.001 | 0.324 ± 0.001 | 0.325 ± 0.001 | 0.326 ± 0.001 |
| | 720 | 0.941 ± 0.002 | 0.928 ± 0.001 | **0.927** ± 0.001 | 0.934 ± 0.004 |

### E.12 ADDITIONAL RESULTS ON DATA SCALING & NOISE ROBUSTNESS ON SIMULATION DATA

We provide additional visualization for the experiment in Section 5.3 - Data Scaling & Noise Robustness, where we examine the data scaling property and noise robustness of our methods on a toy example. In Figure 17a, we show a qualitative result. When there is a scarce amount of data, the root-seeking target drives the linear model to overfit and mimic patterns of noise, which are, in fact, unpredictable. In Figure 17b, we show the noise robustness curve based on the raw MSE we obtained. It is still visible that both RRR and Root Purge are closest to the ground truth. A clearer relative degradation plot in noise scaling is provided in Section 5.

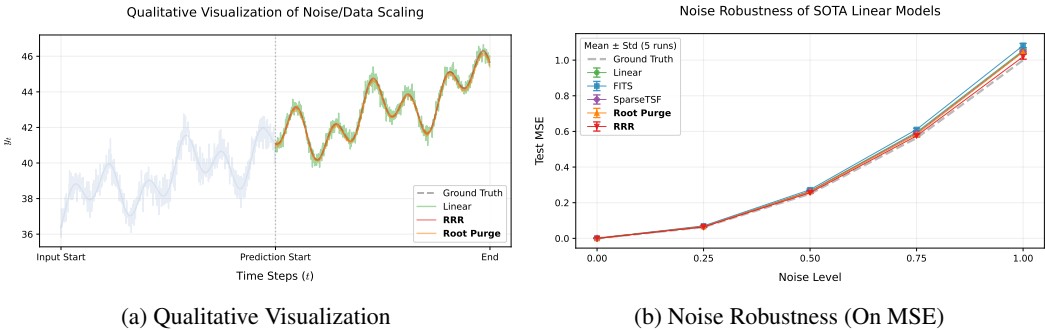

(a) Qualitative Visualization          (b) Noise Robustness (On MSE)

Figure 17: Additional visualization of the data scaling & noise robustness on simulation data. (a) A qualitative visualization of what went wrong for pure root seeking when data is scarce. (b) Test MSE against noise level. It is still visible that RRR and Root Purge show better noise robustness, but a clearer degradation plot in noise scaling is provided in Figure 3.

### E.13 DETAILS OF ROOT ANALYSIS ON SIMULATION DATA AND QUANTITATIVE EVALUATION

As mentioned in the main text, a direct root comparison is infeasible with real-world data due to the lack of access to the underlying noise-free dynamics and their true characteristic roots. We employ the same synthetic dynamics used previously to illustrate the data scaling and noise robustness properties of our methods.

$$y(t) = \sin(2t) + \cos(5t) + 0.5t + \sigma \cdot \varepsilon_t$$

The additive Gaussian noise term is given by $\sigma \cdot \varepsilon_t$. In the noise-free case ($\sigma = 0$), the signal can be represented exactly using six characteristic roots. We fit a linear model with a lookback window and a forecasting horizon each of length 25—this limits the total number of learned roots and offers sufficient capacity to capture the noise-free dynamics *exactly*. We then compare the characteristic roots estimated under four conditions against the ground truth roots.

■ **Ground Truth (noise-free):**
A linear model is fitted to the clean signal via SVD, and its characteristic roots serve as the reference.

1. **Standard Linear Model + Noise:**
A standard linear model is fitted using Ordinary Least Squares (OLS) on the noisy signal, and the root distribution from the first and the twelfth forecasting head is studied.

2. **Root Purge + Noise:**
Our Root Purge method is applied to the same noisy data, and the learned roots are examined.

3. **Rank Reduction + Noise:**
Similarly, our Rank Reduction Regression (RRR) approach is applied. The learned roots are examined.

4. **Standard Linear Model + Gradient Method:**
To control for the optimizer, we include a baseline where a standard linear model is optimized using the gradient method (matching the optimizer used in Root Purge).

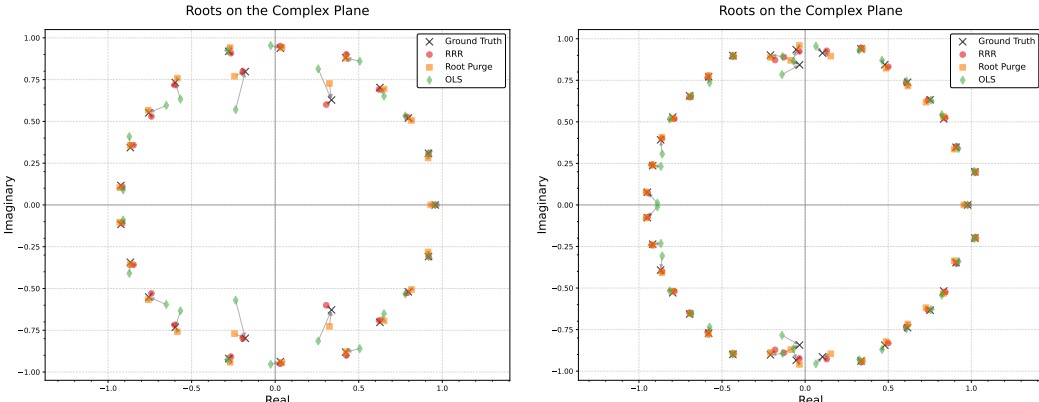

(a) Root Distribution on the 1st forecasting horizon     (b) Root Distribution on 12th forecasting horizon

Figure 18: Root distribution on the (a) 1st and (b) 12th linear time series forecasting horizons, as defined by Section 3, Equation (2). Both RRR and Root Purge yield root distributions visibly closer to the noise-free ground truth than the standard linear model. The arrows denote the ground truth roots toward which the respective methods should converge.

In section 5.3, we see that roots obtained via RRR and Root Purge are closer to the ground truth than those from the standard model. Here, we qualitatively show the root recovery performance in Figure 18 by visualizing the characteristic roots from these models in the two-dimensional complex plane. The roots obtained via RRR and Root Purge are visibly closer to the ground truth than those from the standard model.

Additionally, since Root Purge is optimized with gradient method, we also introduced "Standard Linear Model (gradient method)" as a further baseline in Table 21 to investigate the impact of using gradient based-optimizations.

Table 21: Comparison of Root Distance to Ground Truth across models. The results demonstrate the effectiveness of Rank Reduction and Root Purge in recovering roots closer to the true values compared to standard linear models.

| Model | Root Distance to Ground Truth (mean $\pm$ std) |
|---|---|
| Rank Reduction | $\mathbf{0.036 \pm 0.014}$ |
| Root Purge | $\mathbf{0.045 \pm 0.009}$ |
| Standard Linear Model (OLS) | $0.064 \pm 0.025$ |
| Standard Linear Model (gradient method) | $0.151 \pm 0.045$ |

Our analysis yields the following key observations:

- Both RRR and Root Purge yield root distributions much closer to the noise-free ground truth than the standard linear model.

- The root distributions from the standard model exhibit high variance across runs, whereas our proposed methods produce more stable and concentrated distributions.

- Although the standard model optimized with the gradient method performs worse than its OLS counterpart, Root Purge successfully overcomes this optimizer disadvantage to achieve a smaller root distance.

These results align with the results in Section 5.3 and demonstrate that our proposed techniques enhance the robustness of root learning in the presence of noise.

### E.14    STOP-GRADIENT AND FULL-GRADIENT ROOT PURGE

In Appendix D.7, we analyzed the properties of the two variants of Root Purge from a theoretical perspective. Our results indicate that their theoretical behavior should be highly similar. In this

section, we empirically compare the full-gradient version with the stop-gradient version. The results, summarized in Table 22 for the ETT datasets, show that both variants yield very similar performance. Therefore, we primarily use the stop-gradient version in our experiments, as it greatly simplifies gradient computation and is significantly less memory-intensive than the full-gradient version.

Table 22: Experimental results comparing the stop-gradient and full-gradient versions of Root Purge on ETT datasets with different $\lambda$ values. The stop-gradient variant produces nearly identical performance while being more efficient.

| Dataset | Horizon | Stop Gradient Version | | | Full Gradient Version | | |
|---|---|---|---|---|---|---|---|
| | | $\lambda = 0.125$ | $\lambda = 0.25$ | $\lambda = 0.5$ | $\lambda = 0.125$ | $\lambda = 0.25$ | $\lambda = 0.5$ |
| ETTh1 | 96 | $0.368 \pm 0.000$ | $0.364 \pm 0.000$ | $0.359 \pm 0.000$ | $0.368 \pm 0.000$ | $0.364 \pm 0.000$ | $0.359 \pm 0.000$ |
| | 192 | $0.403 \pm 0.000$ | $0.399 \pm 0.000$ | $0.394 \pm 0.000$ | $0.403 \pm 0.000$ | $0.399 \pm 0.000$ | $0.394 \pm 0.000$ |
| | 336 | $0.427 \pm 0.000$ | $0.425 \pm 0.000$ | $0.423 \pm 0.000$ | $0.427 \pm 0.000$ | $0.425 \pm 0.001$ | $0.423 \pm 0.000$ |
| | 720 | $0.423 \pm 0.000$ | $0.421 \pm 0.000$ | $0.423 \pm 0.000$ | $0.423 \pm 0.000$ | $0.421 \pm 0.000$ | $0.421 \pm 0.001$ |
| ETTh2 | 96 | $0.270 \pm 0.000$ | $0.269 \pm 0.000$ | $0.268 \pm 0.000$ | $0.270 \pm 0.000$ | $0.269 \pm 0.000$ | $0.268 \pm 0.000$ |
| | 192 | $0.330 \pm 0.000$ | $0.329 \pm 0.000$ | $0.328 \pm 0.000$ | $0.330 \pm 0.000$ | $0.329 \pm 0.000$ | $0.328 \pm 0.000$ |
| | 336 | $0.354 \pm 0.000$ | $0.355 \pm 0.000$ | $0.356 \pm 0.000$ | $0.354 \pm 0.000$ | $0.355 \pm 0.000$ | $0.356 \pm 0.001$ |
| | 720 | $0.377 \pm 0.000$ | $0.377 \pm 0.000$ | $0.379 \pm 0.000$ | $0.377 \pm 0.000$ | $0.377 \pm 0.000$ | $0.378 \pm 0.000$ |
| ETTm1 | 96 | $0.306 \pm 0.000$ | $0.306 \pm 0.000$ | $0.305 \pm 0.000$ | $0.306 \pm 0.001$ | $0.305 \pm 0.001$ | $0.305 \pm 0.001$ |
| | 192 | $0.335 \pm 0.000$ | $0.334 \pm 0.000$ | $0.333 \pm 0.000$ | $0.335 \pm 0.001$ | $0.334 \pm 0.000$ | $0.333 \pm 0.000$ |
| | 336 | $0.363 \pm 0.000$ | $0.361 \pm 0.000$ | $0.360 \pm 0.000$ | $0.362 \pm 0.001$ | $0.361 \pm 0.000$ | $0.360 \pm 0.000$ |
| | 720 | $0.413 \pm 0.000$ | $0.412 \pm 0.000$ | $0.413 \pm 0.000$ | $0.413 \pm 0.000$ | $0.412 \pm 0.000$ | $0.412 \pm 0.000$ |
| ETTm2 | 96 | $0.162 \pm 0.000$ | $0.161 \pm 0.000$ | $0.161 \pm 0.000$ | $0.161 \pm 0.000$ | $0.161 \pm 0.000$ | $0.161 \pm 0.001$ |
| | 192 | $0.217 \pm 0.000$ | $0.217 \pm 0.000$ | $0.216 \pm 0.000$ | $0.216 \pm 0.000$ | $0.216 \pm 0.000$ | $0.216 \pm 0.000$ |
| | 336 | $0.269 \pm 0.000$ | $0.269 \pm 0.000$ | $0.269 \pm 0.000$ | $0.269 \pm 0.000$ | $0.269 \pm 0.001$ | $0.269 \pm 0.000$ |
| | 720 | $0.350 \pm 0.000$ | $0.350 \pm 0.000$ | $0.355 \pm 0.000$ | $0.350 \pm 0.000$ | $0.351 \pm 0.000$ | $0.354 \pm 0.000$ |

# F  DISCUSSION

## F.1  LIMITATION AND FUTURE WORK

While our analysis provides valuable insights into linear models for time series forecasting, a key limitation is their inability to fully capture nonlinear dynamics present in many real-world signals. Although linear systems offer strong interpretability and robustness, their expressiveness is inherently constrained, especially when the lookback window is limited to a small number (e.g. in short-term forecasting tasks such as M4; see Appendix E.10 for details). To allow a linear model to capture complex dynamics, the lookback window must be sufficiently long. A promising direction for future work is to extend our framework toward nonlinear dynamics, investigating whether similar core principles—such as dominant modes or root structures—can be identified and generalized. This could inspire the design of more interpretable and theoretically grounded nonlinear models. Additionally, our focus has been on forecasting tasks; it would be interesting to explore the implications of characteristic root structures in classification settings. In particular, understanding how random projections may act as a form of null space learning—both geometrically as classifiers and algebraically as approximations of SVD—could provide a new perspective on the design and regularization of time-series classifiers.

## F.2  BROADER IMPACT

The broader impact of this work lies in its potential to reshape how time series forecasting is approached in both research and applied settings. By grounding model design in classical linear systems theory—specifically, through the lens of characteristic roots—this study offers a rigorous foundation for building more interpretable and robust forecasting models. The insights into noise sensitivity and data-scaling inefficiency can inform the development of lightweight, resource-efficient alternatives to overparameterized deep models, making high-quality forecasting more accessible in low-data or low-compute environments. Moreover, the proposed root-centric regularization techniques may be extended to other domains involving dynamical systems, such as control, finance, and climate modeling. Ultimately, this work invites the time series and broader AI communities to rethink the foundational principles guiding model design. Rather than relying on increasingly intricate architectures or large-scale pretraining, we advocate for theory-driven approaches that focus on capturing the essential dynamics of time series data. Our study demonstrates that a deeper understanding of fundamental structures—such as characteristic roots—can lead to models that are

not only more interpretable and robust, but also more efficient and generalizable. By highlighting the power of simple, principled modeling grounded in classical system theory, we hope to shift the focus toward uncovering the core mechanisms that govern temporal behavior. This perspective encourages the community to pursue models that align with the inherent structure of time series, fostering progress through insight rather than complexity.

## G  DECLARATION

We used large language models (LLMs) to polish writing.

**Full set of rank-MSE trade-off curves for RRR and DWRR in the following pages.** $\longrightarrow$

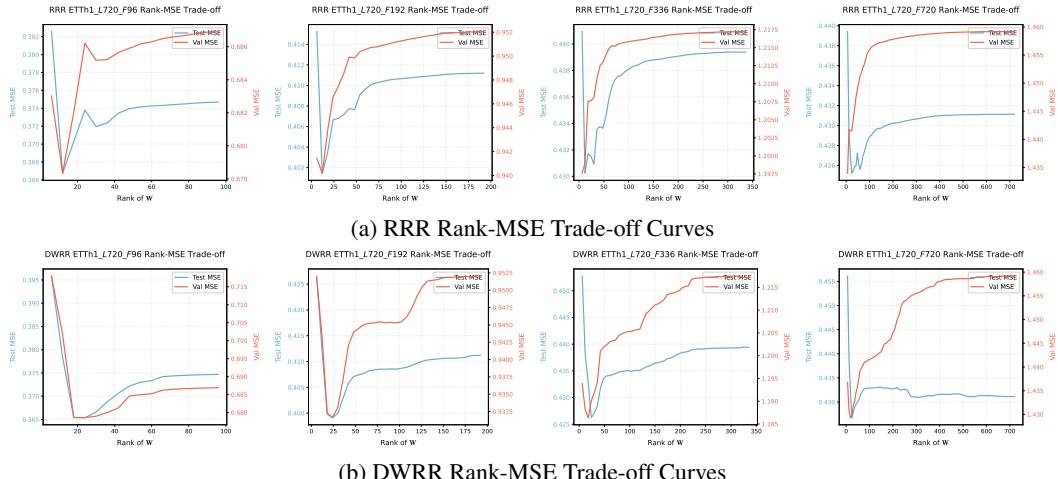

Figure 19: Rank-MSE Trade-off Curves on ETTh1

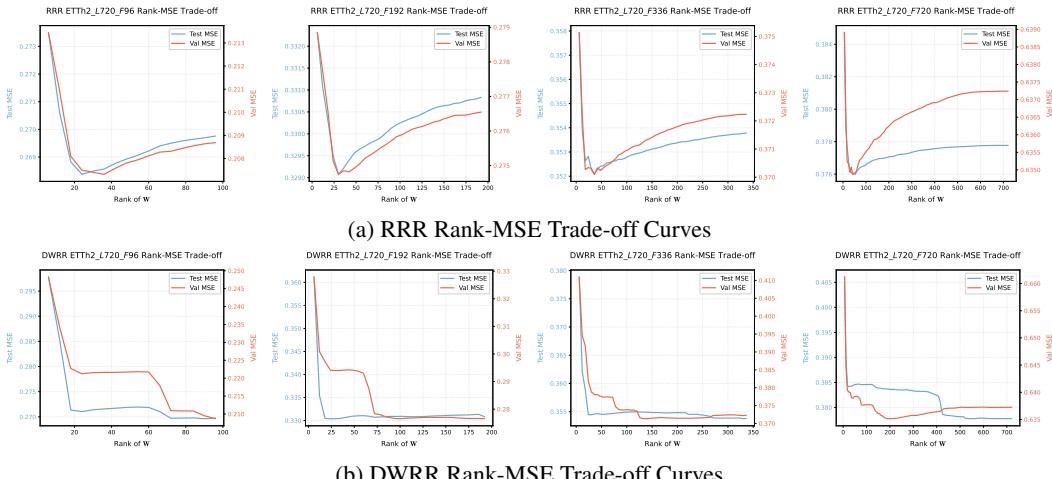

Figure 20: Rank-MSE Trade-off Curves on ETTh2

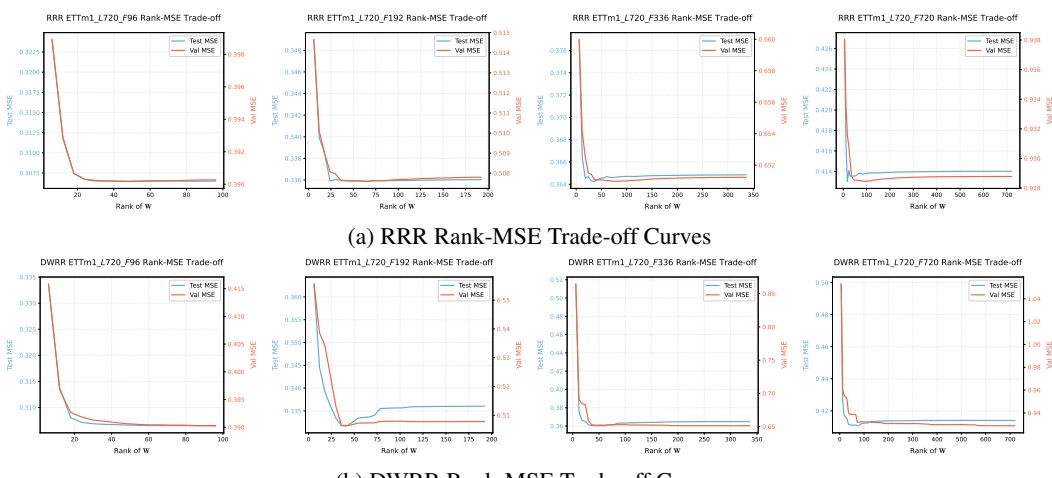

Figure 21: Rank-MSE Trade-off Curves on ETTm1

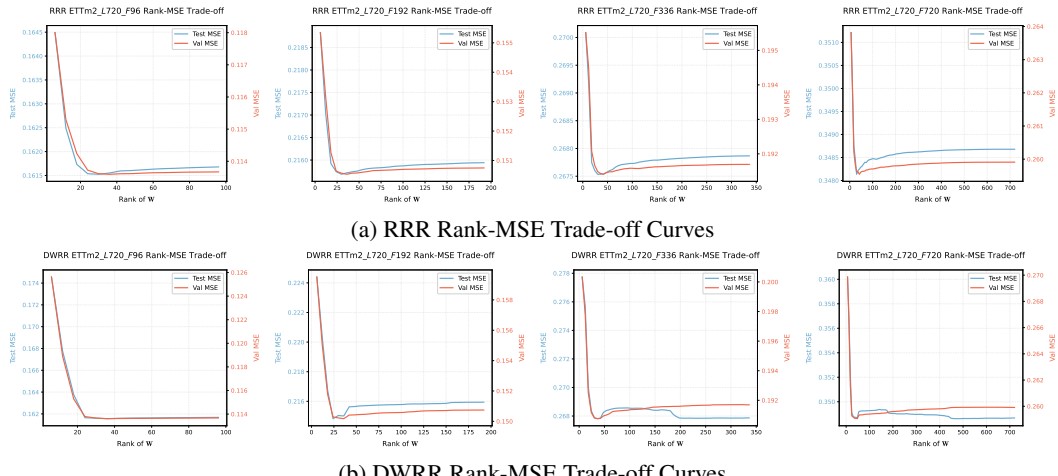

Figure 22: Rank-MSE Trade-off Curves on ETTm2

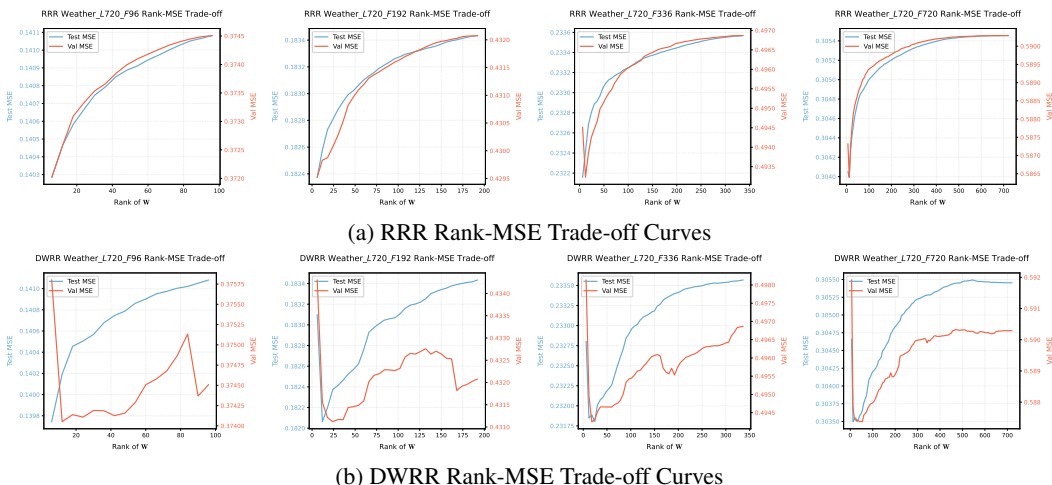

Figure 23: Rank-MSE Trade-off Curves on Weather

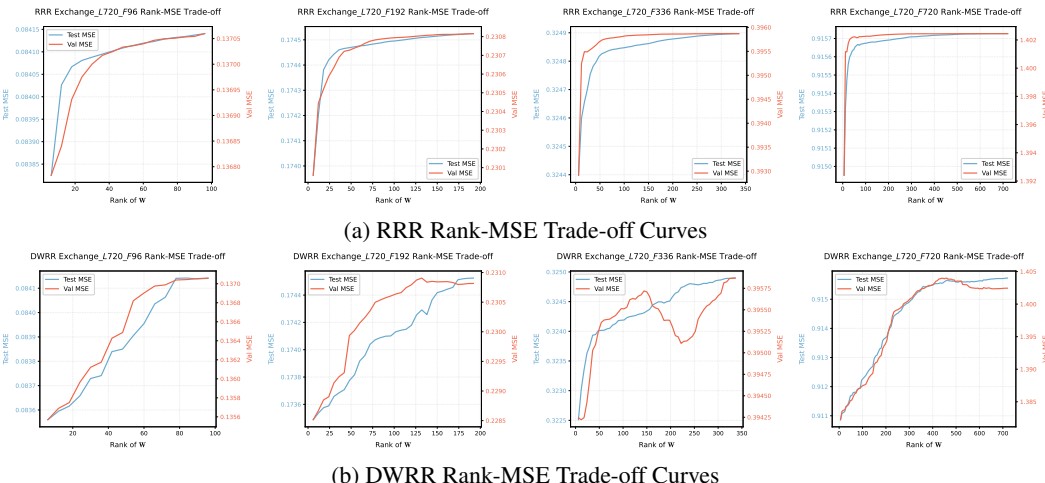

Figure 24: Rank-MSE Trade-off Curves on Exchange

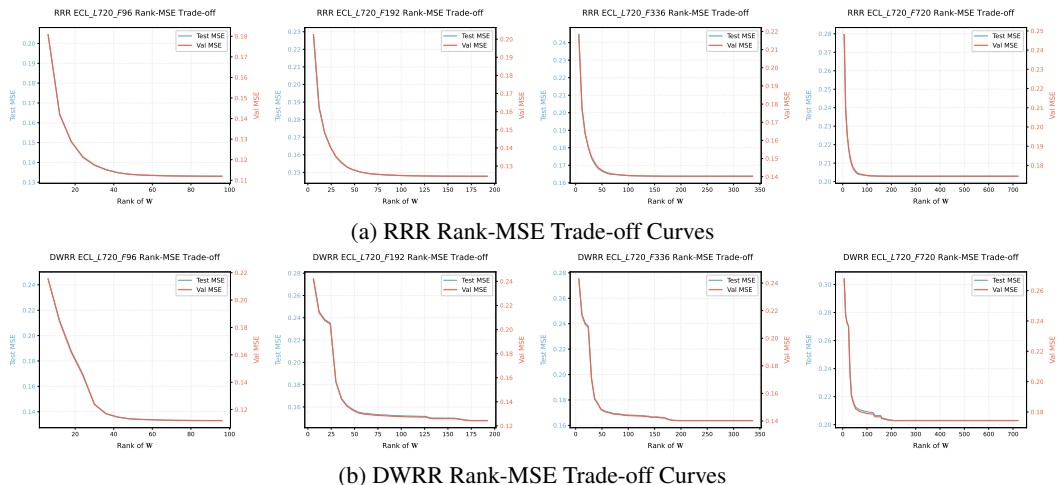

Figure 25: Rank-MSE Trade-off Curves on Electricity. Although the curves for these larger datasets appear flat at this scale, numerical analysis reveals that the optimal model is not achieved at full rank, indicating that slight rank truncation remains beneficial despite data scaling.

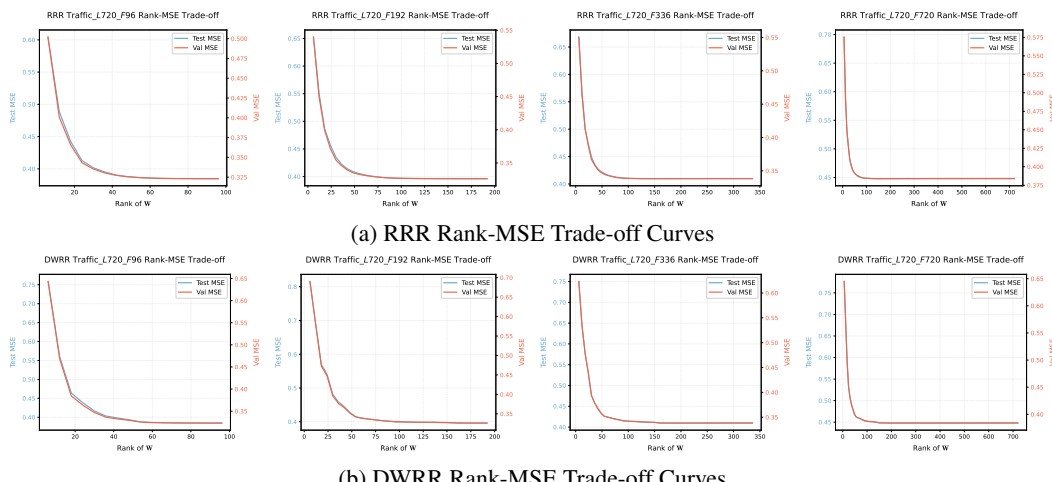

Figure 26: Rank-MSE Trade-off Curves on Traffic. Although the curves for these larger datasets appear flat at this scale, numerical analysis reveals that the optimal model is not achieved at full rank, indicating that slight rank truncation remains beneficial despite data scaling.

