# OpenReview forum: "Characteristic Root Analysis and Regularization for Linear Time Series Forecasting"
_ICLR.cc/2026/Conference — ICLR 2026 Poster_

### Official Review · Reviewer_zgN9 · 2025-10-22

**Soundness:** 3
**Presentation:** 3
**Contribution:** 3
**Rating:** 6
**Confidence:** 4

**Summary:**

This paper advances a linear time-series forecasting framework centered on characteristic roots: in the noise-free regime, it shows that characteristic roots govern long-term dynamics; in noisy settings, it exposes the MSE training data-scaling law and the risk of spurious roots, indicating that simply enlarging the sample size is insufficient for robust generalization. Accordingly, the authors introduce two complementary structural regularization strategies: (i) low-rank constraints on the parameter matrix via Reduced-Rank Regression (RRR) and Direct Weight Rank Reduction (DWRR) to recover low-dimensional latent dynamics; and (ii) Root Purge, an adaptive loss that learns a noise null space and performs online root restructuring, with training applicable in either the time or frequency domain. Evaluated on benchmarks including Traffic, Electricity, Weather, Exchange, and ETT, the methods deliver strong results.

**Strengths:**

1. According to the reported results, the proposed methods achieve strong empirical performance across multiple datasets, which is particularly notable given the simplicity of the designs.

2. The paper provides substantial theoretical analysis with clear exposition to clarify the core concepts.

3. A clear roadmap is presented, and overall readability is high.

**Weaknesses:**

1. The core insight “preserving dominant characteristic roots to suppress noise” is not novel and aligns with fundamental conclusions from classical Singular Spectrum Analysis (SSA).

2. The technical contributions of DWRR and Root Purge appear simple.

3. The Root Purge objective may induce model degeneration: the model could prefer mapping any input to zero, and the paper offers no theoretical guarantees to prevent or mitigate this failure mode.

4. The theoretical analysis depends on i.i.d. Gaussian noise and stationarity assumptions to justify data efficiency and the regularizer; these assumptions often do not hold in practice, and no generalization beyond them is provided.

**Questions:**

1. Under complex, non-stationary noise typical of real-world series, do the proposed methods retain theoretical advantages?

2. Can the approach be extended to more complex deep learning architectures?

---

> ### Author Response · Authors · 2025-11-20
>
> Thank you very much for your valuable and insightful comments! Below, we address your concerns point by point.
>
> ---
> Weakness 1: The core insight “preserving dominant characteristic roots to suppress noise” is not novel and aligns with fundamental conclusions from classical Singular Spectrum Analysis (SSA).
> ---
>
> **Response.** Thank you for raising this point — it is an insightful observation, and we appreciate the opportunity to clarify the distinction. We have also provided a systematic discussion of SSA and related work in Appendix B.1.
>
> At first glance, our method may appear similar to SSA because both involve spectral analysis. However, the core principles, operational mechanisms, and theoretical motivations of the two approaches differ fundamentally.
>
> - **Singular Spectrum Analysis (SSA)**
> SSA is essentially an application of Taken's embedding theorem from dynamical-systems theory [1, 2]. When one cannot directly observe the full state-space evolution, Taken's theorem [3] states that a delay-coordinate embedding can reconstruct a space that is topologically conjugate to the true underlying dynamics. SSA uses this principle to perform Hankelization in the reconstructed state space.
>
>   During forecasting, SSA then requires de-Hankelization to enforce certain consistency constraints （e.g. overlapped part of the original series must be the same). However, this step is heuristic: it lacks theoretical guarantees and is known to introduce additional reconstruction error.
>
> - **Our Method**
> By contrast, our work identifies a different underlying principle of generalization in time-series learning: we leverage both
>     - **the analytic properties** of the governing differential equations (Corollary 1), and
>     - **the algebraic properties** of the governing differential equations (Corollary 2).
>
>   This perspective leads us to directly characterize how one should extract the true characteristic roots of the dynamics in the presence of noise. Unlike SSA, which operates in a reconstructed state space, we show that performing spectral analysis in the weight space is significantly more effective for isolating the dominant dynamical modes.
>
>   Furthermore, in our approach, Hankelization is not intended to reconstruct a hidden state space or approximate the original dynamics via a Taken-style embedding. Rather, it serves a much simpler and more direct role:
>     - to slice the raw time series into fixed-length training segments, and
>     - to express the optimization objective in a more compact and tractable form.
>
>   Thus, the Hankelization in our method is a data preprocessing and formulation device, not a dynamical reconstruction mechanism as in SSA.
>
> - **Summary of Key Differences**
> We summarize the key differences below:
>
> | Aspect | SSA | Our Method |
> |--------|-----|------------|
> | **Theoretical foundation** | Taken's embedding theorem | Analytic & algebraic properties of differential equations |
> | **Working space** | Reconstructed state space | Weight space of the learned operator |
> | **Use of Hankelization** | Reconstruct hidden dynamics | Slice raw time series for optimization |
> | **De-Hankelization** | Required; no theoretical guarantee | Not needed; not part of our pipeline |
> | **Objective** | Approximate state evolution | Identify true characteristic roots under noise |
>
> For these reasons, even though both approaches involve spectral tools, the essence, methodology, and objectives are substantially different. We thank the reviewer again for the valuable comment, and we hope this clarifies why our contribution is distinct from classical SSA.
>
> #### References
>
> 1. Golyandina, N., Nekrutkin, V., & Zhigljavsky, A. A. (2001). Analysis of time series structure: SSA and related techniques. CRC press.
> 2. Hassani, H. (2007). Singular spectrum analysis: methodology and comparison.
> 3. Noakes, L. (1991). The Takens embedding theorem. International Journal of Bifurcation and Chaos, 1(04), 867-872.

---

> ### Author Response · Authors · 2025-11-20
>
> ---
> Weakness 2: The technical contributions of DWRR and Root Purge appear simple.
> ---
>
> **Response.** Thank you for the comment. We genuinely appreciate this observation, and we would actually interpret it as a compliment rather than a weakness.
>
> - **Simplicity is a feature, not a flaw — consistent with engineering principles**
> From an engineering-philosophy perspective, we believe that the most elegant and effective solutions are often simple ones that capture the essence of the problem. The deep learning community has repeatedly witnessed this: Dropout, BatchNorm, RevIN, ResNet skip connections—all are conceptually simple, yet they solved foundational problems precisely because they distilled the core issue to its essence.
> Simplicity is in line with Occam's Razor: do not introduce unnecessary complexity. When a simple method reliably addresses a complex phenomenon, that typically indicates that the method is grounded in the right underlying principle.
>
> - **Our method is derived *ab initio* from first principles**
> Conceptually, our method is derived from first principles. Instead of starting from architectural heuristics or empirical design patterns, we return to the fundamental questions:
>     - What is the core difficulty of time-series forecasting?
>     - What is the essence of generalization in dynamical systems?
>
>   By analyzing the analytic and algebraic properties of differential equations (Corollary 1 and 2), we derive algorithms (DWRR and Root Purge) that look simple on the surface precisely because they are rooted in the correct theoretical primitives. Simplicity here is the consequence of identifying the right abstraction, not a lack of depth.
>
>   As in phenomenology, we aim to "return to the thing itself"—to the essential structure of the problem, rather than building layers of unnecessary machinery.
>
> - **Simplicity has long been associated with mathematical and scientific beauty**
> There is also a long intellectual tradition, both Eastern and Western, that views simplicity as a hallmark of truth:
>     - In ancient Greek philosophy, from Pythagoras to Plato, the highest natural laws were believed to be simple, harmonious, and mathematically expressive.
>     - In the Eastern tradition, the idea of **大道至简** ("the great way is supremely simple") captures thousands of years of philosophical and aesthetic wisdom.
>
>   By adopting the simplest operator that correctly captures the fundamental roots of the dynamics, DWRR and Root Purge follow this tradition of mathematical elegance.
>
> We sincerely thank the reviewer for raising this point. In our view, the fact that the final algorithms are simple is precisely what demonstrates the strength and clarity of the underlying theoretical insight. Simplicity, when achieved through first principles, is often the strongest indication that the method has captured the true nature of the problem.

---

> ### Author Response · Authors · 2025-11-20
>
> ---
> Weakness 3: The Root Purge objective may induce model degeneration: the model could prefer mapping any input to zero, and the paper offers no theoretical guarantees to prevent or mitigate this failure mode.
> ---
>
> **Response.** We sincerely thank the reviewer for raising this important point regarding potential model degeneration under the Root Purge objective. We would like to kindly argue first that model degeneration is __never__ observed with the reasonable $\lambda$ values used in our paper. Further, we kindly argue that over-regularization is a well-known challenge in machine learning, and it is **not specific to our method** and similarly affects widely used techniques such as L1/L2 regularization, dropout, spectral penalties, and norm-based constraints. As with any regularization strategy, the practical solution lies in controlling its strength rather than attempting to rule out extreme pathological cases entirely.
>
> In Root Purge, while a very large $\lambda$ could in principle drive the model toward trivial outputs, this behavior is no different from what occurs with overly strong classical regularizers. In practice, however, the goal is to use a **moderate $\lambda$** that enhances generalization while maintaining full expressive capacity. Our empirical studies consistently demonstrate that Root Purge is stable and beneficial across a broad range of reasonable hyperparameter settings.
>
> - **Empirical performance in Table 1**
> With appropriately chosen $\lambda$, Root Purge yields substantial performance gains across datasets and architectures. A degenerate solution would cause severe degradation, which we never observe.
>
> - **Ablation studies in Figure 2 and Table 13**
> Our ablations show that Root Purge provides improvements even for small values of $\lambda$, and maintains consistent benefits across a wide range. Performance only degrades when $\lambda$ becomes unrealistically large—precisely the same behavior observed with any strong regularizer. These results indicate that Root Purge does not inherently push the model toward trivial outputs under typical usage.
>
> - **Singular value dynamics in Figure 3**
> By analyzing the evolution of singular values of the weight matrix $\mathbf{W}$, we observe that the dominant singular values remain stable even under relatively strong regularization. This directly counters the concern that Root Purge collapses the model by shrinking all singular values to zero. Instead, Root Purge selectively suppresses unstable modes while preserving the core representational capacity.
>
> In summary, while extreme regularization can theoretically cause degeneration, this behavior is neither unique to Root Purge nor observed in any realistic setting. Our empirical results, ablation studies, and singular-value analyses collectively demonstrate that Root Purge is robust, stable, and effective when used with reasonable hyperparameter values.

---

> ### Author Response · Authors · 2025-11-20
>
> ---
> Weakness 4: The theoretical analysis depends on i.i.d. Gaussian noise and stationarity assumptions to justify data efficiency and the regularizer; these assumptions often do not hold in practice, and no generalization beyond them is provided.
> ---
>
> **Response.** We thank the reviewer for this important observation regarding the assumptions underlying our theoretical analysis.
>
> - **Purpose of the Gaussian and stationarity assumptions**
> In Section 3.2 (Proposition 1), we adopt the i.i.d. Gaussian noise and stationarity assumptions primarily to **facilitate analytical tractability**. These assumptions allow us to derive a closed-form expression that elucidates the noise sensitivity and scaling properties of characteristic roots. Without them, a rigorous closed-form analysis is not feasible.
>
> - **Relaxation of Gaussianity**
> Upon revisiting the proof in Appendix D.3 and Proposition 1, we note that the Gaussian assumption can be **relaxed to any zero-mean noise with finite second-order moments**. We have revised the manuscript to clarify this point and emphasize that the theoretical insights hold under a broader class of noise distributions.
>
> - **Algorithmic robustness in practice**
> Importantly, in both our algorithm design and experimental evaluation, **no assumptions on the data distribution are required**. Root Purge and rank-regularization operate directly on the characteristic roots of the estimated linear model, making the approach fully applicable to real-world time series, including those with non-Gaussian, heteroscedastic, or non-stationary noise.
>
> We have updated the manuscript to explicitly highlight these clarifications, demonstrating that our theoretical analysis provides insightful guidance while our method remains practically robust across diverse datasets.

---

> ### Author Response · Authors · 2025-11-20
>
> ---
> Question 1: Under complex, non-stationary noise typical of real-world series, do the proposed methods retain theoretical advantages?
> ---
>
> **Response.**
>
> Thank you for this insightful question, which allows us to extend the discussion from the previous point. We fully agree that real-world time series often exhibit complex, non-stationary noise patterns.
>
> Building directly from our previous response, where we clarified that **our theoretical insights hold for any zero-mean noise with finite moments** and that our algorithm is entirely **distribution-agnostic**, we argue that our methods retain their advantages for the following key reasons:
>
> 1. **Noise-Agnostic Regularization Framework**:
>    Both our proposed methods—Root Purge and rank reduction techniques—are fundamentally **distribution-agnostic**. They do not assume specific noise properties (Gaussianity, stationarity, etc.). Instead, they operate on the principle that:
>    - Noise typically manifests as **spurious characteristic roots** and contributes to **small singular values** in the learned dynamics
>    - By promoting low-rank structure or learning a noise-suppressing null space, our methods **filter out components that do not generalize**, regardless of their statistical distributions.
>
>
> 2. **Empirical Validation Across Diverse Real-World Settings**:
> Our extensive experiments across **8 real-world benchmarks** (__Table 1, Appendix E.4__) include datasets with inherently complex, non-stationary noise profiles (See Figure 12 for some examples). The consistent superiority of our methods—particularly on smaller or noisier datasets like Exchange and ETT—provides strong evidence that they **retain advantages under realistic, non-ideal noise conditions**.

---

> ### Author Response · Authors · 2025-11-20
>
> ---
> Question 2: Can the approach be extended to more complex deep learning architectures?
> ---
>
> **Response.**
>
> We sincerely appreciate the reviewer's feedback regarding the generalization of our methods on non-linear models. This brings up a broader and thought-provoking direction: how to generalize the idea of rank reduction to nonlinear architectures. Unlike in linear models, where spectral properties offer clear guidance, the definition and utility of "low-rank structure" in deep nonlinear models remain underexplored. Investigating principled ways to impose and leverage low-rank constraints in such models—potentially via architectural design, regularization, or representation learning—is a compelling area for future work, and we have noted it as one of the directions we plan to pursue.
>
> To empirically probe this question, we applied DWRR to Transformer and PatchTST models by inserting DWRR into their linear blocks to reduce effective rank. The results are shown below:
>
> |Dataset|Horizon|Transformer|PatchTST|
> |-|-|-|-|
> | ETTh1|96|0.573→0.553|0.404→0.404|
> | |192|0.555→0.522|0.437→0.436|
> | |336|0.637→0.565|0.453→0.453|
> | |720|0.755→0.676|0.481→0.477|
> | ETTh2|96|0.348→0.344|0.290→0.290|
> | |192|0.473→0.469|0.352→0.351|
> | |336|0.474→0.473|0.373→0.371|
> | |720|0.519→0.496|0.406→0.403|
> | ETTm1|96|0.376→0.356|0.298→0.296|
> | |192|0.445→0.416|0.330→0.330|
> | |336|0.473→0.458|0.371→0.369|
> | |720|0.530→0.526|0.419→0.417|
>
> As for Root Purge, the situation is more nuanced (For this section, we kindly ask the reviewer to take $z_{\text{fut}} = \hat{y}_{\text{fut}}$ as we cannot get the equation compliation to work properly):
>
> 1. Since there is no direct analogue of the rank–nullity principle in nonlinear settings, we apply Root Purge only to the final linear projection layer.
> 2. In the linear model, Root Purge follows the formulation in __Equation (3)__. Since we assume we cannot forecast future noise, the model forecast ($z_{\text{fut}}$) and the noisy ground truth ($y_{\text{fut}}$) can form a pair where their difference estimates the noise on the signal. Extending this to nonlinear architectures requires feeding $z_{\text{fut}}$ and $y_{\text{fut}}$ back into the model to obtain a pair of latent encoding, so $z_{\text{fut}}$ must span at least a full lookback window. Consequently, results are available only for forecasting horizons greater than or equal to the lookback length.
>
> We have summarized our results on a MLP model as follows:
>
> |Dataset|Horizon|MLP|
> |-|-|-|
> | ETTh1|336|0.438→0.429|
> | |720|0.455→0.447|
> | ETTh2|336|0.381→0.368|
> | |720|0.413→0.408|
> | ETTm1|336|0.364→0.360|
> | |720|0.428→0.424|
>
>
> These results indicate that our methods can yield performance improvements even in more complex architectures, suggesting meaningful potential for generalization to nonlinear cases.
>
> That said, we respectfully highlight several reservations from theoretical and practical perspectives:
> - **Lack of clear physical interpretation**: Our methods are primarily motivated by linear prediction, where input–output relationships are governed by a well-defined linear operator and can be analyzed via characteristic roots, linearity, and noise sensitivity. In contrast, the linear submodules of deep nonlinear architectures serve mainly as feature aggregators rather than physically meaningful linear mappings. As a result, applying our methods in these settings currently lacks a solid theoretical grounding.
> - **Model selection efficiency**: In linear models, DWRR can be hyperparameter-tuned efficiently through fast post-training cross-validation. For nonlinear models, this tuning becomes substantially more computationally expensive, limiting practicality.

---

### Official Review · Reviewer_G42f · 2025-10-31

**Soundness:** 3
**Presentation:** 4
**Contribution:** 3
**Rating:** 8
**Confidence:** 2

**Summary:**

This work presents a comprehensive study of linear models for time series forecasting, focusing on the role of characteristic roots in shaping model expressivity. Experiments validate our theoretical claims and demonstrate the effectiveness of both methods across a range of forecasting tasks.

**Strengths:**

1. The research motivation of this manuscript is clear and its content is very rich。

2. This manuscript is theoretical.

3. Sufficient experiments have verified the effectiveness of the proposed method.

**Weaknesses:**

1. Theoretical analysis is an advantage of this work, but it can easily make it difficult for readers to understand. It is necessary to provide intuitive explanations in the key derivation process

2. The manuscript lacks a comprehensive discussion of related work, which hampers readers, particularly those less familiar with the domain. Thus, it remains unclear how the proposed approach advances the state of the art or distinguishes itself from existing methods.

**Questions:**

1. What is the transferability of the method?Can linear models be used in other areas than time series learning?

2. Why is the frequency-domain linear layer necessary for Root-Purge instead of a mere implementation detail?

---

> ### Author Response · Authors · 2025-11-20
>
> Thank you very much for your valuable and insightful comments! Below, we address your concerns point by point.
>
> ---
> Weakness 1: Theoretical analysis is an advantage of this work, but it can easily make it difficult for readers to understand. It is necessary to provide intuitive explanations in the key derivation process
> ---
>
> **Response:**
>
> We thank the reviewer for this valuable suggestion. In response, we have substantially improved the accessibility of the theoretical section. Specifically, we have added
> 1. a mind map summarizing the key concepts and their relationships in __Appendix A, Figure 5__, and
> 2. a concise, reader-oriented summary of the main theoretical results in __Appendix A, Table 4__.
>
> Given the length of the paper (65 pages), we additionally introduced a detailed **table of contents** to help readers navigate the derivations and theoretical arguments more intuitively. We hope these additions make the theoretical flow clearer and easier to follow.

---

> ### Author Response · Authors · 2025-11-20
>
> ---
> Weakness 2: The manuscript lacks a comprehensive discussion of related work, which hampers readers, particularly those less familiar with the domain. Thus, it remains unclear how the proposed approach advances the state of the art or distinguishes itself from existing methods.
> ---
>
> **Response.** We thank the reviewer for emphasizing the need for a comprehensive discussion of prior work.
>
> While the “Related Work” section in the main text is concise, we provide an extensive discussion in Appendix B.1, which situates our method relative to multiple prior approaches. Here we summarize the key distinctions below:
>
> - **Transformer-based and deep time-series models** focus on capturing complex non-linear dependencies and long-range patterns using attention mechanisms and seasonal-trend decomposition. In contrast, our method leverages linear characteristic root analysis to identify and regularize spurious roots in linear models, offering a principled and interpretable mechanism to improve long-term forecast robustness without relying on large neural architectures.
>
> - **Minimalist and linear models** emphasize efficiency by separating trend and seasonality or exploiting low-dimensional linear structure. Our work goes further by analyzing the full spectrum of characteristic roots and systematically purging those corresponding to noise or unstable dynamics, which improves both stability and accuracy over standard linear decompositions.
>
> - **Spectral and low-rank methods** (e.g., SSA) exploit low-rank structure or singular value decomposition for denoising and dimensionality reduction. Our approach extends these ideas by directly connecting low-rank recovery with characteristic roots, using Root Purge to remove spurious roots while preserving true latent dynamics, providing a theoretically grounded and practically effective regularization strategy.
>
> - **Koopman operator-inspired methods** extract linear representations from nonlinear dynamics via operator-theoretic frameworks. Unlike these approaches, our method remains within a linear time-series forecasting paradigm, but leverages characteristic root analysis to achieve robust long-term prediction and interpretability with minimal model complexity.
>
> In the revised manuscript, we now explicitly direct readers from the main text to __Appendix B.1__, highlighting how our method builds on and differs from prior work.
>
> We appreciate the reviewer’s feedback, which allows us to make these distinctions more visible and accessible to all readers, including those less familiar with the domain.

---

> ### Author Response · Authors · 2025-11-20
>
> ---
> Question 1: What is the transferability of the method? Can linear models be used in other areas than time series learning?
> ---
>
> **Response.** Thank you for the thoughtful questions. Since the second question concerns the generality of linear models, we address that one first.
>
> - **On the Generality of Linear Models Beyond Time-Series Learning**
> Regarding the second question, the broad applicability of linear models is well-established in both classical machine learning and statistics. Linear regression, PCA, and related techniques offer solid theoretical grounding, interpretability, and well-understood optimization properties. In Appendix B.1, we briefly survey the development of linear prediction theory, which has historically played a central role in many areas of data analysis.
>
>   Importantly, the relevance of linear models has not diminished in the deep learning era. On the contrary, linear components are fundamental to a wide range of modern techniques, such as:
>     - **LoRA** [1] - low-rank adaptation of large language models;
>     - **Linear probing** [2] - model analysis through linear classifiers;
>     - **Model editing** [3] - modifying model behavior via linear interventions;
>     - **Model pruning** [4] - compressing models using linear constraints.
>
>   Furthermore, recent work has taken a differential-geometric perspective and defined new operations under which neural networks become linear [5]. These developments collectively reinforce that linear structures remain central to contemporary machine-learning methodologies.
>
> - **On the Transferability of Our Method**
> The transferability of our method can be discussed from two complementary perspectives:
>
>     1. **Generality of Spectral Methods**
>     In the narrow sense, the spectral method we use is based on SVD. Its utility extends far beyond time-series analysis. SVD and related spectral tools are widely used in:
>         - Signal processing
>         - Recommendation systems
>         - Community detection
>         - Ranking problems
>         - Structured matrix recovery
>
>         These applications rely on spectral methods' ability to uncover latent low-rank or orthogonal structures that are intrinsic to a wide variety of datasets. We provide a more detailed discussion of spectral methods in __Appendix B.1__, and we also refer the reviewer to [6], which offers an excellent introductory overview of the theoretical and practical power of spectral techniques.
>
>     2. **Generality of Adaptive Decomposition**
>
>        A second key aspect is the adaptive decomposition that our method performs. Our approach learns an operator whose null space is aligned with the intrinsic structure of the data—crucially, without requiring any prior knowledge of that structure. This adaptivity makes the method fundamentally data-driven: any scenario where one aims to identify or exploit a meaningful null space, or more broadly a latent linear structure, stands to benefit from our approach. Thus, any task that involves discovering linear constraints, learning low-rank projections, or extracting structured subspaces from data is a potential application area. This includes far more than time-series learning.
>
> We sincerely appreciate the reviewer's question. We believe our method has substantial potential for broader application across multiple domains, precisely because it combines the universality of spectral techniques with the flexibility of adaptive, data-driven operator learning.
>
> #### References
>
> 1. Hu, E. J., Shen, Y., Wallis, P., Allen-Zhu, Z., Li, Y., Wang, S., ... & Chen, W. (2022). Lora: Low-rank adaptation of large language models. ICLR, 1(2), 3.
> 2. Sorscher, B., Geirhos, R., Shekhar, S., Ganguli, S., & Morcos, A. (2022). Beyond neural scaling laws: beating power law scaling via data pruning. Advances in Neural Information Processing Systems, 35, 19523-19536.
> 3. Meng, K., Sharma, A. S., Andonian, A. J., Belinkov, Y., & Bau, D. Mass-Editing Memory in a Transformer. ICLR, 2023.
> 4. Hao, J., Huang, Q., Liu, H., Xiao, X., Ren, Z., & Yu, J. (2025). A Token is Worth over 1,000 Tokens: Efficient Knowledge Distillation through Low-Rank Clone. arXiv preprint arXiv:2505.12781.
> 5. Berman, N., Hallak, A., & Shocher, A. (2025). Who Said Neural Networks Aren't Linear?. arXiv preprint arXiv:2510.08570.
> 6. Chen, Y., Chi, Y., Fan, J., & Ma, C. (2021). Spectral methods for data science: A statistical perspective. Foundations and Trends® in Machine Learning, 14(5), 566-806.

---

> ### Author Response · Authors · 2025-11-20
>
> ---
> Question 2: Why is the frequency-domain linear layer necessary for Root-Purge instead of a mere implementation detail?
> ---
>
> **Response.** We thank the reviewer for this insightful question regarding the necessity of the frequency-domain linear layer in Root Purge.
> - **Practical Motivation**
> The frequency-domain linear layer has been adopted in recent time series works such as FITS [1] and FreTS [2], which demonstrate **improved training stability and empirical performance**. We have also confirmed this observation in our experiments: see **Table 11** in the Appendix, where frequency-domain linear layers show consistent advantages in gradient-based Root Purge training. Since Root Purge relies on gradient-based optimization, we follow this design choice to leverage these practical benefits.
>
> - **Effectiveness in the Time Domain**
> Root Purge is not inherently tied to the frequency domain. We conducted parallel experiments using a **time-domain linear layer**, reported in **Appendix E.7**, and observed **strong performance improvements** as well. This indicates that the algorithm itself is robust and effective regardless of whether the linear layer operates in the time or frequency domain.
>
>
> - **Implicit Bias and Training Dynamics**
> Finally, while the detailed investigation of training dynamics is potentially beyond the scope of this paper, we note that the observed empirical differences may arise from **implicit biases introduced by gradient descent**. Similar phenomena have been studied in the context of deep linear networks (networks with identity activations) as discussed in [4].
>
> In summary, the frequency-domain linear layer is **not strictly necessary** for Root Purge to be effective, but it provides practical advantages for gradient-based optimization. Both theoretical and empirical evidence supports that Root Purge is robust across domain choices.
>
> #### Reference
> 1. Xu, Z., Zeng, A., & Xu, Q. FITS: Modeling Time Series with $10 k$ Parameters. In The Twelfth International Conference on Learning Representations.
> 2. Yi, K., Zhang, Q., Fan, W., Wang, S., Wang, P., He, H., ... & Niu, Z. (2023). Frequency-domain MLPs are more effective learners in time series forecasting. Advances in Neural Information Processing Systems, 36, 76656-76679.
> 3. Toner, W., & Darlow, L. N. (2024, July). An Analysis of Linear Time Series Forecasting Models. In International Conference on Machine Learning (pp. 48404-48427). PMLR.
> 4. Vardi, G. (2023). On the implicit bias in deep-learning algorithms. Communications of the ACM, 66(6), 86-93.

---

> > ### Comment · Reviewer_G42f · 2025-11-28
> > **I still maintain a positive score**
> >
> > Thanks to the authors for their careful feedback, they have addressed my concerns. I still maintain a positive score.

---

### Official Review · Reviewer_xcu7 · 2025-10-31

**Soundness:** 2
**Presentation:** 2
**Contribution:** 3
**Rating:** 4
**Confidence:** 4

**Summary:**

This paper analyzes linear time series forecasting via characteristic roots , identifying that noise creates "spurious roots" and a "data scaling law" that demands excessive data to mitigate this noise. To improve robustness and data efficiency, it proposes two structural regularization strategies: (1) post-hoc Rank Reduction (RRR) and (2) a novel "Root Purge" method. Root Purge is an adaptive training regularizer that encourages the model to learn a null space for the estimated noise (residuals).

**Strengths:**

1.Originality and Clarity: The "Root Purge" loss function is a novel, elegant, and intuitive regularization technique.

2.Quality and Significance: The paper provides a valuable insight into why simple linear models fail in noisy, low-data regimes.

**Weaknesses:**

1.Hyperparameter Sensitivity: The methods' performance relies heavily on tuning $\rho$ or $\lambda$. This data-specific tuning is difficult to estimate a priori and undermines the claim of robustness.

2.Contingent on Low-Rank Assumption: The methods are motivated by an assumption that the true signal is low-rank. However, on large, complex datasets, the paper's results show the optimal rank is often full , and the proposed methods offer minimal benefit . This limits the methods' generality.

3.Theory-Practice Gap (Trends): The theoretical analysis assumes homogeneous equations (constant/zero bias) . This is a poor fit for real-world data with non-stationary, dynamic trends.

**Questions:**

1. Does the Low-Rank Assumption limit the generality of method, making it effective only for a specific subset of (simpler) time series datasets?

2.If we encounter a complex trend in the future that Instance Normalization can't flatten, does your Root Purge method just fail? How can you convince us that it's still robust in that scenario?

---

> ### Author Response · Authors · 2025-11-20
>
> Thank you very much for your valuable and insightful comments! Below, we address your concerns point by point.
>
> ---
> Weakness 1: **Hyperparameter Sensitivity**: The methods' performance relies heavily on tuning $\rho$ or $\lambda$. This data-specific tuning is difficult to estimate a priori and undermines the claim of robustness.
> ---
>
> **Response:** We sincerely thank the reviewer for raising this point. We would like to clarify that our methods actually exhibit low hyperparameter sensitivity in practice:
>
> - **Only a single hyperparameter is required for each model.**
> $\rho$ is the __only__ hyperparameter for RRR and DWRR, and $\lambda$ is the __only__ hyperparameter for Root Purge. This keeps the search space minimal and avoids the complexity typically associated with multi-parameter tuning in time-series forecasting models.
>
> - **For RRR and DWRR: $\rho$ is selected via post-training cross-validation rather than manual tuning.**
> Since $\rho$ is determined automatically using a simple  validation procedure with **Rank-MSE trade-off** on a validation set, it does not require problem-specific heuristics or extensive hyperparameter sweeps. This makes the procedure practical and reproducible across datasets. We have showcased some representative examples for the Rank-MSE trade-off curves in **Figures 15 and 16 in Appendix E**.
>
> - **For Root Purge, the model is not sensitive to the choice of $\lambda$.**
> We observe that $\lambda$ is effective even at very small values (e.g., 1/32) and it offers benefit for $\lambda$ up to, e.g., 1/2. Additionally, the model performance follows a clear and smooth trend: first increasing and then decreasing within a reasonable range (e.g., $\lambda\in\{1/32, 1/16, 1/8, 1/4, 1/2, 1, 2\}$); see __Figure 2__. More importantly, as a rule of thumb, a good choice of lambda is usually one of $\{0.125, 0.25, 0.5\}$, aligning with the setting we used in the main experiment. As a result, the optimal value can be identified reliably and with minimal search effort.
>
> Overall, we believe these properties indicate that our approach is stable and easy to tune, rather than heavily reliant on dataset-specific hyperparameters.

---

> ### Author Response · Authors · 2025-11-20
>
> ---
> Weakness 2: **Contingent on Low-Rank Assumption**: The methods are motivated by an assumption that the true signal is low-rank. However, on large, complex datasets, the paper's results show the optimal rank is often full , and the proposed methods offer minimal benefit. This limits the methods' generality.
> ---
>
> **Response.** Thank you for this thoughtful comment. We apologize for any confusion caused by the presentation in the initial submission. After re-examining both the RRR and DWRR experiment logs (included in the original __supplementary zip__, under `Rank_Reduction` folder with logs `RRR.log` and `DWRR.log`), we confirm that the optimal ranks tends to be below full rank even for complex data such as ECL and Traffic. For convenience, we summarize the results below. We have also revised our manuscript to prevent further confusion.
>
> | Data   | Horizon | RRR optimal rank | DWRR optimal rank |
> |--------|---------|-----|------|
> | ECL    | 96      | 96  | 96   |
> |        | 192     | 180 | 192  |
> |        | 336     | 204 | 240  |
> |        | 720     | 294 | 210  |
> | Traffic| 96      | 96  | 96   |
> |        | 192     | 172 | 192  |
> |        | 336     | 144 | 174  |
> |        | 720     | 150 | 198  |
>
> In addition, we address the concern from several perspectives.
>
> - **Low-rank structure is a well-established assumption in time-series modeling**
> The low-rank assumption is not specific to our work; rather, it is a foundational principle that has been extensively used across time-series analysis and signal processing [1,2]. Many classical models, such as ARMA processes, seasonal patterns, and systems with smooth dynamics, exhibit strong low-dimensional structure in their predictive operators or latent states. We survey the relevant literature in __Appendix B.1__ and explain why low-rank structure appears naturally in many practical time-series settings.
>
> - **"Low-rank" is a relative notion in linear prediction**
> Importantly, rank must be understood relative to model capacity, not as an absolute property. In linear prediction, we can choose a longer lookback window to increase model capacity. Furthermore, by adopting individual channel modeling, we can effectively extend the degree of freedom /rank to enhance model capacity, as demonstrated by the results in __Table 2__, where Root Purge brings significantly more improvements than in CI setting. Under such settings, "low-rank" refers to the structure of the signal relative to this enlarged capacity. The effective rank that matters is therefore problem-dependent and adjustable, constrained mainly by computational budget rather than theoretical limitations.
>
> - **Even complex dynamics admit effective reduced-order models**
> Even when the underlying dynamics are highly complex or full-rank in an absolute sense, reduced-order modeling is a standard and powerful approach. Many real-world high-dimensional dynamics systems are governed by extremely can be approximated well by low-dimensional representations. e.g.,
>     - Newtonian mechanics approximating general relativity;
>     - Navier–Stokes approximations for fluid systems;
>     - Koopman operator truncations in nonlinear dynamics.
>
>   Our method leverages this same principle.
>
> - **In practice, the "true rank" is unknowable — low-rank is a modeling device**
> For real-world datasets, we never have access to the true underlying dynamics or its exact rank. In this sense, the idea of a "true low rank" is not falsifiable and should be viewed as a modeling abstraction rather than an observable property. This aligns with the long-standing philosophical distinction between instrumentalism and scientific realism, where practical models are valued for their predictive utility even when the underlying mechanisms cannot be fully known.
>
>   All algorithms in machine learning rely on such abstractions to represent a complex world in a tractable mathematical form. Low-rank structure is one of these abstractions, and it has repeatedly proven both useful and empirically effective.
>
> - **Empirical results demonstrate robustness even on large, complex datasets**
> Our experiments show that we achieve state-of-the-art performance on nearly all datasets, including those in the "large, complex" category. This provides strong empirical evidence that our method is not dependent on an overly restrictive notion of low rank. Instead, it remains robust and effective across diverse real-world conditions.
>
> #### References
>
> 1. Liu, G. (2022). Time series forecasting via learning convolutionally low-rank models. IEEE Transactions on Information Theory, 68(5), 3362-3380.
> 2. Agarwal, A., Amjad, M. J., Shah, D., & Shen, D. (2018). Model agnostic time series analysis via matrix estimation. Proceedings of the ACM on Measurement and Analysis of Computing Systems, 2(3), 1-39.

---

> ### Author Response · Authors · 2025-11-20
>
> ---
> Weakness 3: Theory-Practice Gap (Trends): The theoretical analysis assumes homogeneous equations (constant/zero bias) . This is a poor fit for real-world data with non-stationary, dynamic trends.
> ---
>
> **Response.** We sincerely thank the reviewer for raising this important point regarding the applicability of our homogeneous formulation to real-world data with non-stationary trends. We clarify below that the theoretical framework based on homogeneous linear difference equations is in fact **fully expressive** for the trend components relevant in practice. Our response is grounded in two well-established facts in linear systems theory.
>
> 1. **Nonhomogeneous linear systems can always be rewritten as homogeneous systems of higher order**
> Consider the nonhomogeneous linear recurrence
> \begin{equation}
> y_t+a_1 y_{t-1}+\cdots+a_n y_{t-n}=f(t).
> \end{equation}
> Suppose the forcing term  $f(t)$ itself satisfies a linear homogeneous difference equation of order $m$:
> \begin{equation}
> f(t)+b_1 f(t-1)+\cdots+b_m f(t-m)=0.
> \end{equation}
> This setup covers all structured trend families commonly used in practice—**polynomials, exponentials, sinusoids, and their finite sums**.
>
>     By successively shifting the original equation and substituting out all appearances of $f(t-k)$ using the above relation, one obtains a **single homogeneous linear difference equation of order** $n+m$ involving only $y_t$. This is a classical construction in linear recurrence theory.
>
>     **Implication**:
> Any linear forecasting model with a structured trend component (seasonal, polynomial, exponential, oscillatory, etc.) is **exactly representable** as a homogeneous recurrence after a finite increase in order. All trend types examined in Corollary 1 fall within this category.
>
> 2. **Arbitrary continuous trends can be approximated by solutions of homogeneous systems**
>
>     To address more general real-world nonstationary trends, we invoke a standard approximation principle based on the following theorem:
>
>     **Stone–Weierstrass Approximation Theorem:** Any continuous function on a compact interval can be uniformly approximated by polynomials.
>
>     Since a polynomial of degree $d$ satisfies $\Delta^{d+1} P_d(t) = 0$, it solves a homogeneous linear difference equation of finite order. Therefore, we have the following:
>
>     **Corollary:** For any continuous forcing term $f(t)$ and any $\epsilon>0$, there exists a polynomial $P_d(t)$—hence a function satisfying a homogeneous difference equation—such that $\sup_t |f(t) - P_d(t)| < \epsilon.$
>
>     Replacing $f(t)$ with $P_d(t)$ yields a linear system that is:
>
>     - **arbitrarily close** to the original nonhomogeneous system in prediction error, and
>     - **exactly reducible** to a higher-order homogeneous system.
>
>     Thus, **homogeneous linear difference equations form a universal approximating family for models with continuous trend components.**
>
> 3. **Practical relevance to forecasting models**
> State-of-the-art linear forecasting architectures implicitly rely on the fact that trends are well captured by polynomial, exponential, or sinusoidal components or by low-order finite-dimensional linear generators.
>
>     Decades of empirical evidence show that these trend families are sufficient to model real-world nonstationarity across domains (economics, climate, energy, engineering, etc.).
>
>     Therefore, the homogeneous formulation underlying our theoretical analysis is **not an idealized simplification**, but rather:
>     - **aligned with the functional families used in modern forecasting practice**,
>     - **expressive enough to capture real-world nonstationary dynamics**, and
>     - **mathematically general**, in that any structured trend (and any continuous trend, up to arbitrary precision) falls within its representational capacity.
>
> **Conclusion:**
>
> - Any nonhomogeneous linear recurrence with a structured trend is **exactly** reducible to a homogeneous recurrence of higher order.
> - Any continuous trend can be approximated **arbitrarily well** by a function satisfying a homogeneous linear difference equation (via Stone–Weierstrass Approximation Theorem).
> - Consequently, the homogeneous assumption in our theoretical analysis is **broad, expressive, and consistent with established forecasting practice**, rather than restrictive.
>
> We thank the reviewer again for raising this important point and hope the clarification resolves their concern.

---

> ### Author Response · Authors · 2025-11-20
>
> ---
> Question 1: Does the Low-Rank Assumption limit the generality of method, making it effective only for a specific subset of (simpler) time series datasets?
> ---
>
> **Response.** Please refer to our response to the Weakness 2 above.

---

> ### Author Response · Authors · 2025-11-20
>
> ---
> Question 2: If we encounter a complex trend in the future that Instance Normalization can't flatten, does your Root Purge method just fail? How can you convince us that it's still robust in that scenario?
> ---
>
> **Response:**
>
> We sincerely thank the reviewer for raising this important question regarding the robustness of our method when Instance Normalization (IN) may not fully “flatten” complex trends. We interpret “flatten” as “remove trend,” and please kindly let us know if this interpretation differs from your intention. We would like to offer several clarifications.
>
> - **IN is not designed to remove trends, and it does not succeed at doing so even in simple cases**
> As originally proposed in [1], IN is primarily intended to mitigate distribution shift by removing the mean/variance discrepancy across segments. It is not a trend-removal mechanism.
> Indeed, IN cannot fully remove even simple deterministic trends (e.g., exponential or high-order polynomial). This is also reflected empirically in NLinear [2], where replacing IN with a simple “subtract-the-last-value” operation yields similar performance. This shows that IN’s ability to “flatten trends” plays a limited role in practice, since even if we weaken the trend removal capability of IN, it remains competitive.
>
> - **Theoretical role of IN: it always inserts a fixed unit root**
> In our analysis (__Remark 1; Appendix C.4__), we show that IN’s effect in linear models is equivalent to imposing a fixed characteristic root at 1, corresponding to shift-invariance. Formally, IN forces the general solution to take the form $$y_t = f(t) + C$$ where the value of $C$ is being lively determined. This is a hard constraint on the characteristic polynomial and does not depend on the complexity of the underlying trend. Thus, IN does not “fail” in the sense relevant to our method: it always guarantees the presence of a __fixed__ unit root and therefore always performs its theoretical function.
>
> - **Root Purge operates independently of IN**
> Root Purge focuses exclusively on the __modifiable__ characteristic roots. Its purpose is to identify and suppress roots associated with noise or spurious temporal patterns. Since IN only affects the single __fixed__ root at 1, and has no impact on the remaining roots learned by the model, Root Purge operates on a completely separate part of the spectrum. Therefore, even if IN imperfectly removes trends, Root Purge continues to function as intended.
>
> - **Empirical Evidence on the Robustness of Root Purge**
> As shown in __Table 1__, Root Purge consistently achieves strong performance across datasets with varying levels of non-stationarity. For example, in ETTh1, there are strong drifts and complex trends too complex to be removed by IN as previously discussed, yet our method remains robust and achieves state-of-the-art results. This confirms that Root Purge does not depend on IN successfully flattening complex structure.
>
> #### Reference
>
> 1. Kim, T., Kim, J., Tae, Y., Park, C., Choi, J. H., & Choo, J. (2021, May). Reversible instance normalization for accurate time-series forecasting against distribution shift. In International conference on learning representations.
> 2. Zeng, A., Chen, M., Zhang, L., & Xu, Q. (2023, June). Are transformers effective for time series forecasting?. In Proceedings of the AAAI conference on artificial intelligence (Vol. 37, No. 9, pp. 11121-11128).

---

### Author Response · Authors · 2025-11-27

---
Overall Response
---

We thank all reviewers for their valuable feedback and constructive comments, which have significantly improved the clarity and quality of our manuscript. We are encouraged that the reviewers recognized our work as a novel and important contribution to the field. A summary of their key observations is provided below:

* Reviewer **xcu7** described the Root Purge loss as "a novel, elegant, and intuitive regularization technique" and acknowledged that our paper offers "a valuable insight" into the failures of linear time series models in noisy, low-data regimes.

* Reviewer **G42f** highlighted the "clear motivation" and "rich theoretical content" of the manuscript, noting that "sufficient experiments" confirm the method’s effectiveness.

* Reviewer **zgN9** acknowledged the "strong empirical performance" achieved through "simple designs", supported by "substantial theoretical analysis" presented with clarity and high readability.

We have addressed each reviewer’s concerns in detail below their respective reviews. The following updates have been incorporated into the manuscript:

1. We have updated Appendix A.1 to include **a detailed mind-map**, added a **summary table of theorems** (Table 4), and provided a **content table** (page 14) for easier navigation through the paper. These additions are intended to enhance readability and flow, particularly in facilitating a quick understanding of the theoretical insights.

2. We have extended **Proposition 1** (lines 213–215) and **Proposition 6** (Appendix D.3) to **apply to any zero-mean noise with finite second moments**. Since our proofs rely on the law of large numbers, the central limit theorem, and Slutsky’s theorem—all of which are distribution-agnostic—our conclusions hold under these broader conditions.

3. We have **clarified concerns regarding the rank of complex datasets** such as Electricity and Traffic (lines 2910–2915, 2994-3001, 3469–3495) by including numerical analyses from RRR/DWRR experiment logs in the supplementary material. This clarification aims to demonstrate the commonality of low-rank structures and why a default regularization can potentially be beneficial for time series forecasting.

---

### Author Response · Authors · 2025-12-03
**Summary of Rebuttal to AC**

Dear AC,

Thank you for handling the reviews and taking the time to read our rebuttal. To help streamline your assessment, we provide below a concise summary of each reviewer’s comments, our responses, and the key outcomes from our discussions.


## Reviewer xcu7

| Concerns | Our Responses/Arguments |
| :- | :- |
| W1: Hyperparameter Sensitivity | Each method (RRR, DWRR, Root Purge) uses only one hyperparameter. RRR/DWRR tuning is post-training and computationally inexpensive. Root Purge sensitivity is empirically validated (Figure 2). |
| W2/Q1: Low Rank Assumption & Generality | Experiment logs show the optimal rank is rarely full. Low-rank is a flexible modeling device whose meaning depends on model capacity (e.g., INC or longer lookback → effectively higher attainable rank). Lots of prior works confirm the generality of low rank modeling and our strong empirical results show robustness of our methods on complex datasets. |
| W3: Focus on only homogeneous difference equations | Many non-homogeneous equations can be rewritten as higher-order homogeneous ones. By the Stone–Weierstrass theorem, homogeneous linear recurrences universally approximate dynamics, consistent with the strong performance of linear forecasting models in practice. |
| Q2: IN fail to flatten trend → Root Purge failure? | IN does not remove trends; it enforces a fixed unit root. Root Purge operates on modifiable roots only and is independent of IN. Empirically, it also performs well on real-world data with complex trends. |

__Discusion:__ We did not receive any feedback about our response before the rating was reverted.

---

## Reviewer G42f

| Concerns | Our Responses/Arguments |
| :- | :- |
| W1: Add Intuitive Explanation of Theories. | Added a mind map (Fig. 5), a theory summary table (Table 4), and a content table to improve conceptual flow and accessibility. |
| W2: Insufficient Related Work | Clarified that an expanded related-work discussion appears in Appendix B.1 and added a direct pointer in the main text for seamless navigation. |
| Q1: Transferability; use of linear models beyond time series | Highlighted that linear models underpin regression, PCA, LoRA, probing, editing, pruning, and linearized networks. Our method transfers broadly via spectral tools (SVD), adaptive decomposition, and applies to tasks involving null-space discovery, low-rank structure, and operator learning. |
| Q2: Why frequency-domain linear layer? | Clarified that it is optional but improves stability and gradient quality (FITS, FreTS; Table 11). Also demonstrated that a time-domain version works effectively (Appendix E.7); differences likely arise from implicit gradient descent bias rather than architectural necessity. |

__Discusion: Reviewer G42f acknowledged that our clarifications resolved their concerns, and maintained their score.__

---

## Reviewer zgN9

| Concerns | Our Responses/Arguments |
| :- | :- |
| W1: Similar to SSA; novelty unclear | We discuss our novelty from SSA in theoretical foundations, operating domain, use of Hankelization, and objective. Full side-by-side comparison is in Appendix B.1. |
| W2: DWRR & Root Purge seem simple | Simplicity is a feature, not a weakness—mirrored in RevIN, Dropout, ResNet. Methods arise directly from first-principles analysis, reflecting correct abstraction rather than superficiality. |
| W3: Root Purge may degenerate model | Empirically no degeneration observed. Like any regularizer, extreme hyperparameters can over-shrink, but our method is stable for reasonable $\lambda$, with strong evidence (Table 1; Figure 2-3). |
| W4/Q1: Gaussian & stationarity assumption on noise / Robustness under complex noise | Assumptions relaxed: proofs hold for any zero-mean finite-variance noise. Our methods are distribution-agnostic by design, with robustness verified empirically. |
| Q2: Extension to nonlinear models | Preliminary experiments show consistent gains for nonlinear models. Full theoretical grounding and tuning-cost reduction remain promising directions for future work. |

__Discusion:__ We did not receive any feedback about our response before the rating was reverted.

---

### Meta-Review · Area_Chair_wv1q · 2026-01-02

**Summary:**

The paper begins with a theoretical analysis of linear forecasting models, then develops novel methods based on the theoretical insights. Experiments conducted on multiple benchmark datasets demonstrate that the proposed approach outperforms current state-of-the-art methods. Most reviewers acknowledge the paper’s contributions and recommend acceptance. Reviewer xcu7 raised concerns regarding hyperparameter sensitivity, assumption validity, and practical value, which I believe have been adequately addressed in the author rebuttal. Therefore, I would recommend accepting the paper.

The following references, also cited in the rebuttal, should be included in the paper to support the proposed method. In particular, it appears that [1] also falls within the category of linear models.

[1] Liu, G. (2022). Time series forecasting via learning convolutionally low-rank models. IEEE Transactions on Information Theory, 68(5), 3362–3380.

[2] Agarwal, A., Amjad, M. J., Shah, D., & Shen, D. (2018). Model agnostic time series analysis via matrix estimation. Proceedings of the ACM on Measurement and Analysis of Computing Systems, 2(3), 1–39.

**Reviewer Scores:**

NA

---

### Decision · Program_Chairs · 2026-01-26

Accept (Poster)